# Universality in Transfer Learning for Linear Models

**Reza Ghane**[*]
Department of Electrical Engineering
California Institute of Technology
Pasadena, CA 91125
rghanekh@caltech.edu

**Danil Akhtiamov**[*]
Department of Computing + Mathematical Sciences
California Institute of Technology
Pasadena, CA 91125
dakhtiam@caltech.edu

**Babak Hassibi**
Department of Electrical Engineering
Department of Computing + Mathematical Sciences
California Institute of Technology
Pasadena, CA 91125
hassibi@caltech.edu

## Abstract

We study the problem of transfer learning and fine-tuning in linear models for both regression and binary classification. In particular, we consider the use of stochastic gradient descent (SGD) on a linear model initialized with pretrained weights and using a small training data set from the target distribution. In the asymptotic regime of large models, we provide an exact and rigorous analysis and relate the generalization errors (in regression) and classification errors (in binary classification) for the pretrained and fine-tuned models. In particular, we give conditions under which the fine-tuned model outperforms the pretrained one. An important aspect of our work is that all the results are "universal", in the sense that they depend only on the first and second order statistics of the target distribution. They thus extend well beyond the standard Gaussian assumptions commonly made in the literature. Furthermore, our universality results extend beyond standard SGD training to the test error of a classification task trained using ridge regression.

## 1 Introduction

Deep neural networks have revolutionized the way data processing and statistical inference are conducted. Despite their ground-breaking performance, these models often require a plethora of training samples, which can make the process of data acquisition expensive. Moreover, with the advent of increasingly complex deep networks and especially Large Language Models (LLMs), the process of training from scratch has also become prohibitively expensive. To alleviate the scarcity of prepared data and the high training costs, various strategies have been proposed. One such method is fine-tuning in which a network previously trained on a source dataset/task different from the target dataset/task is leveraged as the initialization point for training on the target dataset/task. In many practical applications, especially for networks containing billions of parameters, only a subset of weights are updated to adapt the model to the new task. A particularly attractive method involves fine-tuning only the last layer of the network. A foundational question would be: How effective is this procedure? Are there fundamental limits to how much one can achieve by utilization of a model pretrained on a different distribution?

---

[*]Equal Contribution

38th Conference on Neural Information Processing Systems (NeurIPS 2024).

In this work, we would like to investigate this problem rigorously through the lens of linear regression and binary classification in the overparametrized regime, where the number of weights/parameters exceeds the number of data points. To do so, we analyze the performance of regressors/classifiers obtained through performing SGD initialized at a weight $\mathbf{w}_0 \in \mathbb{R}^d$ acquired through training on the source domain. It was shown in Gunasekar et al. [2018] and Azizan and Hassibi [2018] that gradient descent (GD) and stochastic gradient descent(SGD) iterations converge to the optimal solution $\mathbf{w} \in \mathbb{R}^d$ of the following optimization problem:

$$\min_{\mathbf{w}} \|\mathbf{w} - \mathbf{w}_0\|_2^2 \tag{1}$$
$$s.t \quad \mathbf{y} = \mathbf{Xw}$$

where $\mathbf{w}_0 \in \mathbb{R}^d$ is the initialization point for SGD, $\mathbf{X} \in \mathbb{R}^{n \times d}$ is the design/data matrix which is comprised of independent rows reflecting the fact that datapoints $(\mathbf{x}_i, y_i)$ are sampled independently, and $\mathbf{y} \in \mathbb{R}^n$ is the vector of labels. This property of SGD is known as "Implicit Regularization" and will serve as the basis for our analysis as the pretraining step is captured by the vector $\mathbf{w}_0$. Understanding this "interpolating regime" is key to the theoretical analysis of machine learning models as most of the deep neural networks, on account of their highly overparametrized nature, operate in a setting where they are able to attain negligible training error. It is noteworthy that, even for linear models, characterizing the exact performance of fine-tuning approach has been somewhat limited and is inhibited furthermore by the Gaussianity assumption on $\mathbf{X}$. One of our main contributions is to overcome this limitation. In fact, to showcase the ubiquity of our results, we establish a general universality theorem that applies to a large class of data distributions and extends beyond the context of Transfer Learning. To go into more detail, we say that Gaussian universality holds for a data distribution $\mathbb{P}$ and a training algorithm $T$ if the test error obtained by training on data sampled from $\mathbb{P}$ using $T$ is the same as the test error obtained by training on data sampled from the Gaussian distribution $\mathcal{N}(\boldsymbol{\mu}_\mathbb{P}, \boldsymbol{\Sigma}_\mathbb{P})$ using $T$, where $\boldsymbol{\mu}_\mathbb{P}$ and $\boldsymbol{\Sigma}_\mathbb{P}$ stand for the mean and the covariance matrix of $\mathbb{P}$ respectively. In the context of binary classification, we establish universality of the classification error with respect to the distribution of each class. In light of the the latter result, the problem reduces to analyzing the case of the Gaussian design matrices. We allow classes with different non-scalar covariance matrices $\boldsymbol{\Sigma}_1$ and $\boldsymbol{\Sigma}_2$ and build on the results of Akhtiamov et al. [2024] to address the problem in this specific scenario.

## 1.1 Related Works

Transfer Learning has been an active topic of research at least since the 1970's Bozinovski [2020]. It is mainly applied in the situations where obtaining data from the true distribution, which we will refer as the *target distribution*, is costly but there is a cheap way to access data from a *source distribution*, which bears resemblance to the target distribution. The two most popular approaches to transfer learning consist of instance-based transfer learning and fine-tuning.

Instance-based transfer learning incorporates the source dataset, along with the target dataset, and trains the model on this amalgamated dataset. The key insight is that, provided that the source distribution is close enough to the target distibution and a suitable training scheme is chosen, the final performance enjoys an improvement. We refer the reader to the landmark workDai et al. [2007] as well as to the comprehensive overviews covering the empirical advances in this area Tan et al. [2018], Zhuang et al. [2020]. For theoretical analysis, we would like to emphasize a recent work Jain et al. [2024] that characterized the scaling laws for ridge regression over the union of the source and target data in the high dimensional regime.

The present manuscript focuses on fine-tuning in which a model is first pretrained on the source distribution and then fine-tuned using the target data. Dar et al. [2021] analyzes transfer learning for linear regression assuming Gaussianity of the data. The results obtained in this paper, in particular, imply that the generalization error obtained in their work is universal, in the sense that they can be immediately extended to a much broader set of target distributions, which we elaborate on in much greater detail in the main body of the paper.

Gerace et al. [2022] consider the problem of binary classification via a two-layer neural network in a synthetic setting. The source data and the source labels are sampled according to $\mathbf{x} = Relu(\mathbf{Ga})$ and $y = Sign(\mathbf{a}^T \mathbf{z})$ respectively where $\mathbf{G}, \mathbf{a}$ and $\mathbf{z}$ are i.i.d. Gaussian. The target data and labels are generated by perturbing $\mathbf{G}$ and $\mathbf{a}$ and then applying the same rules as for the source data. Gerace et al. [2022] trained the network on the source dataset and then proceeded by only fine-tuning the

last layer on the target domain using the cross-entropy loss with an $\ell_2$ regularizer. The analysis was conducted using the Replica method. The authors of Gerace et al. [2022] compared their results with an equivalent random feature model and observed empirically a certain kind of Gaussian universality for real-world datasets, such as MNIST.

Extension of results obtained under Gaussianity assumptions to non-Gaussian distributions remains a pertinent research direction in which the idea of Gaussian universality plays a central role. Montanari and Nguyen [2017] proved the universality of the generalization error for the elastic net, assuming the design matrix $\mathbf{A}$ is i.i.d subgaussian. Panahi and Hassibi [2017] generalized the result to the problem of regularized regression with a quadratic loss and a convex separable regularizer which is either $f(\cdot) = \| \cdot \|_1$ (LASSO) or strongly convex. Han and Shen [2023] generalized Panahi and Hassibi [2017]'s results to non-separable Lipschitz test functions and provided non-asymptotic bounds for the concentration of solutions. In the random geometry literature, Oymak and Tropp [2018] showed universality of the embedding dimension of randomized dimension reduction linear maps with i.i.d entries satisfying certain moment conditions. Abbasi et al. [2019] proved universality of the recovery threshold for minimizing strongly convex functions under linear measurements constraint, assuming the rows are iid and the norm of the mean is asymptotically negligible compared to the noise. Lahiry and Sur [2023], proved the universality of generalization error for ridge regression and LASSO when the rows are distributed iid with a specific block structure dependence per row, $\mathbf{AD}$ where $\mathbf{A}$ has zero mean iid random subgaussian vectors per row and $\mathbf{D}$ being diagonal.

Leveraging universality is not limited to the signal recovery literature. As an example, Hu and Lu [2022], Bosch et al. [2023], Schröder et al. [2023], have analyzed the performance of the random feature models by replacing nonlinear functions of Gaussians with the corresponding Gaussian distribution having the same mean and covariance in single-layer and multiple-layer scenarios, respectively, through a universality argument, referred to as the Gaussian equivalence principle. In a more general setting, under subgaussianity assumption on the data, Montanari and Saeed [2022] proved the universality of the training error for general loss and regularizer and test error when the regularizer is strongly convex and the loss is convex. Hastie et al. [2022] proved universality for the the minimal $\ell_2$-norm linear interpolators for the data generated via a very specific rule $\mathbf{x} = \sigma(\mathbf{W}\boldsymbol{\Sigma}^{\frac{1}{2}}\mathbf{z})$, where $\mathbf{z}$ is i.i.d zero mean and satisfies several other technical properties and $\boldsymbol{\Sigma}$ is a deterministic PSD matrix. Dandi et al. [2024] considered the universality of mixture distributions in a similar setting to Montanari and Saeed [2022] and proved the universality of the free energy of the test error.

## 1.2 Outline of the Paper

In Section 2.1, we formally define the optimization problem for which we aim to establish universality. Section 2.2 outlines our primary contributions. In Section 3, we present our main universality result, accompanied by several insightful remarks. Section 4 focuses on our findings related to fine-tuning in the contexts of linear regression and binary classification. In Section 5, we validate our theoretical results through empirical experiments. Finally, we conclude with a summary and discussion in Section 6.

## 2 Problem Formulation

### 2.1 The Setting

We use bold letters for vectors and matrices. We denote the $i$'th largest singular value of matrix $\mathbf{A}$ by $s_i(\mathbf{A})$. We consider the proportional regime where $d = \Theta(n)$ and $\frac{d}{n} \to \kappa > 1$. We refer to Section A.1 for the other notations and definitions necessary for the rest of this exposition.

Given a training dataset $\{(\mathbf{x}_i, y_i)\}_{i=1}^n$ and a model with a fixed architecture, the conventional approach of learning the weights for the model consists of choosing a loss function $\mathcal{L}$ and finding $w$ minimizing $\frac{1}{n}\sum_{i=1}^n \mathcal{L}(\mathbf{x}_i, \mathbf{w}_i, y_i)$. In this exposition we focus on linear models and loss functions of the form $\mathcal{L}(\mathbf{x}_i, \mathbf{w}_i, y_i) = \ell(y_i - \mathbf{x}_i^T\mathbf{w}_i)$, where $\ell$ is differentiable and $\ell(0) = 0$. Azizan and Hassibi [2018] characterized behavior of a broad family of optimizers, called *stochastic mirror descent* (SMD) algorithms, for linear models in the over-parametrized regime. For a strictly convex function $g : \mathbb{R}^d \to \mathbb{R}$, the update rule of the SMD with a *mirror g* and a learning rate $\eta > 0$ is defined as

$$\nabla g(\mathbf{w}_t) = \nabla g(\mathbf{w}_{t-1}) - \eta \nabla \mathcal{L}_t(\mathbf{w}_{t-1}), \quad t \geq 1, \tag{2}$$

where $\eta > 0$ is the learning rate and $\mathcal{L}_t$ is the loss function $\mathcal{L}$ evaluated at a point chosen at random in the dataset corresponding to the $t$'th iteration. Due to strict convexity, $\nabla g(\cdot)$ defines an invertible transformation, which is why (2) is indeed a well-defined update rule. Note also that this includes SGD as a special case, which corresponds to taking $g(\cdot) = \| \cdot \|_2^2$. Azizan and Hassibi [2018] show that applying SMD initialized at $\mathbf{w}_0$ to minimize $\frac{1}{n} \sum_{i=1}^n \ell(y_i - \mathbf{x}_i^T \mathbf{w})$ yields a weight vector defined by the following optimization problem:

$$\min_{\mathbf{w}} D_g(\mathbf{w}, \mathbf{w}_0)$$
$$s.t \quad y_i = \mathbf{x}_i^T \mathbf{w}, \quad 1 \le i \le n$$

We construct the random matrix $\mathbf{A} \in \mathbb{R}^{n \times d}$ by setting its rows equal to $\mathbf{x}_i$. Hence, stating in a more general fashion, we are interested in the analysis of the following optimization problem, with the main case of interest being when $f$ is a quadratic:

$$\min_{\mathbf{w}} f(\mathbf{w})$$
$$s.t \quad \mathbf{y} = \mathbf{A}\mathbf{w}$$

As the objective is comprised of minimizing a strongly convex function $f$ over a closed convex set, it has a unique minimizer. By using a Lagrange multiplier $\lambda \in \mathbb{R}$, this convex optimization problem can be written in the unconstrained form:

$$\Phi(\mathbf{A}) := \sup_{\lambda > 0} \min_w \frac{\lambda}{2} \|\mathbf{A}\mathbf{w} - \mathbf{y}\|_2^2 + f(\mathbf{w}) \tag{3}$$

Considering the objective without the $\sup_{\lambda > 0}$ yields the following regularized linear regression problem for which we will also establish a universality theorem:

$$\Phi_\lambda(\mathbf{A}) := \min_w \frac{\lambda}{2} \|\mathbf{A}\mathbf{w} - \mathbf{y}\|_2^2 + f(\mathbf{w}) \tag{4}$$

Note that (4) captures the case of explicit regularization, which is of highest importance whenever there is a high amount of noise or label corruption present. Furthermore, by deliberately choosing the regularizer $f$, it is possible to obtain solutions that exhibit certain behavior, viz being sparse or compressed. In order to tackle optimizations such as 3, 4, we prove their equivalence to a problem with a suitable Gaussian design $\mathbf{G}$ which is described in Definition 4 in Section A.1. Our approach for proving universality will be the Lindeberg approach Lindeberg [1922] Chatterjee [2006].

## 2.2 Our Contributions

### 2.2.1 Transfer Learning

**Linear Regression** We extend the results from Dar et al. [2021] by providing precise expressions for the generalization error in linear regression tasks. In addition, we demonstrate that the test error is always lower-bounded by a quantity that is attained only when the covariance matrix of the data is scalar. Furthermore, we identify specific conditions under which fine-tuning is successful.

**Binary Classification** We present the first precise characterization of the classification error for linear models trained using stochastic gradient descent (SGD) on data drawn from a general mixture distribution with arbitrary covariance matrices. Moreover, we delineate the regimes in which fine-tuning proves effective.

### 2.2.2 Universality

**Proof of universality for implicit regularization.** In the context of SGD and, more generally, SMD (see 2) with a mirror $f$ satisfying the assumptions of Theorem 1, we prove universality of the generalization error as well as of the value of the corresponding implicit regularization objective for a wide range of data distributions characterized by Assumptions 1. Note that this cannot be reduced to any known results on universality of constrained objectives as the latter assume that the constraints are deterministic, i.e the optimization variables belong to some deterministic set $\mathcal{C}$ chatacterized by the constraints. While to analyze SMD one has to deal with constraints of the form $\mathbf{A}\mathbf{w} = \mathbf{y}$ which represents a random polytope. To the best of our knowledge, the only other paper dealing with universality in the context of implicit regularization is Hastie et al. [2022]. They study minimal

$\ell_2$-norm linear interpolators, but they work with a very specific (random feature) model for data distributions defined by $\mathbf{x} = \sigma(\mathbf{W}\boldsymbol{\Sigma}^{\frac{1}{2}}\mathbf{z})$, where $\mathbf{z}$ is i.i.d zero mean and $\mathbf{W}$ is i.i.d. Gaussian. Our paper generalizes theirs in two non-trivial ways, as it allows for arbitrary smooth convex strongly convex mirrors $f(\mathbf{w})$ as well as more general data distributions.

**Relaxing assumptions for explicit regularization.** To the best of the authors' knowledge, all previous results on universality assumed either that the data points $\mathbf{x} \in \mathbb{R}^d$ have i.i.d. entries or that the rows are independent and $\mathbf{x}$ is subgaussian. We relax these assumptions. That is, we show that, for the quadratic loss function and any strongly convex not necessarily differentiable regularizer universality holds as long as the rows are i.i.d., all moments of the $\mathbf{x}$ up to the 6th satisfy $\mathbb{E}_{\mathbf{x}}|(\mathbf{x} - \mathbb{E}\mathbf{x})^T\mathbf{v}|^q \leq K\frac{\|\mathbf{v}\|_2^q}{d^{q/2}}$ for $\mathbf{v} \in \mathbb{R}^d$ and $Var(\mathbf{x}^T\mathbf{C}\mathbf{x}) \to 0$ for any $\mathbf{C}$ of bounded operator norm.

**Extending universality to mixtures of distributions.** Since the main motivation behind this work is the study of generalization in classification tasks, we focus on data matrices sampled from mixture distributions. Note that most previous papers on universality, such as Montanari and Saeed [2022], do not apply to this setting directly. As an illustration to why results of Montanari and Saeed [2022] cannot just be applied to the mixture distributions directly, consider $\mathbf{P} = \frac{1}{2}\mathcal{N}(\boldsymbol{\mu}, \mathbf{I}) + \frac{1}{2}\mathcal{N}(-\boldsymbol{\mu}, \mathbf{I})$ corresponding to a mixture of two classes with antipodal means and the resulting classification error depends on $\|\boldsymbol{\mu}\|$, while $\mathbf{P}$ has mean $\mathbf{0}$ and covariance $\mathbf{I}$ meaning that the matching gaussian distribution $\mathcal{N}(0, \mathbf{I})$ does not contain any information about the classes.

However, there is one prior paper Dandi et al. [2024] that studies universality specifically in the context of mixture distributions. It is worth mentioning that their definition of universality is different. Namely, they prove universality of the expectation with respect to the Gibbs distribution, which suffices to show universality of the train but not of the test error.

**Allowing for non-vanishing means.** In Definition 2, we assume only that $\mathbb{E}_{\mathbf{a}}\|\mathbf{a} - \boldsymbol{\mu}\|_2^2 = O(1)$, whereas in Abbasi et al. [2019], Montanari and Saeed [2022] and Dandi et al. [2024] they have $\frac{\|\boldsymbol{\mu}\|_2^2}{\mathbb{E}_{\mathbf{a}}\|\mathbf{a}-\boldsymbol{\mu}\|_2^2} \to 0$, implying that the norms of the means are negligible compared to the norms of the data points on average. To our knowledge, our work is the first to tackle it.

## 3 Main Result

In this section, we present our main universality theorem and before doing so, we list the required technical assumptions. Consider the following convex optimization problem

$$\Phi_\lambda(\mathbf{A}) := \min_w \frac{\lambda}{2n}\|\mathbf{A}\mathbf{w} - \mathbf{y}\|_2^2 + \frac{1}{n}f(\mathbf{w}) \tag{5}$$

Denote the solution to the optimization problem 5 above by $\mathbf{w}_{\Phi_\lambda(\mathbf{A})}$. Then we assume the following:

**Assumptions 1.**      *1. $\mathbf{A}$ is a regular block matrix and $\mathbf{G}$ is its matching Gaussian matrix as in Section A.1.*

2. *The regularizer $f(\cdot)$ is regular also as Section A.1 or there exists a sequence of regular $f_m$'s converging uniformly to $f$.*

3. *$f(\cdot)$ is $M$-strongly convex.*

4. *$\mathbb{E}_{\mathbf{y}}\|\mathbf{y}\|_2^2 = O(d)$ with $\mathbb{E}|y_j|^q = O(1)$ for $j \in [n]$ and $q \in [6]$ and is independent of $\mathbf{A}$.*

5. *$s_{\min}(\mathbf{A}\mathbf{A}^T) = \Omega(1)$ with high probability,*

6. *$Var(\|\mathbf{y}\|_2^2) \to 0$*

7. *$\|\nabla f(\mathbf{w}_{\Phi_\lambda(\mathbf{A})})\|_2 \leq K_f\sqrt{n}$ for a constant $K_f$ independent of $\lambda$.*

**Theorem 1.** *If $\Psi = \Phi = \sup_{\lambda>0}\Phi_\lambda$, suppose parts (1) - (7) of Assumptions 1 hold and for $\Psi = \Phi_\lambda$, only assume parts (1) - (4) of Assumptions 1. Then*

1. *(**Universality of the training error**) For any $L$-Lipschitz $g : \mathbb{R}^d \to \mathbb{R}$, and every $t_1 > 0, t_2 > t_1, c \in \mathbb{R}$ the following holds:*

   - *If $\mathbb{P}(|g(\Psi(\mathbf{G})) - c| > t_1) \to 0$ then $\mathbb{P}(|g(\Psi(\mathbf{A})) - c| > t_2) \to 0$.*

- *Furthermore, if g has bounded second derivative, then*

$$\lim_{n \to \infty} \left| \mathbb{E}_{\mathbf{A}, \mathbf{y}} g(\Psi(\mathbf{A})) - \mathbb{E}_{\mathbf{G}, \mathbf{y}} g(\Psi(\mathbf{G})) \right| = 0$$

2. **(Universality of the solution)** *For a function $\psi$ with $\nabla^2 \psi \leq q\mathbf{I}$ if either $\psi$ is regular (Definition 1) or there is a sequence of regular functions converging uniformly to $\psi$,*

   - *If for every $t > 0$, $\mathbb{P}(|\Psi(\mathbf{G}) - c| > t) \to 0$ then for $\mathbf{B} \in \{\mathbf{A}, \mathbf{G}\}$, there exists $c_1 \in \mathbb{R}$ such that for any $t_1 > 0$ we have $\mathbb{P}(|\psi(\mathbf{w}_{\Psi(\mathbf{B})}) - c_1| > t_1) \to 0$.*
   - *Furthermore, if $\psi$ is bounded*

$$\lim_{n \to \infty} \left| \mathbb{E}_{\mathbf{A}, \mathbf{y}} \psi(\mathbf{w}_{\Psi(\mathbf{A})}) - \mathbb{E}_{\mathbf{G}, \mathbf{y}} \psi(\mathbf{w}_{\Psi(\mathbf{G})}) \right| = 0$$

To sum up, Theorem 1 says that the vector of weights $\mathbf{w_A}$ trained on data sampled from a non-gaussian distribution $\mathbf{A}$ either via SMD with a mirror $f$ or by minimizing the least squares objective with regularizer $f$, shares many similar characteristics with the vector of weights $\mathbf{w_G}$ trained on data sampled from the matching GMM $\mathbf{G}$ via the same optimization procedure under certain technical assumptions on $\mathbf{A}$ and $f$. By "similar characteristics", we mean that for any $\psi : \mathbb{R}^d \to \mathbb{R}$ with bounded Hessian, $\psi(\mathbf{w_A}) = \psi(\mathbf{w_G})$ holds in the limit.

We would like to point out a few remarks.

**Remark 1.** *Note that $\psi$ need not be convex, in fact any linear combination of regular $\psi$ is also regular. An immediate result is the universality of the empirical distribution of the coordinates of solutions.*

**Remark 2.** *The assumption $\|\nabla f(\mathbf{w}_{\Phi_\lambda(\mathbf{A})})\|_2 = O(\sqrt{n})$ in Theorem 1 is satisfied for any function $f$ with a locally lipschitz gradient, i.e $\|\nabla f(\mathbf{u})\|_2 \leq K(1 + \|\mathbf{u}\|_2)$. In particular it applies to $f(\cdot) = \|\cdot\|_2^2$.*

**Remark 3.** *In both parts of Theorem 1, the regularizer is allowed to be an M-strongly convex uniform limit of differentiable convex functions, which implies that $f(\mathbf{w}) = t\|\mathbf{w}\|_1 + M\|\mathbf{w}\|_2^2, t > 0, M > 0$, known as the elastic net, is also included in our results.*

**Remark 4.** *In Theorem 1, the optimal value of $\Phi_\lambda$ is only achieved when $\lambda \to \infty$ and is not attained at any finite $\lambda$. This means that proving the results for $\Psi = \Phi$ in Theorem 1 requires additional ideas apart from the case $\Psi = \Phi_\lambda$ in Theorem 1, as the latter makes use of the boundedness of $\lambda$ extensively. To tackle this issue, we present a uniform convergence result in $\lambda$ which might be of independent interest.*

**Remark 5.** *The condition $s_{\min}(\mathbf{A}\mathbf{A}^T) = \Omega(1)$ can be satisfied for a variety of random matrix models. If for each block of $\mathbf{A}$, $\mathbf{A}_i = \tilde{\mathbf{A}}_i \Sigma_i^{1/2}$ where $\tilde{\mathbf{A}}_i$ has iid entries and $s_{\min}(\Sigma_i) = \Omega(1)$, then by the Bai-Yin's law Bai and Yin [2008] the condition is satisfied. The second family is comprised of blocks with independent and identical rows where the norm of rows have exponential concentration. We prove this in the Appendix. One particular instance will be the distributions satsifying LCP from Definition 3.*

**Remark 6.** *The second assumption in Assumptions 1 is not too restrictive and is in particular satisfied for $\psi(\mathbf{w}) = \mathbb{P}(\mathbf{a}^T \mathbf{w} > c)$. We have with high probability for $\Psi \in \{\Phi_\lambda, \Phi\}$*

$$\lim_{n \to \infty} \left| \mathbb{P}(\mathbf{a}^T \mathbf{w}_{\Psi(\mathbf{A})} > c) - \mathbb{P}(\mathbf{a}^T \mathbf{w}_{\Psi(\mathbf{G})} > c) \right| = 0$$

*For $\mathbf{a}$ independent of $\mathbf{A}, \mathbf{G}$ but sampled from the same distribution as the rows of $\mathbf{A}$, respectively. Note that by this argument we have reduced the problem of verifying CLT for $\mathbf{a}^T \mathbf{w}_{\Psi(\mathbf{A})}$ to its Gaussian counterpart, $\mathbf{a}^T \mathbf{w}_{\Psi(\mathbf{G})}$. Then, specifically for the applications presented in this paper, we provide a proof that $\mathbf{a}^T \mathbf{w}_{\Psi(\mathbf{G})}$ satisfies CLT w.r.t. to randomness in $\mathbf{a}$ with high probability in $\mathbf{G}$. However, in general the latter CLT condition has to be verified on a case by case basis.*

## 4 Applications: Transfer Learning

### 4.1 Regression

We consider the classical problem of recovering the best linear regressor for the following linear model

$$\mathbf{y} = \mathbf{X}\mathbf{w}_* + \mathbf{z} \tag{6}$$

Here, $\mathbf{w}_*$ is the ground truth, the rows of the data matrix $\mathbf{X}$ are i.i.d with $\mathbb{E}\mathbf{x}_i = \mathbf{0}$ and $\mathbb{E}\mathbf{x}_i\mathbf{x}_i^T =: \mathbf{R}_x$, $\mathbf{z}$ is centered and is independent of $\mathbf{X}$ and satisfies $\mathbb{E}_{\mathbf{z}}\|\mathbf{z}\|_2^2 = \sigma^2 n$. For a given $\hat{\mathbf{w}}$, its generalization error is defined by:

$$e_{gen}(\hat{\mathbf{w}}) := \mathbb{E}_x(\mathbf{x}^T\mathbf{w}_* - \mathbf{x}^T\hat{\mathbf{w}})^2 = (\hat{\mathbf{w}} - \mathbf{w}_*)^T\mathbf{R}_x(\hat{\mathbf{w}} - \mathbf{w}_*)$$

Now in order to recover $\mathbf{w}_*$ given observations $(\mathbf{y}, \mathbf{X})$, we choose to optimize the least squares objective, $\min_{\mathbf{w}}\|\mathbf{y} - \mathbf{X}\mathbf{w}\|_2^2$, and to do so, we leverage SGD. We would like to investigate how useful having a pretrained classifier $\mathbf{w}_0$ can be for the recovery of $\mathbf{w}_*$. As pointed out earlier, SGD initialized from $\mathbf{w} = \mathbf{w}_0$, by its implicit regularization property, converges to the solution $\hat{\mathbf{w}}$ of (1). We denote $e_a := e_{gen}(\mathbf{w}_0)$; hence the name "a priori error", $e_a$. We provide the assumptions necessary for our results on linear regression in Section A.2. In what follows we give a precise characterization of the posterior error, $e_p$, of the model after training with SGD, in terms of $e_a$ and the other parameters of the problem. In order to leverage Theorem 1 to establish the universality of the generalization error, it is sufficient to apply a change of variable $\mathbf{w}' := \mathbf{R}_x^{-1/2}\mathbf{w}$ and use $\psi(\mathbf{w}) = \|\mathbf{w}\|_2^2$ as the test function.

**Theorem 2.** *Under Assumptions 3 in Section A.2, the generalization error of the SGD solution initialized from $\mathbf{w}_0$ converges in probability to*

$$e_p = \frac{2 - \kappa(1 - t)}{\kappa(1 - t) - 1}\sigma^2 + \frac{t}{\kappa(1 - t) - 1}e_a \tag{7}$$

*With $t = \int \frac{4p(r)}{(2 + r\theta)^2}dr$ where $\theta$ is found through $\frac{\kappa - 1}{\kappa} = \frac{1}{\theta}S_p(-\frac{1}{\theta})$ (12). And $\kappa > 1$ is the proportional constant, i.e for $\mathbf{X} \in \mathbb{R}^{n \times d}$, $\frac{d}{n} \to \kappa$. Moreover, for any distribution $p(r)$ we have*

$$e_p \geq \frac{1}{\kappa - 1}\sigma^2 + \frac{\kappa - 1}{\kappa}e_a \tag{8}$$

*The lower bound is attained if and only if $p(r) = \delta(r - r_0)$ for some $r_0 > 0$*

The following remark is immediate:

**Remark 7.** *Theorem 2 entails that, depending on the noise level present in the data, training could be even potentially harmful. Indeed, if $\sigma^2 \geq e_a$, then $e_p \geq \frac{1}{\kappa - 1}\sigma^2 + \frac{\kappa - 1}{\kappa}e_a \geq \sigma(2\sqrt{e_a} - \sigma) \geq e_a$ for any $\kappa > 1$ and it is therefore more appropriate to use vector $\mathbf{w}_0$ instead of performing fine-tuning. Moreover, if the covariance $\mathbf{R}_x$ is scalar, then the converse is true. Namely, if $e_a \geq \sigma^2$, then the best achievable error corresponds to $\kappa_* = \frac{\sqrt{e_a}}{\sqrt{e_a} - \sigma}$ and equals $\sigma(2\sqrt{e_a} - \sigma)$, which is less than $e_a$. Therefore, transfer learning contributes to improving the test performance when the variance of the noise is not too high, but the model has to be fine-tuned on the correct amount of target data.*

### 4.2 Classification

#### 4.2.1 Problem setting

Consider a binary classification task and let $\mathbf{X}$ stand for the data matrix and $\mathbf{y}$ denote the vector of labels where each $y_i = \pm 1$ depending on what class the $i$-th point $\mathbf{x}_i$ falls into. We denote $\boldsymbol{\mu}_\ell := \mathbb{E}\mathbf{x}_i$ and $\boldsymbol{\Sigma}_\ell := \mathbb{E}\mathbf{x}_i\mathbf{x}_i^T - \boldsymbol{\mu}_\ell\boldsymbol{\mu}_\ell^T$, for the mean and covariance of each datapoint $\mathbf{x}_i$, respectively and $\ell \in \{1, 2\}$ depending on the class of $\mathbf{x}_i$. After learning a linear classifier $\mathbf{w}$, we assign labels to new previously unseen points according to $\mathbf{y}_{new} = sign(\mathbf{w}^T\mathbf{x}_{new})$. Without loss of generality we will assume that the first $\frac{n}{2}$ rows of $\mathbf{X}$ are sampled from the first class and the remaining rows are sampled from the second as we can permute the rows otherwise. Modulo such permutation, it is straightforward to see that $\mathbf{X}$ satisfies parts (1) - (3) of Definition 2. Since the topic of the present paper is fine-tuning, we assume that the following steps are performed:

1. Obtain a pre-trained classifier $\mathbf{w}_0$
2. Renormalize $\mathbf{w}_0$ obtained during the previous step using the target data via

$$\min_\alpha \|\mathbf{y} - \alpha\mathbf{X}\mathbf{w}_0\|^2$$

This yields:

$$\alpha = \frac{\mathbf{y}^T\mathbf{X}\mathbf{w}}{\|\mathbf{X}\mathbf{w}\|^2} \tag{9}$$

After finding $\alpha$ take $\mathbf{w}_0' := \alpha \mathbf{w}_0$. This transform preserves the direction of $\mathbf{w}_0$, while setting its squared magnitude to $\alpha^2 \|\mathbf{w}_0\|^2 = \frac{(\mathbf{y}^T \mathbf{X} \mathbf{w}_0)^2 \|\mathbf{w}_0\|^2}{\|\mathbf{X} \mathbf{w}_0\|^4}$, which depends only on the direction of $\mathbf{w}_0$ but not on its magnitude anymore. We find applying this transform very meaningful, as it does not change the classification error for the source data but simplifies learning for the regression problem defined by the target data.

3. Learn the final classifier from the target data $\mathbf{X} \in \mathbb{R}^{n \times d}$ and labels $\mathbf{y} \in \mathbb{R}^n$ using SGD initialized at $\mathbf{w}_0'$, obtained from the previous step.

We will use the assumptions in Section A.3 to define further details of the classification task we consider. Note that in practice the main difficulty of the classification task in the over-parametrized regime ($n < d$) arises due to the fact that $\boldsymbol{\mu}_1, \boldsymbol{\mu}_2, \boldsymbol{\Sigma}_1$ and $\boldsymbol{\Sigma}_2$ are not known and cannot be estimated reliably. Nevertheless, we would like to start with characterizing the optimal performance in the scenario where these are provided to us by an oracle. Lemma 1 in Section A.3 provides such a characterization under certain symmetry assumptions. In view of Lemma 1 it is natural to introduce the following assumption:

**Assumptions 2.** *The initialization point $\mathbf{w}_0$ satisfies*

$$\mathbf{w}_0 = t_* \mathbf{w}_* + t_\eta \boldsymbol{\eta} \tag{10}$$

*where $\mathbf{w}_* = (\boldsymbol{\Sigma}_1 + \boldsymbol{\Sigma}_2)^{-1}(\boldsymbol{\mu}_1 - \boldsymbol{\mu}_2)$ is the optimal classifier defined as in Lemma 1. $\boldsymbol{\eta}$ is a noise vector with $\|\boldsymbol{\eta}\|^2 = 1 - r$ and for any deterministic matrix $\mathbf{C}$ of bounded operator norm it holds that $\boldsymbol{\eta}^T \mathbf{C} \boldsymbol{\eta}$ converges in probability to $\frac{Tr(\mathbf{C})(1-r)}{d}$. Note that the smaller the ratio $\frac{t_\eta}{t_*}$ is, the better performance $\mathbf{w}_0$ has.*

Indeed, 10 captures the closeness of the initilization point $\mathbf{w}_0$ to the optimal classifier $\mathbf{w}_*$. For example, in the isotropic case $\boldsymbol{\Sigma}_1 = \boldsymbol{\Sigma}_2 = \sigma^2 \mathbf{I}$, if $\mathbf{w}_0$ has the classification error $e_a$, this means that we should take $\frac{t_\eta}{t_*} = \frac{d}{2\sigma^2} \sqrt{\frac{d(1-r)}{\sigma^2 Q^{-1}(e_a)^2} - 2}$. Finally, we would like to remark that, under notation and assumptions from Theorem 1, the following corollary is implied by Theorem 1 modulo Theorem 4 presented in the Appendix along with the explanation of the implication:

**Corollary 1** (Universality of the classification error for SGD and ridge regression objectives)**.** *If $f(\mathbf{w}) = \|\mathbf{w} - \mathbf{w}_0\|_2^2$, when $\|\mathbf{w}_0\|_2^2 = O(d)$ then for $\mathbf{a}, \mathbf{g}$ independent of $\mathbf{A}, \mathbf{G}$ but sampled from the same distribution as their rows, respectively, we have with high probability for $\Psi \in \{\Phi_\lambda, \Phi\}$*

$$\lim_{n \to \infty} \left| \mathbb{P}(\mathbf{a}^T \mathbf{w}_{\Psi(\mathbf{A})} > 0) - \mathbb{P}(\mathbf{g}^T \mathbf{w}_{\Psi(\mathbf{G})} > 0) \right| = 0$$

### 4.2.2 Analysis of the Classification Error

**Theorem 3.** *Let $\mathbf{w}_0$ be an initialization point that satisfies Assumption 2 and $\mathbf{X}$ be a data matrix satisfying Assumptions 4 in Section A.3. Then the classification error of the SGD solution initialized at $\alpha \mathbf{w}_0$ for $\alpha$ defined by (9) and trained on $\mathbf{X}, \mathbf{y}$ is given by $e_P = Q\left( \frac{-\gamma + 2}{2\sqrt{\tau^2 - \frac{\gamma^2}{4}}} \right)$ where $\gamma$ and $\tau$ are determined in terms of a $\theta$ solving $\frac{2\sqrt{2}}{\theta \sqrt{n}} S_{\Sigma_1 + \Sigma_2}(-\frac{2\sqrt{2}}{\theta \sqrt{n}}) = 1 - \frac{1}{\kappa}$ whose expressions are provided in the Appendix (cf. Section I, equation 40) with $\frac{d}{n} \to \kappa > 1$.*

The expressions from Theorem 3 can be simplified drastically in the case of scalar covariance matrices.

**Corollary 2.** *Under the notation from Theorem 3, assume $\Sigma_1 = \Sigma_2 = \frac{\sigma^2 \mathbf{I}_d}{d}$ and define $\rho := \frac{d(1-r)}{\sigma^2}$. Let $e_a$ be the classification error of $\mathbf{w}_0$ initialized according to 10. Then*

$$e_P(\kappa, \rho) \xrightarrow{\mathbb{P}} Q\left( \frac{\left(Q^{-1}(e_a)^2(2\kappa + \rho - 2) + \rho\right)\sqrt{\kappa(\kappa - 1)}}{\sqrt{(2\kappa + \rho)((4\kappa - 2)Q^{-1}(e_a)^4 + Q^{-1}(e_a)^2\left(2\kappa^3 + \kappa^2\rho - 2\kappa(\rho - 1) + \rho\right) + 2\kappa^2)}} \right) \tag{11}$$

We arrive at the following conclusion summarizing the derivations above.

**Remark 8.**    • *If $\rho = \omega(1)$, then $e_p(\kappa, \infty) \xrightarrow{\mathbb{P}} Q\left(\left(Q^{-1}(e_a) + \frac{1}{Q^{-1}(e_a)}\right)\sqrt{\frac{\kappa}{\kappa-1}}\right) < e_a$ and fine-tuning always succeeds. It is observed that for $e_a > Q(1)$, the worse $e_a$ is, the better $e_p$ will be.*

• *If $\rho = \Theta(1)$, then the fine-tuning step may or may not succeed depending on whether $e_p(\kappa, \rho) < e_a$ or not. See Section 5.2 for further (empirical) explorations on the usefulness of the fine-tuning of the pre-trained solution for this regime.*

• *If $\rho = o(1)$, then the classification error of any linear classifier goes to $\frac{1}{2}$ as $n \to \infty$ since it is lower bounded by $Q(\sqrt{\frac{\rho}{2}})$ according to Lemma 1. Thus, any kind of learning will fail in this regime.*

An interesting regime is where the number of samples is much lower than the number of parameters, which naturally rises in the context of fine-tuning large models and corresponds to letting $\kappa \gg 1$. For this regime, we have

$$e_p \approx Q\left(Q^{-1}(e_a) + \frac{1}{2}\left(\frac{\rho - 1}{Q^{-1}(e_a)} - 3Q^{-1}(e_a)\right)\frac{1}{\kappa}\right)$$

We observe that if $\rho < 1$, transfer learning always fails independent of value of $e_a$ for $\kappa \gg 1$ and training would not help with improving performance. On the other hand, if $e_a > Q(\sqrt{\frac{\rho-1}{3}}) > Q(\sqrt{\frac{\rho}{2}})$, for large enough $\kappa$, transfer learning always achieves an error less than $e_a$.

## 5 Numerical experiments

### 5.1 Regression

To corroborate our findings, we plotted the generalization error of the weight obtained through running SGD according to the Assumptions 3 in Section 4.1 with respect to $\kappa = \frac{d}{n}$. To do so, we fixed $d = 1000$ and varied $n$ across different values. We used CVXPY (Grant and Boyd [2014], Agrawal et al. [2018]) to solve (1) efficiently on a Laptop CPU. To verify the universality of our results, we initially constructed a centered random matrix $\mathbf{X}'$ with i.i.d components according to the distributions $\mathcal{N}(0,1)$, $Ber(0.5)$, and $\chi^2(1)$ and using a correlation matrix $\mathbf{R}_x$ we defined $\mathbf{X} := \mathbf{X}'\mathbf{R}_x^{1/2}$. On the other hand, we generated $\mathbf{R}_x$ according to the following three distributions: single level $p(r) := \delta(r - 1)$, bilevel $p(r) := 0.3\delta(r - 1) + 0.7\delta(r - 5)$ and uniform on the interval $[1, 5]$. We specifically consider these cases as they are common in the literature and we use the parameter $\sigma^2$ for the component-wise variance of $\mathbf{z}$ in 6. Additionally, $\mathbf{w}_0$ is chosen according to Assumptions 3 in such a manner that $e_a = 1$. In both Figures 1, 2 the blue line represents the prediction 7 made by Theorem 2, the red line depicts the lower bound 8. The markers showcase the performance of weights obtained under different distributions as described earlier. It can be seen that from Figures 1, 2, for the bilevel and uniform distributions, depending on the value of $\sigma$, transfer learning might not be beneficial as discussed in Remark 7. In particular, in Figures 1c and 2c, the generalization error is always lower bounded by $e_a = 1$ and only in $\kappa \to \infty$ can get close to 1. Finally, the single-level distribution on $\mathbf{R}_x$ is always a lower bound for the generalization error of various distributions on $\mathbf{R}_x$.

### 5.2 Classification

Similar to the preceding subsection, we experimented with sampling the entries of $\mathbf{X}$ independently from three different centered distributions: normal, Bernoulli, and $\chi^2$. We also sampled the means $\boldsymbol{\mu}_1$ and $\boldsymbol{\mu}_2$ from $\mathcal{N}(0, \frac{1}{d}\mathbf{I}_d)$ with a cross-correlation $r = \mathbb{E}[\mu_{1i}\mu_{2i}] = 0.9$ and added them to the corresponding rows of $\mathbf{X}$. For Figures 3, we fixed $\rho = 0.8, 2, 5$ respectively and plotted the classification error predicted by Corollary 2 as a solid red line, empirically observed classification errors for the normal, Bernoulli and $\chi^2$ entries as black squares, green circles and red triangles respectively. The blue lines depict the classification error at the initialization. It can be seen that there is a close match between the empirical errors between points from different distributions as well as with the theoretical prediction, thus validating both Theorem 1 and Theorem 3. Note that fixing $\rho$

in this setting corresponds to fixing $\sigma^2$ as $\rho = \frac{d(1-r)}{\sigma^2}$. It is also worth mentioning that fine-tuning improves performance in the setting of Figure 3c but hurts it for Figure 3a. Moreover, note that in Figure 3b, although for smaller $\kappa$ transfer learning hurts, past a certain $\kappa$, training improves the performance. Also we observe that by increasing $\rho$ across the three plots, the classification task becomes easier and fine-tuning improves performance as supported by Remark 8.

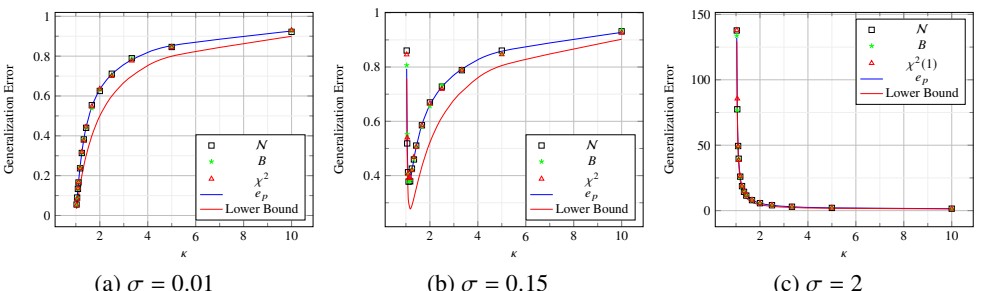

(a) $\sigma = 0.01$           (b) $\sigma = 0.15$           (c) $\sigma = 2$

Figure 1: Generalization error for the bilevel distribution on the covariance of the data

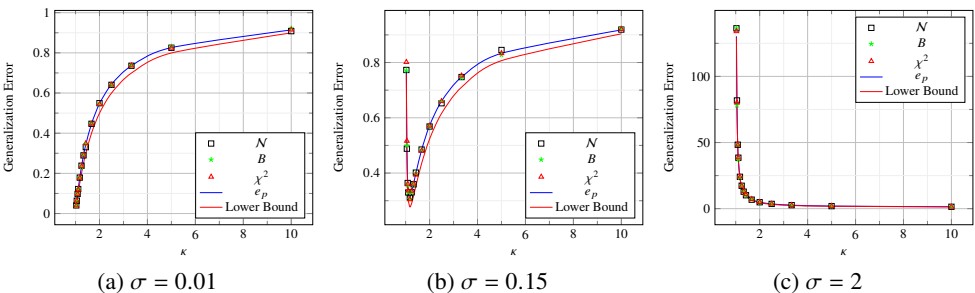

(a) $\sigma = 0.01$           (b) $\sigma = 0.15$           (c) $\sigma = 2$

Figure 2: Generalization error for the uniform distribution on the covariance of the data

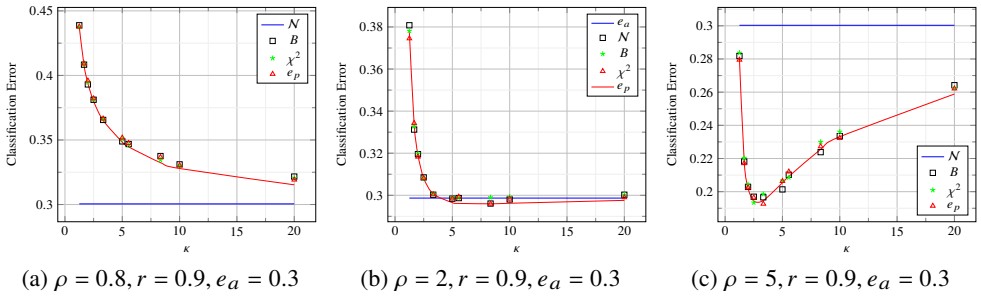

(a) $\rho = 0.8, r = 0.9, e_a = 0.3$     (b) $\rho = 2, r = 0.9, e_a = 0.3$     (c) $\rho = 5, r = 0.9, e_a = 0.3$

Figure 3: Classification error

# 6 Conclusion and future work

We presented a novel Gaussian universality result and used it to study the problem of transfer learning in linear models, for both regression and binary classification. In particular, we were able to precisely relate the performance of the pretrained model to that of the fine-tuned model trained via SGD and, as a result, identified situations where transfer learning helps and where it or does not. Possible future directions include investigating other problems where the universality result may be useful, extending the results to potential functions that are not necessarily convex nor separable, as well as exploring the implications of universality for objectives with explicit regularization.

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

# A  Notations, Definitions and Assumptions

## A.1  Notations and Definitions

We call a convex function $f : \mathbb{R}^d \to \mathbb{R}$ separable if $f(\mathbf{w}) = \sum_{i=1}^{d} f_i(w_i)$, where $f_i : \mathbb{R} \to \mathbb{R}$ are convex. Examples of such functions include $\| \cdot \|_p^p$ for every $p \geq 1$. A function $f : \mathbb{R}^d \to \mathbb{R}$ is $M$-strongly convex for $M > 0$ if for every $\mathbf{w} \in \mathbb{R}^d$, $f(\mathbf{w}) - M\|\mathbf{w}\|_2^2$ is convex.

Functions satisfying the following definition will be instrumental in the statements of our results:

**Definition 1.** *We call a function $f : \mathbb{R}^d \to \mathbb{R}$ regular if it satisfies the following conditions*

1. *$f$ is convex with $f(0) = O(d)$.*

2. *$f$ is three times differentiable.*

3. *$f$ satisfies the following third-order condition for some $C_f > 0$:*

$$\sum_{i,j,k=1}^{d} \frac{\partial^3 f}{\partial w_i \partial w_j \partial w_k} v_i v_j v_k \leq C_f \sum_{i=1}^{d} |v_i|^3$$

Note that separable functions with bounded third order derivatives satisfy the third assumption of Definition 1. For the description of the main results on regression and classification we recall several probability theory concepts.

Given a real-valued measure $\mu$, its Stieltjes transform is defined as

$$S_\mu(z) := \int_{\mathbb{R}} \frac{1}{r - z} \mu(dr), \quad z \in \mathbb{H}_+ \cup \mathbb{R} \setminus Supp(\mu) \tag{12}$$

With $\mathbb{H}_+ = \{z : Im(z) > 0\}$. Moreover, we denote convergence in probability and weak convergence for measures by $\xrightarrow{\mathbb{P}}$ and $\rightsquigarrow$ respectively. For a random variable $X$, we sometimes use $Var(X)$ for the variance of $X$. For a convex differentiable function $g$, the Bregman divergence is defined by

$$D_g(\mathbf{w}, \mathbf{w}_0) := g(\mathbf{w}) - g(\mathbf{w}_0) - \nabla g(\mathbf{w}_0)^T (\mathbf{w} - \mathbf{w}_0)$$

Next, we will provide a description of the design matrices investigated in this paper. It will be clear soon that these assumptions are often satisfied in practice.

**Definition 2.** [2] *We call a random matrix $\mathbf{A} \in \mathbb{R}^{n \times d}$ block-regular if it satisfies the following properties:*

1. *$\mathbf{A}^T = \begin{pmatrix} \mathbf{A}_1^T & \mathbf{A}_2^T & \ldots & \mathbf{A}_k^T \end{pmatrix}$ with $k$ being finite and for each $\mathbf{A}_i \in \mathbb{R}^{n_i \times d}$, $n_i = \Theta(n)$*

2. *For $1 \leq i \leq k$, the rows of $\mathbf{A}_i$ are distributed independently and identically.*

3. *$\mathbb{E}\mathbb{A}_i =: \mathbb{1}\boldsymbol{\mu}_i^T$*

4. *Let $\mathbf{a}$ be any row of $\mathbf{A}$ and $\boldsymbol{\mu}$ be its mean. Then for any deterministic vector $\mathbf{v} \in \mathbb{R}^d$, and $q \in \mathbb{N}$, $q \leq 6$, there exists a constant $K > 0$ such that $\mathbb{E}_{\mathbf{a}} |(\mathbf{a} - \boldsymbol{\mu})^T \mathbf{v}|^q \leq K \frac{\|\mathbf{v}\|_2^q}{d^{q/2}}$*

5. *For any deterministic matrix $\mathbf{C} \in \mathbb{R}^{d \times d}$ of bounded operator norm we have $Var(\mathbf{a}^T \mathbf{C} \mathbf{a}) \to 0$ as $d \to \infty$.*

6. *Each $\|\boldsymbol{\mu}_i\|_2^2 = O(1)$*

Note that assumptions 1-3 from Definition 2 are satisfied for the design matrix both in the classification and regression scenarios, and the assumption $\|\boldsymbol{\mu}_i\|_2^2 = O(1)$ is just a matter of normalization. While assumptions 4 and 5 might appear obscure at first, any $\mathbf{a} \in \mathbb{R}^d$ satisfying the Lipschitz Concentration Property (LCP) also satisfies the aforementioned assumptions. Recall that the LCP is defined as follows.

---

[2]We would like to point out that in this definition and throughout this work in general, mathematically speaking, we are dealing with sequences of vectors and matrices with increasing dimensions. So, for example, assumption 5 can be formally stated as follows: given any sequence of $\mathbf{C}_i \in \mathbb{R}^{d_i \times d_i}$ satisfying $\|\mathbf{C}_i\|_{op} \leq K$ for some constant $K > 0$ and all $i > 0$, we have that $Var(\mathbf{a}_i \mathbf{C}_i \mathbf{a}_i) \to 0$ as $i \to \infty$

**Definition 3.** *We say that a distribution $\mathbf{a} \in \mathbb{R}^d$ satisfies LCP if the following inequality holds for any Lipschitz function $\phi : \mathbb{R}^d \to \mathbb{R}$*

$$\mathbb{P}(|\phi(\mathbf{a}) - \mathbb{E}\phi(\mathbf{a})| > t) \leq 2 \exp\left(-\frac{dt^2}{2\|\phi\|_{Lip}^2}\right)$$

.

It is also worth mentioning that, as shown in Seddik et al. [2020], any data distribution from which one can sample using a GAN satisfies Definition 3. Since it is known that many natural datasets can be approximated well by GAN-generated data in practice, we thus find this assumption realistic. These assumptions also bear significance for the instance-based approach to transfer learning as utilizing GANs can remedy the dearth of data. What is more, these assumptions are not limited to the subgaussians and include many other subexponential distributions, including $\chi^2$ which appears naturally in many signal processing applications such as Phase Retrieval. In order to tackle optimizations such as 3, 4, we prove their equivalence to a problem with a suitable Gaussian design $\mathbf{G}$. For this purpose we will need the following definition:

**Definition 4.** *We call a Gaussian matrix $\mathbf{G}$ matching a block-regular $\mathbf{A}$ if $\mathbf{G}$ is block-regular as well, $\mathbb{E}\mathbf{G}_i = \mathbb{E}\mathbf{A}_i = \mathbb{1}\boldsymbol{\mu}_i^T$ and for any row $\mathbf{g}$ of $\mathbf{G}_i$ and any row $\mathbf{a}$ of $\mathbf{A}_i$, it holds that $\mathbb{E}\mathbf{g}\mathbf{g}^T = \mathbb{E}\mathbf{a}\mathbf{a}^T$.*

### A.2 Assumptions for Theorem 2

**Assumptions 3.**  *1. $\mathbf{R}_x > \mathbf{0}$ is diagonal without loss of generality.*

2. *Let $\hat{p}_N$ be the empirical spectral density of $\mathbf{R}_x$, then $\hat{p}_N \rightsquigarrow p$ for some probability distribution $p$.*

3. *$\mathbf{w}_0$ is a white perturbation of true $\mathbf{w}_*$ in the $\mathbf{R}_x$ basis. That is, $\mathbf{w}_0 = \mathbf{w}_* + \mathbf{R}_x^{-1/2}\boldsymbol{\xi}$ where $\mathbb{E}\boldsymbol{\xi} = \mathbf{0}$, $\mathbb{E}\boldsymbol{\xi}\boldsymbol{\xi}^T = \frac{e_a}{d}\mathbf{I}$ for a fixed $e_a > 0$.*

The third assumption from Assumptions 3 represents the fact that $\mathbf{w}_0$ is a perturbation of the ground truth $\mathbf{w}_*$ in the eigenbasis of $R_x$.

### A.3 Assumptions for Theorem 3

**Assumptions 4.**  *1. The data matrix $\mathbf{X}$ satisfies parts (4)-(6) of Definition 2*

2. *The means $\boldsymbol{\mu}_1, \boldsymbol{\mu}_2$ are of norm 1 and for any deterministic matrix $\mathbf{C}$ of bounded operator norm it holds that $\boldsymbol{\mu}_1^T\mathbf{C}\boldsymbol{\mu}_1, \boldsymbol{\mu}_2^T\mathbf{C}\boldsymbol{\mu}_2$ and $\boldsymbol{\mu}_1^T\mathbf{C}\boldsymbol{\mu}_2$ converge in probability to $\frac{Tr(\mathbf{C})}{d}, \frac{Tr(\mathbf{C})}{d}$ and $\frac{rTr(\mathbf{C})}{d}$ respectively.*

3. *$\boldsymbol{\Sigma}_1, \boldsymbol{\Sigma}_2$ are diagonal.*

4. *Let $\hat{p}_N$ be the joint empirical spectral density of $\boldsymbol{\Sigma}_1, \boldsymbol{\Sigma}_2$, then $\hat{p}_N \rightsquigarrow p$, where the distribution $p$ is such that $p(s_1, s_2) = p(s_2, s_1)$ holds for all $s_1, s_2$.*

For a discussion of the first assumption from Assumptions 4, see the paragraph after Definition 2. The second assumption from Assumptions 4 is satisfied, for example, for any random vector of the form $\boldsymbol{\Sigma}^{1/2}\mathbf{z}$, where $\boldsymbol{\Sigma}$ is an arbitrary *PSD* matrix and $\mathbf{z}$ is i.i.d. The third assumption postulates that the means are normalized generic $d$-dimensional vectors with an angle $\arccos r$ between them. The fourth assumption says that $\boldsymbol{\Sigma}_1$ and $\boldsymbol{\Sigma}_2$ are simultaneously diagonalizable and, finally, the fifth assumption simply introduces notation for the joint density function of the eigenvalues of $\boldsymbol{\Sigma}_1$ and $\boldsymbol{\Sigma}_2$.

**Lemma 1.** *Assume that the feature vectors are equally likely to be drawn from class 1 or class 2 and Assumptions 4 hold. Then the optimal classifier is given by $\mathbf{w}_* = (\boldsymbol{\Sigma}_1 + \boldsymbol{\Sigma}_2)^{-1}(\boldsymbol{\mu}_1 - \boldsymbol{\mu}_2)$ and its classification error is equal to the following, where $Q(\cdot)$ is the integral of the tail of the standard normal distribution.*

$$\frac{1}{2}Q\left(\frac{Tr(\boldsymbol{\Sigma}_1 + \boldsymbol{\Sigma}_2)^{-1}\sqrt{1-r}}{\sqrt{2Tr(\boldsymbol{\Sigma}_1(\boldsymbol{\Sigma}_1 + \boldsymbol{\Sigma}_2)^{-2})}}\right) + \frac{1}{2}Q\left(\frac{Tr(\boldsymbol{\Sigma}_1 + \boldsymbol{\Sigma}_2)^{-1}\sqrt{1-r}}{\sqrt{2Tr(\boldsymbol{\Sigma}_2(\boldsymbol{\Sigma}_1 + \boldsymbol{\Sigma}_2)^{-2})}}\right)$$

# B Proof of Theorem 1

In this section we provide the complete proof of Theorem 1.

## B.1 Proof of part 1 of Theorem 1 when $\Psi = \Phi_\lambda$

We prove the universality of the objective value and then use it to prove the universality of a $\psi$ applied to the optimal solutions. To do so, we construct the perturbed objective. Let

$$\Phi_{\lambda,\epsilon}(\mathbf{A}) := \frac{1}{n} \min_{w \in \mathcal{S}_w} \frac{\lambda}{2} \|\mathbf{A}\mathbf{w} - \mathbf{y}\|^2 + f(\mathbf{w}) + \epsilon\psi(\mathbf{w})$$

Where $|\epsilon|$ is small enough that $\epsilon\psi(\mathbf{w}) + f(\mathbf{w})$ is $\rho$-strongly convex. For such $\epsilon$'s, we will show that $\Phi_{\lambda,\epsilon}(\mathbf{A})$ and $\Phi_{\lambda,\epsilon}(\mathbf{B})$ converge in probability to the same value and use that to deduce a similar result involving $\psi(\mathbf{w}_{\Phi_{\lambda,0}(\mathbf{A})}), \psi(\mathbf{w}_{\Phi_{\lambda,0}(\mathbf{B})})$. A key step is to reduce the proof to upper bounding the difference of expectations, to do so, the following proposition will be instrumental whose proof is given in Appendix D.

**Proposition 1.** *If for any Lipschitz function $g : \mathbb{R} \to \mathbb{R}$ and $t_1 > 0$, $\mathbb{P}\Big(|g(\Phi_{\lambda,\epsilon}(\mathbf{B})) - c| > t_1\Big) \to 0$ and for every twice differentiable $\tilde{g} : \mathbb{R} \to \mathbb{R}$ with bounded second derivative, we have that $\lim_{n\to\infty}\Big|\mathbb{E}_{\mathbf{A},\mathbf{y}}\tilde{g}(\Phi_{\lambda,\epsilon}(\mathbf{B})) - \mathbb{E}_{\mathbf{B},\mathbf{y}}\tilde{g}(\Phi_{\lambda,\epsilon}(\mathbf{A}))\Big| \to 0$ then for any $t_2 > t_1$.*

$$\lim_{n\to\infty} \mathbb{P}\Big(|g(\Phi_{\lambda,\epsilon}(\mathbf{A})) - c| > t_2\Big) = 0$$

Thus it suffices to prove $\lim_{n\to\infty}\Big|\mathbb{E}_{\mathbf{A},\mathbf{y}}g(\Phi_\lambda(\mathbf{B})) - \mathbb{E}_{\mathbf{B},\mathbf{y}}g(\Phi_\lambda(\mathbf{A}))\Big| = 0$ for every twice differentiable $g$ with bounded second derivative. As stated earlier, we proceed by Lindeberg's approach. Fixing $\lambda$ and $\epsilon$, to simplify notation we use $\Phi(\mathbf{A})$ for $\Phi_{\lambda,\epsilon}(\mathbf{A})$. For $0 \le j \le n$, let $q \in \mathbb{N}$ be the largest such that $j \ge \sum_{l=1}^{q} n_l$, and let $r = j - \sum_{l=1}^{q} n_l$ if , then

$$\hat{\mathbf{A}}_j = \begin{cases} \begin{bmatrix} \mathbf{a}_{1,1} & \mathbf{a}_{1,2} & \dots & \mathbf{a}_{q,r} & \mathbf{b}_{q,r+1} & \dots & \mathbf{b}_{k,n_k} \end{bmatrix}^T & r < n_{q+1} \\ \begin{bmatrix} \mathbf{a}_{1,1} & \mathbf{a}_{1,2} & \dots & \mathbf{a}_{q,r} & \mathbf{b}_{q+1,1} & \dots & \mathbf{b}_{k,n_k} \end{bmatrix}^T & r = n_{q+1} \end{cases}$$

It follwos that $\mathbf{A} = \hat{\mathbf{A}}_0$ and $\mathbf{B} = \hat{\mathbf{A}}_n$. Then we have by a telescopic sum

$$\Big|\mathbb{E}_{\mathbf{A},\mathbf{y}}g(\Phi(\mathbf{A})) - \mathbb{E}_{\mathbf{B},\mathbf{y}}g(\Phi(\mathbf{B}))\Big| = \Big|\mathbb{E}_{\mathbf{A},\mathbf{B},\mathbf{y}}\sum_{j=0}^{n-1} g(\Phi(\hat{\mathbf{A}}_j)) - g(\Phi(\hat{\mathbf{A}}_{j-1}))\Big|$$

$$\le \sum_{j=0}^{n-1}\Big|\mathbb{E}_{\hat{\mathbf{A}}_j,\mathbf{y}}g(\Phi(\hat{\mathbf{A}}_j)) - \mathbb{E}_{\hat{\mathbf{A}}_{j-1},\mathbf{y}}g(\Phi(\hat{\mathbf{A}}_{j-1}))\Big|$$

Now we define a new matrix by dropping the $j$'th row.

$$\tilde{\mathbf{A}}_j = \begin{cases} \begin{bmatrix} \mathbf{a}_{1,1} & \mathbf{a}_{1,2} & \dots & \mathbf{a}_{q,r-1} & \mathbf{b}_{q,r+1} & \dots & \mathbf{b}_{k,n_k} \end{bmatrix}^T & r < n_{q+1} \\ \begin{bmatrix} \mathbf{a}_{1,1} & \mathbf{a}_{1,2} & \dots & \mathbf{a}_{q,r-1} & \mathbf{b}_{q+1,1} & \dots & \mathbf{b}_{k,n_k} \end{bmatrix}^T & r = n_{q+1} \end{cases}$$

Note that

$$\Phi(\hat{\mathbf{A}}_j) = \frac{1}{n} \min_{w \in \mathcal{S}_w} \frac{\lambda}{2} \|\hat{\mathbf{A}}_j\mathbf{w} - \mathbf{y}\|^2 + f(\mathbf{w}) + \epsilon\psi(\mathbf{w})$$

$$= \frac{1}{n} \min_{w \in \mathcal{S}_w} \frac{\lambda}{2} \|\tilde{\mathbf{A}}_j\mathbf{w} - \tilde{\mathbf{y}}\|^2 + \frac{\lambda}{2}(\mathbf{a}_j^T\mathbf{w} - y_j)^2 + f(\mathbf{w}) + \epsilon\psi(\mathbf{w}) =: \mathcal{M}(\mathbf{a}_j)$$

$$\Phi(\hat{\mathbf{A}}_{j-1}) = \frac{1}{n} \min_{w \in \mathcal{S}_w} \frac{\lambda}{2} \|\hat{\mathbf{A}}_j\mathbf{w} - \mathbf{y}\|^2 + f(\mathbf{w}) + \epsilon\psi(\mathbf{w})$$

$$= \frac{1}{n} \min_{w \in \mathcal{S}_w} \frac{\lambda}{2} \|\tilde{\mathbf{A}}_j\mathbf{w} - \tilde{\mathbf{y}}\|^2 + \frac{\lambda}{2}(\mathbf{b}_j^T\mathbf{w} - y_j)^2 + f(\mathbf{w}) + \epsilon\psi(\mathbf{w}) =: \mathcal{M}(\mathbf{b}_j)$$

And we define $\mathcal{M}(\mathbf{a}) := \min_{\mathbf{w}\in\mathcal{S}_w} m(\mathbf{a}, \mathbf{w})$. So, we need to bound $\left|\mathbb{E}_{\hat{\mathbf{A}}_j, \mathbf{y}} g(\Phi(\hat{\mathbf{A}}_j)) - \mathbb{E}_{\hat{\mathbf{A}}_{j-1}, \mathbf{y}} g(\Phi(\hat{\mathbf{A}}_{j-1}))\right|$, to do so, we condition on $\tilde{\mathbf{A}}_j$:

$$\left|\mathbb{E}_{\hat{\mathbf{A}}_j, \mathbf{y}} g(\Phi(\hat{\mathbf{A}}_j)) - \mathbb{E}_{\hat{\mathbf{A}}_{j-1}, \mathbf{y}} g(\Phi(\hat{\mathbf{A}}_{j-1}))\right| = \left|\mathbb{E}_{\tilde{\mathbf{A}}_j, \tilde{\mathbf{y}}} \mathbb{E}_{\mathbf{a}_j, \mathbf{b}_j, y_j}\left[g(\mathcal{M}(\mathbf{a}_j)) - g(\mathcal{M}(\mathbf{b}_j))\Big|\tilde{\mathbf{A}}_j, \tilde{\mathbf{y}}\right]\right|$$

Let us define the optimization whose terms are shared by both of $\mathcal{M}(\mathbf{a}_j)$ and $\mathcal{M}(\mathbf{b}_j)$

$$\Theta := \frac{1}{n}\min_{\mathbf{w}\in\mathcal{S}_w}\frac{\lambda}{2}\|\tilde{\mathbf{A}}_j\mathbf{w} - \tilde{\mathbf{y}}\|^2 + f(\mathbf{w}) + \epsilon\psi(\mathbf{w})$$

Now note that since the second derivative of $g$ is bounded, we have

$$\left|g(\Theta) - g(\mathcal{M}(\mathbf{a}_j)) - g'(\Theta)(\mathcal{M}(\mathbf{a}_j) - \Theta)\right| \le \|g''\|_\infty(\Theta - \mathcal{M}(\mathbf{a}_j))^2$$

$$\left|g(\Theta) - g(\mathcal{M}(\mathbf{b}_j)) - g'(\Theta)(\mathcal{M}(\mathbf{b}_j) - \Theta)\right| \le \|g''\|_\infty(\Theta - \mathcal{M}(\mathbf{b}_j))^2 \qquad (13)$$

Adding and subtracting $g(\Theta)$ and taking expectation and moving inside, we have

$$\left|\mathbb{E}_{\tilde{\mathbf{A}}_j, \tilde{\mathbf{y}}}\mathbb{E}_{\mathbf{a}_j, \mathbf{b}_j, y_j}\left[g(\mathcal{M}(\mathbf{a}_j)) - g(\mathcal{M}(\mathbf{b}_j))\Big|\tilde{\mathbf{A}}_j, \tilde{\mathbf{y}}\right]\right|$$

$$\le \left|\mathbb{E}_{\tilde{\mathbf{A}}_j, \tilde{\mathbf{y}}}\mathbb{E}_{\mathbf{a}_j, \mathbf{b}_j, y_j}\left[g(\Theta) - g(\mathcal{M}(\mathbf{a}_j)) - g'(\Theta)(\mathcal{M}(\mathbf{a}_j) - \Theta)\Big|\tilde{\mathbf{A}}_j, \tilde{\mathbf{y}}\right]\right|$$

$$+ \left|\mathbb{E}_{\tilde{\mathbf{A}}_j, \tilde{\mathbf{y}}}\mathbb{E}_{\mathbf{a}_j, \mathbf{b}_j, y_j}\left[g(\Theta) - g(\mathcal{M}(\mathbf{b}_j)) - g'(\Theta)(\mathcal{M}(\mathbf{b}_j) - \Theta)\Big|\tilde{\mathbf{A}}_j, \tilde{\mathbf{y}}\right]\right|$$

$$+ \left|\mathbb{E}_{\tilde{\mathbf{A}}_j, \tilde{\mathbf{y}}}\mathbb{E}_{\mathbf{a}_j, \mathbf{b}_j, y_j}\left[g'(\Theta)(\mathcal{M}(\mathbf{a}_j) - \mathcal{M}(\mathbf{b}_j))\Big|\tilde{\mathbf{A}}_j, \tilde{\mathbf{y}}\right]\right|$$

Using 13, and by the independence of $\Theta$ and $\mathbf{a}_j, \mathbf{b}_j, y_j$, we have

$$\left|\mathbb{E}_{\tilde{\mathbf{A}}_j, \tilde{\mathbf{y}}}\mathbb{E}_{\mathbf{a}_j, \mathbf{b}_j, y_j}\left[g(\mathcal{M}(\mathbf{a}_j)) - g(\mathcal{M}(\mathbf{b}_j))\Big|\tilde{\mathbf{A}}_j, \tilde{\mathbf{y}}\right]\right| \le \left|\mathbb{E}_{\tilde{\mathbf{A}}_j, \tilde{\mathbf{y}}}g'(\Theta)\mathbb{E}_{\mathbf{a}_j, \mathbf{b}_j, y_j}\left[(\mathcal{M}(\mathbf{a}_j) - \mathcal{M}(\mathbf{b}_j))\Big|\tilde{\mathbf{A}}_j, \tilde{\mathbf{y}}\right]\right|$$

$$+ \|g''\|_\infty\mathbb{E}_{\tilde{\mathbf{A}}_j, \tilde{\mathbf{y}}, \mathbf{a}_j, y_j}(\Theta - \mathcal{M}(\mathbf{a}_j))^2 + \|g''\|_\infty\mathbb{E}_{\tilde{\mathbf{A}}_j, \tilde{\mathbf{y}}, \mathbf{b}_j, y_j}(\Theta - \mathcal{M}(\mathbf{b}_j))^2$$

$$\le \mathbb{E}_{\tilde{\mathbf{A}}_j, \tilde{\mathbf{y}}}|g'(\Theta)|\left|\mathbb{E}_{\mathbf{a}_j, \mathbf{b}_j, y_j}\left[(\mathcal{M}(\mathbf{a}_j) - \mathcal{M}(\mathbf{b}_j))\Big|\tilde{\mathbf{A}}_j, \tilde{\mathbf{y}}\right]\right|$$

$$+ \|g''\|_\infty\left(\mathbb{E}_{\tilde{\mathbf{A}}_j, \tilde{\mathbf{y}}, \mathbf{a}_j, y_j}(\Theta - \mathcal{M}(\mathbf{a}_j))^2 + \mathbb{E}_{\tilde{\mathbf{A}}_j, \tilde{\mathbf{y}}, \mathbf{b}_j, y_j}(\Theta - \mathcal{M}(\mathbf{b}_j))^2\right)$$

$$\le \|g'\|_\infty\mathbb{E}_{\tilde{\mathbf{A}}_j, \tilde{\mathbf{y}}}\left|\mathbb{E}_{\mathbf{a}_j, \mathbf{b}_j, y_j}\left[(\mathcal{M}(\mathbf{a}_j) - \mathcal{M}(\mathbf{b}_j))\Big|\tilde{\mathbf{A}}_j, \tilde{\mathbf{y}}\right]\right|$$

$$+ \|g''\|_\infty\left(\mathbb{E}_{\tilde{\mathbf{A}}_j, \tilde{\mathbf{y}}, \mathbf{a}_j, y_j}(\Theta - \mathcal{M}(\mathbf{a}_j))^2 + \mathbb{E}_{\tilde{\mathbf{A}}_j, \tilde{\mathbf{y}}, \mathbf{b}_j, y_j}(\Theta - \mathcal{M}(\mathbf{b}_j))^2\right) \qquad (14)$$

Thus we focus on $\mathcal{M}(\mathbf{a}_j), \mathcal{M}(b_j)$ conditioned on $\tilde{\mathbf{A}}_j$. From now on, we drop the index $j$ from $\tilde{\mathbf{A}}_j$, $\mathbf{a}_j, \mathbf{b}_j, y_j$. Intuitively, we show that by dropping $(\mathbf{a}^T\mathbf{w} - y)^2$ and $(\mathbf{b}^T\mathbf{w} - y)^2$ and approximating $f, \psi$ by their second-order Taylor expansion, the objective values $\Phi(\hat{\mathbf{A}}_j), \Phi(\hat{\mathbf{A}}_{j-1})$ would not change much. Which implies $\Phi(\hat{\mathbf{A}}_j)$ and $\Phi(\hat{\mathbf{A}}_j)$ are also close in value.

Let $u := \arg\min\frac{\lambda}{2}\|\tilde{\mathbf{A}}\mathbf{w} - \tilde{\mathbf{y}}\|^2 + f(\mathbf{w}) + \epsilon\psi(\mathbf{w})$ and it is unique because of strong convexity. Now we use the second order Taylor expansion of $f + \epsilon\psi$:

$$f(\mathbf{w}) + \epsilon\psi(\mathbf{w}) = f(\mathbf{u}) + \epsilon\psi(\mathbf{u}) + \nabla(f(\mathbf{u}) + \epsilon\psi(\mathbf{u}))^T(\mathbf{w} - \mathbf{u})$$

$$+ \frac{1}{2}(\mathbf{w} - \mathbf{u})^T\nabla^2(f(\mathbf{u}) + \epsilon\psi(\mathbf{u}))(\mathbf{w} - \mathbf{u}) + R(\mathbf{w} - \mathbf{u})$$

Where $\lim_{\mathbf{w}\to\mathbf{u}}\frac{R(\mathbf{w}-\mathbf{u})}{\|\mathbf{w}-\mathbf{u}\|^2} = 0$. Let $\mathbf{g} := \nabla(f(\mathbf{u}) + \epsilon\psi(\mathbf{u}))$, $\mathbf{H} := \nabla^2(f(\mathbf{u}) + \epsilon\psi(\mathbf{u}))$. Moreover, let

$$\mathcal{L}(\mathbf{a}) := \min_{\mathbf{w}\in\mathcal{S}_w}\ell(\mathbf{a}, \mathbf{w}) := \frac{1}{n}\min_{\mathbf{w}\in\mathcal{S}_w}\frac{\lambda}{2}\|\tilde{\mathbf{A}}\mathbf{w} - \tilde{\mathbf{y}}\|^2 + \frac{\lambda}{2}(\mathbf{a}^T\mathbf{w} - y)^2$$

$$+ f(\mathbf{u}) + \epsilon\psi(\mathbf{u}) + \mathbf{g}^T(\mathbf{w} - \mathbf{u}) + \frac{1}{2}(\mathbf{w} - \mathbf{u})^T\mathbf{H}(\mathbf{w} - \mathbf{u})$$

Note that $\mathbf{a}$ and $\mathbf{b}$ match in distribution up to the second moment, which implies that

$$\mathbb{E}_{\mathbf{a},y}(\mathbf{a}^T\mathbf{u}-y)^2 = \mathbb{E}_{\mathbf{b},y}(\mathbf{b}^T\mathbf{u}-y)^2$$

$$\mathbb{E}_{\mathbf{a}}\mathbf{a}^T\mathbf{\Omega}^{-1}\mathbf{a} = \mathbb{E}_{\mathbf{b}}\mathbf{b}^T\mathbf{\Omega}^{-1}\mathbf{b}$$

By triangle inequality we obtain

$$\mathbb{E}_{\tilde{\mathbf{A}},\tilde{\mathbf{y}}}\left|\mathbb{E}_{\mathbf{a},\mathbf{b},y}\left[(\mathcal{M}(\mathbf{a})-\mathcal{M}(\mathbf{b}))\Big|\tilde{\mathbf{A}},\tilde{\mathbf{y}}\right]\right|$$

$$\leq \mathbb{E}_{\tilde{\mathbf{A}},\tilde{\mathbf{y}}}\left|\mathbb{E}_{\mathbf{a},y}\left[(\mathcal{M}(\mathbf{a})-\mathcal{L}(\mathbf{a}))\Big|\tilde{\mathbf{A}},\tilde{\mathbf{y}}\right]\right| + \mathbb{E}_{\tilde{\mathbf{A}},\tilde{\mathbf{y}}}\left|\mathbb{E}_{\mathbf{b},y}\left[(\mathcal{M}(\mathbf{b})-\mathcal{L}(\mathbf{b}))\Big|\tilde{\mathbf{A}},\tilde{\mathbf{y}}\right]\right|$$

$$+ \mathbb{E}_{\tilde{\mathbf{A}},\tilde{\mathbf{y}}}\left|\mathbb{E}_{\mathbf{a},y}\left[\mathcal{L}(\mathbf{a})-\Theta-\frac{\lambda}{2n}\frac{\mathbb{E}_{\mathbf{a},y}(\mathbf{a}^T\mathbf{u}-y)^2}{1+\lambda\mathbb{E}_{\mathbf{a}}\mathbf{a}^T\mathbf{\Omega}^{-1}\mathbf{a}}\Big|\tilde{\mathbf{A}},\tilde{\mathbf{y}}\right]\right| + \mathbb{E}_{\tilde{\mathbf{A}},\tilde{\mathbf{y}}}\left|\mathbb{E}_{\mathbf{b},y}\left[\mathcal{L}(\mathbf{b})-\Theta-\frac{\lambda}{2n}\frac{\mathbb{E}_{\mathbf{b},y}(\mathbf{b}^T\mathbf{u}-y)^2}{1+\lambda\mathbb{E}_{\mathbf{b}}\mathbf{b}^T\mathbf{\Omega}^{-1}\mathbf{b}}\Big|\tilde{\mathbf{A}},\tilde{\mathbf{y}}\right]\right|$$

$$\leq \mathbb{E}_{\tilde{\mathbf{A}},\tilde{\mathbf{y}},\mathbf{a},y}|\mathcal{M}(\mathbf{a})-\mathcal{L}(\mathbf{a})| + \mathbb{E}_{\tilde{\mathbf{A}},\tilde{\mathbf{y}},\mathbf{b},y}|\mathcal{M}(\mathbf{b})-\mathcal{L}(\mathbf{b})|$$

$$+ \mathbb{E}_{\tilde{\mathbf{A}},\tilde{\mathbf{y}}}\left|\mathbb{E}_{\mathbf{a},y}\left[\mathcal{L}(\mathbf{a})-\Theta-\frac{\lambda}{2n}\frac{(\mathbf{a}^T\mathbf{u}-y)^2}{1+\lambda\mathbb{E}_{\mathbf{a}}\mathbf{a}^T\mathbf{\Omega}^{-1}\mathbf{a}}\Big|\tilde{\mathbf{A}},\tilde{\mathbf{y}}\right]\right| + \mathbb{E}_{\tilde{\mathbf{A}},\tilde{\mathbf{y}}}\left|\mathbb{E}_{\mathbf{b},y}\left[\mathcal{L}(\mathbf{b})-\Theta-\frac{\lambda}{2n}\frac{(\mathbf{b}^T\mathbf{u}-y)^2}{1+\lambda\mathbb{E}_{\mathbf{b}}\mathbf{b}^T\mathbf{\Omega}^{-1}\mathbf{b}}\Big|\tilde{\mathbf{A}},\tilde{\mathbf{y}}\right]\right|$$

$$\leq \mathbb{E}_{\tilde{\mathbf{A}},\tilde{\mathbf{y}},\mathbf{a},y}|\mathcal{M}(\mathbf{a})-\mathcal{L}(\mathbf{a})| + \mathbb{E}_{\tilde{\mathbf{A}},\tilde{\mathbf{y}},\mathbf{b},y}|\mathcal{M}(\mathbf{b})-\mathcal{L}(\mathbf{b})|$$

$$+ \frac{\lambda}{2n}\mathbb{E}_{\tilde{\mathbf{A}},\tilde{\mathbf{y}},\mathbf{a},y}\left|\frac{(\mathbf{a}^T\mathbf{u}-y)^2}{1+\lambda\mathbf{a}^T\mathbf{\Omega}^{-1}\mathbf{a}}-\frac{(\mathbf{a}^T\mathbf{u}-y)^2}{1+\lambda\mathbb{E}_{\mathbf{a}}\mathbf{a}^T\mathbf{\Omega}^{-1}\mathbf{a}}\right| + \frac{\lambda}{2n}\mathbb{E}_{\tilde{\mathbf{A}},\tilde{\mathbf{y}},\mathbf{b},y}\left|\frac{(\mathbf{b}^T\mathbf{u}-y)^2}{1+\lambda\mathbf{b}^T\mathbf{\Omega}^{-1}\mathbf{b}}-\frac{(\mathbf{b}^T\mathbf{u}-y)^2}{1+\lambda\mathbb{E}_{\mathbf{b}}\mathbf{b}^T\mathbf{\Omega}^{-1}\mathbf{b}}\right|$$

$$\tag{15}$$

First we will provide an upper bound for the third and fourth terms in 15. As the function $x \mapsto \frac{1}{1+x}$ is 1-Lipschitz and using Cauchy Schwartz we arrive at

$$\frac{\lambda}{2n}\mathbb{E}_{\tilde{\mathbf{A}},\tilde{\mathbf{y}},\mathbf{a},y}\left|\frac{(\mathbf{a}^T\mathbf{u}-y)^2}{1+\lambda\mathbf{a}^T\mathbf{\Omega}^{-1}\mathbf{a}}-\frac{(\mathbf{a}^T\mathbf{u}-y)^2}{1+\lambda\mathbb{E}_{\mathbf{a}}\mathbf{a}^T\mathbf{\Omega}^{-1}\mathbf{a}}\right| \leq \frac{\lambda^2}{2n}\mathbb{E}_{\tilde{\mathbf{A}},\tilde{\mathbf{y}},\mathbf{a},y}\left|(\mathbf{a}^T\mathbf{u}-y)^2(\mathbf{a}^T\mathbf{\Omega}^{-1}\mathbf{a}-\mathbb{E}_{\mathbf{a}}\mathbf{a}^T\mathbf{\Omega}^{-1}\mathbf{a})\right|$$

$$\tag{16}$$

$$\leq \frac{\lambda^2}{2n}\sqrt{\mathbb{E}_{\tilde{\mathbf{A}},\tilde{\mathbf{y}},\mathbf{a},y}(\mathbf{a}^T\mathbf{u}-y)^4\mathbb{E}_{\tilde{\mathbf{A}},\tilde{\mathbf{y}},\mathbf{a},y}(\mathbf{a}^T\mathbf{\Omega}^{-1}\mathbf{a}-\mathbb{E}_{\mathbf{a}}\mathbf{a}^T\mathbf{\Omega}^{-1}\mathbf{a})^2}$$

$$\tag{17}$$

By Lemma 7, there exists a constant $C_1 < \infty$ such that $\mathbb{E}_{\tilde{\mathbf{A}},\tilde{\mathbf{y}},\mathbf{a},y}(\mathbf{a}^T\mathbf{u}-y)^4 < C_1$ for any $n$. Furthermore, by the assumptions, $\lim_{n\to\infty}\mathbb{E}_{\tilde{\mathbf{A}},\tilde{\mathbf{y}},\mathbf{a},y}(\mathbf{a}^T\mathbf{\Omega}^{-1}\mathbf{a}-\mathbb{E}_{\mathbf{a}}\mathbf{a}^T\mathbf{\Omega}^{-1}\mathbf{a})^2 = 0$.

Now for the first and second terms in (15), using Lemma 5, we obtain with probability 1,

$$|\mathcal{M}(\mathbf{a})-\mathcal{L}(\mathbf{a})| \leq \frac{8C_{f+\epsilon\psi}}{n}\|\mathbf{u}-\mathbf{w}_\mathcal{L}\|_3^3 \mathbb{1}\{\|\mathbf{u}-\mathbf{w}_\mathcal{L}\|_3 \leq \frac{\rho}{18C_{f+\epsilon\psi}}\}$$

$$+ \frac{\lambda}{2n}(\mathbf{a}^T\mathbf{u}-y)^2\mathbb{1}\{\|\mathbf{u}-\mathbf{w}_\mathcal{L}\|_3 > \frac{\rho}{18C_{f+\epsilon\psi}}\}$$

Which implies

$$\mathbb{E}_{\tilde{\mathbf{A}},\tilde{\mathbf{y}},\mathbf{a},y}|\mathcal{M}(\mathbf{a})-\mathcal{L}(\mathbf{a})| \leq \frac{8C_{f+\epsilon\psi}}{n}\mathbb{E}_{\tilde{\mathbf{A}},\tilde{\mathbf{y}},\mathbf{a},y}\left[\|\mathbf{u}-\mathbf{w}_\mathcal{L}\|_3^3\mathbb{1}\{\|\mathbf{u}-\mathbf{w}_\mathcal{L}\|_3 \leq \frac{\rho}{18C_{f+\epsilon\psi}}\}\right]$$

$$+ \frac{\lambda}{2n}\mathbb{E}_{\tilde{\mathbf{A}},\tilde{\mathbf{y}},\mathbf{a},y}\left[(\mathbf{a}^T\mathbf{u}-y)^2\mathbb{1}\{\|\mathbf{u}-\mathbf{w}_\mathcal{L}\|_3 > \frac{\rho}{18C_{f+\epsilon\psi}}\}\right]$$

We apply Cauchy-Schwarz to each term:

$$\mathbb{E}_{\tilde{\mathbf{A}},\tilde{\mathbf{y}},\mathbf{a},y}|\mathcal{M}(\mathbf{a})-\mathcal{L}(\mathbf{a})| \leq \frac{8C_{f+\epsilon\psi}}{n}\mathbb{E}_{\tilde{\mathbf{A}},\tilde{\mathbf{y}},\mathbf{a},y}\|\mathbf{u}-\mathbf{w}_\mathcal{L}\|_3^3$$

$$+ \frac{\lambda}{2n}\sqrt{\mathbb{E}_{\tilde{\mathbf{A}},\tilde{\mathbf{y}},\mathbf{a},y}\mathbb{1}^2\{\|\mathbf{u}-\mathbf{w}_\mathcal{L}\|_3 > \frac{\rho}{18C_{f+\epsilon\psi}}\}\mathbb{E}_{\tilde{\mathbf{A}},\tilde{\mathbf{y}},\mathbf{a},y}(\mathbf{a}^T\mathbf{u}-y)^4}$$

$$= \frac{8C_{f+\epsilon\psi}}{n}\mathbb{E}_{\tilde{\mathbf{A}},\tilde{\mathbf{y}},\mathbf{a},y}\|\mathbf{u}-\mathbf{w}_\mathcal{L}\|_3^3$$

$$+ \frac{\lambda}{2n}\sqrt{\mathbb{P}_{\tilde{\mathbf{A}},\tilde{\mathbf{y}},\mathbf{a},y}(\|\mathbf{u}-\mathbf{w}_\mathcal{L}\|_3 > \frac{\rho}{18C_{f+\epsilon\psi}})\mathbb{E}_{\tilde{\mathbf{A}},\tilde{\mathbf{y}},\mathbf{a},y}(\mathbf{a}^T\mathbf{u}-y)^4}$$

To deal with the terms involving $\mathbb{P}$, we leverage Markov's inequality

$$\mathbb{E}_{\tilde{\mathbf{A}},\tilde{\mathbf{y}},\mathbf{a},y}|\mathcal{M}(\mathbf{a}) - \mathcal{L}(\mathbf{a})| \leq \frac{8C_{f+\epsilon\psi}}{n}\mathbb{E}_{\tilde{\mathbf{A}},\tilde{\mathbf{y}},\mathbf{a},y}\|\mathbf{u} - \mathbf{w}_{\mathcal{L}}\|_3^3$$

$$+ \frac{\lambda}{2n}\Big(\frac{18C_{f+\epsilon\psi}}{\rho}\Big)^{3/2}\sqrt{\mathbb{E}_{\tilde{\mathbf{A}},\tilde{\mathbf{y}},\mathbf{a},y}\|\mathbf{u} - \mathbf{w}_{\mathcal{L}}\|_3^3\mathbb{E}_{\tilde{\mathbf{A}},\tilde{\mathbf{y}},\mathbf{a},y}(\mathbf{a}^T\mathbf{u} - y)^4}$$

By Lemma 7, there exists a $C_2 < \infty$ such that $\mathbb{E}_{\tilde{\mathbf{A}},\tilde{\mathbf{y}},\mathbf{a},y}\|\mathbf{u}-\mathbf{w}_{\mathcal{L}}\|_3^3 \leq \frac{C_2}{d}$, and recall that $\mathbb{E}_{\tilde{\mathbf{A}},\tilde{\mathbf{y}},\mathbf{a},y}(\mathbf{a}^T\mathbf{u} - y)^4 < C_1$ for any $n$. Thus

$$\mathbb{E}_{\tilde{\mathbf{A}},\tilde{\mathbf{y}},\mathbf{a},y}|\mathcal{M}(\mathbf{a}) - \mathcal{L}(\mathbf{a})| \leq \frac{8C_2C_{f+\epsilon\psi}}{nd} + \frac{\lambda\sqrt{C_1}}{2n\sqrt{d}}\Big(\frac{18C_{f+\epsilon\psi}}{\rho}\Big)^{3/2} \leq \frac{C_a}{n\sqrt{d}} \tag{18}$$

For some constant $\tilde{C} = \frac{\lambda\sqrt{C_1}}{2}\big(\frac{18C_{f+\epsilon\psi}}{\rho}\big)^{3/2} + 8C_2C_{f+\epsilon\psi}$. Plugging (16) and (18) in (15), yields

$$\mathbb{E}_{\tilde{\mathbf{A}},\tilde{\mathbf{y}}}\Big|\mathbb{E}_{\mathbf{a},\mathbf{b},y}\Big[(\mathcal{M}(\mathbf{a}) - \mathcal{M}(\mathbf{b}))\Big|\tilde{\mathbf{A}},\tilde{\mathbf{y}}\Big]\Big| \leq \frac{\tilde{C}+\tilde{C}'}{n\sqrt{d}} + \frac{\lambda^2\tilde{C}}{2n}\sqrt{\mathbb{E}_{\tilde{\mathbf{A}},\tilde{\mathbf{y}}}Var_{\mathbf{a}}(\mathbf{a}^T\mathbf{\Omega}^{-1}\mathbf{a})} + \frac{\lambda^2\tilde{C}'}{2n}\sqrt{\mathbb{E}_{\tilde{\mathbf{A}},\tilde{\mathbf{y}}}Var_{\mathbf{b}}(\mathbf{b}^T\mathbf{\Omega}^{-1}\mathbf{b})}$$

For the other terms in (14), we have by Lemmas 4, 7:

$$\|g''\|_\infty\mathbb{E}_{\tilde{\mathbf{A}},\tilde{\mathbf{y}},\mathbf{a},y}(\Theta - \mathcal{M}(\mathbf{a}))^2 + \|g''\|_\infty\mathbb{E}_{\tilde{\mathbf{A}},\tilde{\mathbf{y}},\mathbf{b},y}(\Theta - \mathcal{M}(\mathbf{b}))^2$$

$$\leq \|g''\|_\infty\mathbb{E}_{\tilde{\mathbf{A}},\tilde{\mathbf{y}},\mathbf{a},y}\frac{\lambda^2}{4n^2}(\mathbf{a}^T\mathbf{u} - y)^2 + \|g''\|_\infty\mathbb{E}_{\tilde{\mathbf{A}},\tilde{\mathbf{y}},\mathbf{b},y}\frac{\lambda^2}{4n^2}(\mathbf{b}^T\mathbf{u} - y)^2$$

$$\leq \frac{\lambda^2\|g''\|_\infty(\tilde{C}_1 + \tilde{C}_1')}{4n^2}$$

Plugging in (14), we obtain

$$\Big|\mathbb{E}_{\tilde{\mathbf{A}},\tilde{\mathbf{y}}}\mathbb{E}_{\mathbf{a},\mathbf{b},y}\Big[g(\mathcal{M}(\mathbf{a})) - g(\mathcal{M}(\mathbf{b}))\Big|\tilde{\mathbf{A}},\tilde{\mathbf{y}}\Big]\Big|$$

$$\leq \|g'\|_\infty\Big(\frac{\tilde{C}+\tilde{C}'}{n\sqrt{d}} + \frac{\lambda^2\tilde{C}}{2n}\mathbb{E}_{\tilde{\mathbf{A}},\tilde{\mathbf{y}}}Var_{\mathbf{a}}(\mathbf{a}^T\mathbf{\Omega}^{-1}\mathbf{a}) + \frac{\lambda^2\tilde{C}'}{2n}\mathbb{E}_{\tilde{\mathbf{A}},\tilde{\mathbf{y}}}Var_{\mathbf{b}}(\mathbf{b}^T\mathbf{\Omega}^{-1}\mathbf{b})\Big) + \frac{\lambda^2\|g''\|_\infty(\tilde{C}+\tilde{C}')}{4n^2}$$

Therefore,

$$\Big|\mathbb{E}_{\mathbf{A},\mathbf{y}}g(\Phi(\mathbf{A})) - \mathbb{E}_{\mathbf{B},\mathbf{y}}g(\Phi(\mathbf{B}))\Big| \leq \sum_{j=0}^{n-1}\Big|\mathbb{E}_{\hat{\mathbf{A}}_j,\mathbf{y}}g(\Phi(\hat{\mathbf{A}}_j)) - \mathbb{E}_{\hat{\mathbf{A}}_{j-1},\mathbf{y}}g(\Phi(\hat{\mathbf{A}}_{j-1}))\Big|$$

$$\leq \sum_{j=0}^{n-1}\|g'\|_\infty\Big(\frac{\tilde{C}+\tilde{C}'}{n\sqrt{d}} + \frac{\lambda^2\tilde{C}}{2n}\mathbb{E}_{\tilde{\mathbf{A}}_j,\tilde{\mathbf{y}}_j}Var_{\mathbf{a}_j}(\mathbf{a}_j^T\mathbf{\Omega}^{-1}\mathbf{a}_j) + \frac{\lambda^2\tilde{C}'}{2n}\mathbb{E}_{\tilde{\mathbf{A}}_j,\tilde{\mathbf{y}}_j}Var_{\mathbf{b}_j}(\mathbf{b}_j^T\mathbf{\Omega}^{-1}\mathbf{b}_j)\Big) + \frac{\lambda^2\|g''\|_\infty(\tilde{C}+\tilde{C}')}{4n^2}$$

$$= \frac{\|g'\|_\infty(\tilde{C}+\tilde{C}')}{\sqrt{d}} + \frac{\lambda^2\|g''\|_\infty(\tilde{C}+\tilde{C}')}{4n}$$

$$+ \sum_{j=0}^{n-1}\|g'\|_\infty\Big(\frac{\lambda^2\tilde{C}}{2n}\sqrt{\mathbb{E}_{\tilde{\mathbf{A}}_j,\tilde{\mathbf{y}}_j}Var_{\mathbf{a}_j}(\mathbf{a}_j^T\mathbf{\Omega}^{-1}\mathbf{a}_j)} + \frac{\lambda^2\tilde{C}'}{2n}\sqrt{\mathbb{E}_{\tilde{\mathbf{A}}_j,\tilde{\mathbf{y}}_j}Var_{\mathbf{b}_j}(\mathbf{b}_j^T\mathbf{\Omega}^{-1}\mathbf{b}_j)}\Big)$$

Now note that since $\|\mathbf{\Omega}\|_{op} \leq \frac{1}{\rho}$ with probability 1, $\lim_{n\to\infty}Var_{\mathbf{a}_j}(\mathbf{a}_j^T\mathbf{\Omega}^{-1}\mathbf{a}_j) = 0$ from assumptions, thus for every class $[j]$, there exists a function $\zeta_{[j]}(n)$ that $Var_{\mathbf{a}_j}(\mathbf{a}_j^T\mathbf{\Omega}^{-1}\mathbf{a}_j) \leq \zeta_{[j]}^2(n)$ for every $n \in \mathbb{N}$ and $\lim_{n\to\infty}\zeta_{[j]}(n) = 0$. Thus

$$\Big|\mathbb{E}_{\mathbf{A},\mathbf{y}}g(\Phi(\mathbf{A})) - \mathbb{E}_{\mathbf{B},\mathbf{y}}g(\Phi(\mathbf{B}))\Big| \leq \frac{\|g'\|_\infty(\tilde{C}+\tilde{C}')}{\sqrt{d}} + \frac{\lambda^2\|g''\|_\infty(\tilde{C}+\tilde{C}')}{4n} + \lambda^2\|g'\|_\infty(\tilde{C}+\tilde{C}')\sum_{i=1}^k\frac{n_i}{2n}\zeta_i(n)$$

$$\leq \frac{\|g'\|_\infty(\tilde{C}+\tilde{C}')}{\sqrt{d}} + \frac{\lambda^2\|g''\|_\infty(\tilde{C}+\tilde{C}')}{4n} + \lambda^2\|g'\|_\infty(\tilde{C}+\tilde{C}')\max_{1\leq i\leq k}\frac{n_i}{2n}\zeta_i(n)$$

Now since $k$ is finite and $n_i = \Theta(n)$ for every $1 \leq i \leq k$, we have that $\lim_{n\to\infty}\max_{1\leq i\leq k}\frac{n_i}{2n}\zeta_i(n) = 0$. Thus for every $\lambda, \epsilon > 0$

$$\lim_{n\to\infty}\Big|\mathbb{E}_{\mathbf{A},\mathbf{y}}g(\Phi_{\lambda,\epsilon}(\mathbf{A})) - \mathbb{E}_{\mathbf{B},\mathbf{y}}g(\Phi_{\lambda,\epsilon}(\mathbf{B}))\Big| = 0$$

and the proof of part 1 of Theorem 1 is concluded.

## B.2 Proof of part 1 of Theorem 1 when $\Psi = \Phi$

Now to prove the part 1 of Theorem 1, we will combine the following lemma with the previous section. For the ease of notation, we drop the dependency on $\epsilon$. Recall that since $f$ is $\rho$-strongly convex, we can write

$$\Phi_\lambda^n(A) = \min_{\mathbf{w}} \frac{\lambda}{2}\|\mathbf{A}\mathbf{w} - \mathbf{y}\|_2^2 + \frac{\rho}{2}\|\mathbf{w}\|_2^2 + h(\mathbf{w})$$

For some convex $h : \mathbb{R}^d \to \mathbb{R}$.

**Lemma 2.** *If for every $\lambda > 0$, we have $\Phi_\lambda^n(A) \xrightarrow{\mathbb{P}} c_\lambda$, then for every Lipschitz function $g : \mathbb{R} \to \mathbb{R}$, we have $g(\sup_{\lambda>0} \Phi_\lambda^n(A)) \xrightarrow{\mathbb{P}} g(\sup_{\lambda>0} c_\lambda)$ and $g(\sup_{\lambda>0} \Phi_\lambda^n(B)) \xrightarrow{\mathbb{P}} g(\sup_{\lambda>0} c_\lambda)$*

*Proof.* First note that for a given $t > 0$, we have for every Lipschitz function $g$

$$\mathbb{P}(|g(\Phi_\lambda^n) - g(c_\lambda)| > t) \le \mathbb{P}(|\Phi_\lambda^n - c_\lambda| > \frac{t}{L})$$

Letting $\delta := \frac{t}{L}, \tilde{\delta}_1 > 0$, we know that for every $\lambda \ge 0$, there exists $N_1 \in \mathbb{N}$ such that for every $n \ge N_1$

$$\mathbb{P}(|\Phi_\lambda^n - c_\lambda| > \delta) < \tilde{\delta}_1$$

We will show that an $R_1 > 0$ can be chosen in such a way that there exists $N_2 \in \mathbb{N}$ such that for every $n \ge N_2$

$$\mathbb{P}(\sup_{\lambda>0} \Phi_\lambda^n - \sup_{0<\lambda<R_1} \Phi_\lambda^n > \delta) \le \tilde{\delta}_2$$

To see this, by the monotonicity of $\Phi_\lambda^n$ in $\lambda$, we have by the fundamental theorem of calculus for a given $R_1$

$$\mathbb{P}(\sup_{\lambda>0} \Phi_\lambda^n - \sup_{0<\lambda<R_1} \Phi_\lambda^n > \delta) = \mathbb{P}(\lim_{\lambda\to\infty} \Phi_\lambda^n - \Phi_{R_1}^n > \delta) = \mathbb{P}\left(\int_{R_1}^\infty \frac{\partial \Phi_\lambda^n}{\partial \lambda} ds > \delta\right)$$

By Danskin's theorem,

$$\mathbb{P}(\sup_{\lambda>0} \Phi_\lambda^n - \sup_{0<\lambda<R_1} \Phi_\lambda^n > \delta) = \mathbb{P}\left(\frac{1}{n}\int_{R_1}^\infty \|\mathbf{A}\mathbf{w}(s) - \mathbf{y}\|^2 ds > \delta\right) \tag{19}$$

Now for the gradient of the main optimization we have, by $g := \nabla h(\mathbf{w})$

$$\lambda \mathbf{A}^T(\mathbf{A}\mathbf{w} - \mathbf{y}) + \rho\mathbf{w} + \mathbf{g} = 0$$

Which implies $\mathbf{w} = (\lambda\mathbf{A}^T\mathbf{A} + \rho\mathbf{I})^{-1}(\lambda\mathbf{A}^T\mathbf{y} - \mathbf{g})$. Hence by matrix inversion lemma,

$$\mathbf{A}\mathbf{w} - \mathbf{y} = -(\mathbf{I} - \mathbf{A}(\lambda\mathbf{A}^T\mathbf{A} + \rho\mathbf{I})^{-1}\lambda\mathbf{A}^T)\mathbf{y} - \mathbf{A}(\lambda\mathbf{A}^T\mathbf{A} + \rho\mathbf{I})^{-1}\mathbf{g}$$
$$= -\rho(\lambda\mathbf{A}\mathbf{A}^T + \rho I)^{-1}\mathbf{y} - \mathbf{A}(\lambda\mathbf{A}^T\mathbf{A} + \rho\mathbf{I})^{-1}\mathbf{g} = -\rho(\lambda\mathbf{A}\mathbf{A}^T + \rho I)^{-1}\mathbf{y} - (\lambda\mathbf{A}\mathbf{A}^T + \rho I)^{-1}\mathbf{A}\mathbf{g}$$

By triangle inequality and definition of the operator norm we have

$$\|\mathbf{A}\mathbf{w} + \mathbf{y}\|_2^2 \le 4\rho^2\|(\lambda\mathbf{A}\mathbf{A}^T + \rho I)^{-1}\|_{op}^2\|\mathbf{y}\|_2^2 + 4\|(\lambda\mathbf{A}\mathbf{A}^T + \rho I)^{-1}\mathbf{A}\|_{op}^2\|\mathbf{g}\|_2^2$$

Whence plugging back in 19 yields

$$\mathbb{P}(\sup_{\lambda>0} \Phi_\lambda^n - \sup_{0<\lambda<R_1} \Phi_\lambda^n > \delta) \le \mathbb{P}\left(\frac{4}{n}\int_{R_1}^\infty \rho^2\|(s\mathbf{A}\mathbf{A}^T + \rho I)^{-1}\|_{op}^2\|\mathbf{y}\|_2^2 + \|(s\mathbf{A}\mathbf{A}^T + \rho I)^{-1}\mathbf{A}\|_{op}^2\|\mathbf{g}(s)\|_2^2 ds > \delta\right)$$

$$\le \mathbb{P}\left(\frac{4\rho^2\|\mathbf{y}\|_2^2}{n}\int_{R_1}^\infty \|(s\mathbf{A}\mathbf{A}^T + \rho I)^{-1}\|_{op}^2 > \frac{\delta}{2}\right)$$

$$+ \mathbb{P}\left(\frac{4}{n}\int_{R_1}^\infty \|(s\mathbf{A}\mathbf{A}^T + \rho I)^{-1}\mathbf{A}\|_{op}^2\|\mathbf{g}(s)\|_2^2 ds > \frac{\delta}{2}\right)$$

For the norm of $\mathbf{g}$, by assumption, there exists a $C_g$ independent of $\lambda$, such that $\frac{\|\mathbf{g}(s)\|_2^2}{n} \le C_g$ with high probability.

Consider the following the events

$$\mathcal{T}_y := \left\{ \frac{\|\mathbf{y}\|_2^2}{n} \le C_y \right\}$$

$$\mathcal{T}_g := \left\{ \frac{\|\mathbf{g}(s)\|_2^2}{n} \le C_g \right\}$$

$$\mathcal{T}_A := \left\{ s_{\min}(\mathbf{A}\mathbf{A}^T) \ge C_A \right\}$$

We further have

$$\mathbb{P}\left( \frac{4\rho^2 \|\mathbf{y}\|_2^2}{n} \int_{R_1}^{\infty} \|(s\mathbf{A}\mathbf{A}^T + \rho I)^{-1}\|_{op}^2 ds > \frac{\delta}{2} \right) \le \mathbb{P}(\mathcal{T}_y^c) + \mathbb{P}\left( 4\rho^2 C_y \int_{R_1}^{\infty} \|(s\mathbf{A}\mathbf{A}^T + \rho I)^{-1}\|_{op}^2 ds > \frac{\delta}{2} \right)$$

$$\mathbb{P}\left( \frac{4}{n} \int_{R_1}^{\infty} \|(s\mathbf{A}\mathbf{A}^T + \rho I)^{-1}\mathbf{A}\|_{op}^2 \|\mathbf{g}(s)\|_2^2 ds > \frac{\delta}{2} \right) \le \mathbb{P}(\mathcal{T}_g^c) + \mathbb{P}\left( 4C_g \int_{R_1}^{\infty} \|(s\mathbf{A}\mathbf{A}^T + \rho I)^{-1}\mathbf{A}\|_{op}^2 ds > \frac{\delta}{2} \right)$$

$$(20)$$

Thus we need to focus on the operator norm of each term in 20. After diagonalizing and using the fact that for any two rectangular matrices $\mathbf{X}, \mathbf{Y}$ with matching dimensions, $spec(\mathbf{X}\mathbf{Y}) = spec(\mathbf{Y}\mathbf{X})$

$$\|(\lambda \mathbf{A}\mathbf{A}^T + \rho I)^{-1}\|_{op}^2 = \max_{1 \le i \le n} \frac{1}{(\lambda s_i(\mathbf{A}\mathbf{A}^T) + \rho)^2} = \frac{1}{(\lambda s_{\min}(\mathbf{A}\mathbf{A}^T) + \rho)^2}$$

$$\|(\lambda \mathbf{A}\mathbf{A}^T + \rho I)^{-1}\mathbf{A}\|_{op}^2 = \max_{1 \le i \le n} \frac{s_i(\mathbf{A}\mathbf{A}^T)}{(\lambda s_i(\mathbf{A}\mathbf{A}^T) + \rho)^2}$$

Now by the event $\mathcal{T}_A$, $s_{\min}(\mathbf{A}\mathbf{A}^T) \ge C_A$, then for a large enough $R_1$ such that $\frac{\rho}{R_1} \le C_A$, then for every $\lambda \ge R_1$

$$\|(\lambda \mathbf{A}\mathbf{A}^T + \rho I)^{-1}\|_{op}^2 \le \frac{1}{(\lambda C_A + \rho)^2}$$

$$\|(\lambda \mathbf{A}\mathbf{A}^T + \rho I)^{-1}\mathbf{A}\|_{op}^2 \le \frac{C_A}{(\lambda C_A + \rho)^2}$$

Bounding the integrals implies

$$\int_{R_1}^{\infty} \|(s\mathbf{A}\mathbf{A}^T + \rho I)^{-1}\|_{op}^2 ds \le \frac{1}{C_A} \frac{1}{C_A R_1 + \rho}$$

$$\int_{R_1}^{\infty} \|(s\mathbf{A}\mathbf{A}^T + \rho I)^{-1}\mathbf{A}\|_{op}^2 ds \le \frac{1}{C_A R_1 + \rho}$$

Summarizing,

$$\mathbb{P}(\sup_{\lambda > 0} \Phi_\lambda^n - \sup_{0 < \lambda < R_1} \Phi_\lambda^n > \delta) \le \mathbb{P}\left( \frac{4\rho^2 C_y}{C_A(C_A R_1 + \rho)} > \frac{\delta}{2} \right) + \mathbb{P}\left( \frac{C_g}{C_A R_1 + \rho} > \frac{\delta}{2} \right) + \mathbb{P}(\mathcal{T}_y^c) + \mathbb{P}(\mathcal{T}_g^c) + 2\mathbb{P}(\mathcal{T}_A^c)$$

$$(21)$$

Now we choose $R_1$ to be large enough such that the first two terms vanish. Therefore there exists an $N_2 \in \mathbb{N}$ such that for every $n \ge N_2$, $\mathbb{P}(\mathcal{T}_y^c) + \mathbb{P}(\mathcal{T}_g^c) + 2\mathbb{P}(\mathcal{T}_A^c) \le \tilde{\delta}_2$. Thus for such $R_1$ and $N_2$, we have

$$\mathbb{P}(\sup_{\lambda > 0} \Phi_\lambda^n - \sup_{0 < \lambda < R_1} \Phi_\lambda^n > \delta) \le \tilde{\delta}_2$$

Now we can let $R_2 > 0$ be chosen in such a way that:

$$\sup_{\lambda > 0} c_\lambda - \sup_{0 < \lambda < R_2} c_\lambda < \tilde{\delta}$$

Take the maximum of $R_1$ and $R_2$ as $R$. By triangle inequality we observe

$$\mathbb{P}(|\sup_{\lambda>0}\Phi_\lambda^n - \sup_{\lambda>0} c_\lambda| > \delta) \le \mathbb{P}(|\sup_{0<\lambda<R}\Phi_\lambda^n - \sup_{\lambda>0} c_\lambda| > \delta) + \mathbb{P}(\sup_{0<\lambda}\Phi_\lambda^n - \sup_{0<\lambda<R}\Phi_\lambda^n > \delta)$$

$$\le \mathbb{P}(|\sup_{0<\lambda<R}\Phi_\lambda^n - \sup_{0<\lambda<R} c_\lambda| > \delta) + \mathbb{P}(\sup_{\lambda>0} c_\lambda - \sup_{0<\lambda<R} c_\lambda > \delta) + \mathbb{P}(\sup_{0<\lambda}\Phi_\lambda^n - \sup_{0<\lambda<R}\Phi_\lambda^n > \delta)$$

$$\le \mathbb{P}(|\sup_{0<\lambda<R}\Phi_\lambda^n - \sup_{0<\lambda<R} c_\lambda| > \delta) + 0 + \tilde{\delta}_1$$

$$\le \tilde{\delta}_1 + \mathbb{P}(\sup_{0<\lambda<R} |\Phi_\lambda^n - c_\lambda| > \delta)$$

$$\le \tilde{\delta}_1 + \mathbb{P}(\sup_{0\le\lambda\le R} |\Phi_\lambda^n - c_\lambda| > \delta)$$

First we prove $c_\lambda$ is concave in $\lambda$. Then by appealing to the Convexity Lemma (lemma 7.75 in Liese and Miescke [2008]), the result follows. Note that $\Phi_\lambda^n$ is concave in $\lambda$ as it is the pointwise-minimum of a concave function in $\lambda$ everywhere. Now to prove the concavity of $c_\lambda$, for a given pair of $\lambda_1, \lambda_2 \in [0, R]$ and $0 < \theta < 1$, we prove the deterministic event $\{c_{\theta\lambda_1+(1-\theta)\lambda_2} - \theta c_{\lambda_1} - (1-\theta)c_{\lambda_2} < 0\}$ has probability zero. First note that

$$\Phi_{\theta\lambda_1+(1-\theta)\lambda_2}^n - \theta\Phi_{\lambda_1}^n - (1-\theta)\Phi_{\lambda_2}^n \ge 0$$

Thus by union bound for $\epsilon > 0$

$$\mathbb{P}\left(\theta c_{\lambda_1} + (1-\theta)c_{\lambda_2} - c_{\theta\lambda_1+(1-\theta)\lambda_2} > \epsilon\right)$$

$$\le \mathbb{P}\left(\theta(c_{\lambda_1} - \Phi_{\lambda_1}^n) + (1-\theta)(c_{\lambda_2} - \Phi_{\lambda_2}^n) + \Phi_{\theta\lambda_1+(1-\theta)\lambda_2}^n - c_{\theta\lambda_1+(1-\theta)\lambda_2} > \epsilon\right)$$

$$\le \mathbb{P}\left(\theta|c_{\lambda_1} - \Phi_{\lambda_1}^n| + (1-\theta)|c_{\lambda_2} - \Phi_{\lambda_2}^n| + |\Phi_{\theta\lambda_1+(1-\theta)\lambda_2}^n - c_{\theta\lambda_1+(1-\theta)\lambda_2}| > \epsilon\right)$$

$$\le \mathbb{P}\left(\theta|c_{\lambda_1} - \Phi_{\lambda_1}^n| > \frac{\epsilon}{3}\right) + \mathbb{P}\left((1-\theta)|c_{\lambda_2} - \Phi_{\lambda_2}^n| > \frac{\epsilon}{3}\right) + \mathbb{P}\left(|\Phi_{\theta\lambda_1+(1-\theta)\lambda_2}^n - c_{\theta\lambda_1+(1-\theta)\lambda_2}| > \frac{\epsilon}{3}\right)$$

Now by taking $n$ to be large enough, the RHS can be made arbitrarily small for every $\epsilon$ and $\theta$, thus $c_\lambda$ is concave. Furthermore, since $[0, R]$ is a compact set, we can appeal to the convexity lemma 7.75 in Liese and Miescke [2008] and the first claim follows. For the second claim, note that so far we have proved

$$\Phi_\lambda(A) \xrightarrow{\mathbb{P}} c_\lambda \implies \sup_{\lambda>0}\Phi_\lambda(A) \xrightarrow{\mathbb{P}} \sup_{\lambda>0} c_\lambda$$

Then since $|\Phi_\lambda(B) - \Phi_\lambda(A)| \xrightarrow{\mathbb{P}} 0$ thus $\Phi_\lambda(B) \xrightarrow{\mathbb{P}} c_\lambda$, repeating the same argument this time for $\Phi_\lambda(B)$ yields

$$\sup_{\lambda>0}\Phi_\lambda(B) \xrightarrow{\mathbb{P}} \sup_{\lambda>0} c_\lambda$$

$\square$

### B.2.1  Sequence of Regularizers

We prove that by constraining ourselves to a large enough compact set $\mathcal{S}_\mathbf{w}$, if $f_m \to f$ uniformly, then the universality results also hold. Note for the $\Phi_\lambda$ case, we have that by setting $\mathbf{w} = \mathbf{0}$ in the optimization, we have $\|\mathbf{w}_{\Phi_\lambda(\mathbf{A})}\|_2 \le C_w\sqrt{n}$ for some $C_w > 0$ with high probability, thus we can instead set $\mathcal{S}_\mathbf{w} := C_w\sqrt{n}$ and consider the following equivalent constrained optimization problem

$$\tilde{\Phi}_{\lambda,\epsilon}^m(\mathbf{A}) := \frac{1}{n}\min_{\mathbf{w}\in\mathcal{S}_\mathbf{w}}\frac{\lambda}{2}\|\mathbf{A}\mathbf{w} - \mathbf{y}\|_2^2 + f_m(\mathbf{w}) + \epsilon\psi(\mathbf{w})$$

Now consider a sequence of regular functions $f_m$, converging uniformly to $f$ on $\mathcal{S}_\mathbf{w}$. Take $m$ large enough such that $\|f - f_m\|_\infty \le n\delta$ for any $n$ which implies $|\tilde{\Phi}_{\lambda,\epsilon}^m(\mathbf{A}) - \tilde{\Phi}_{\lambda,\epsilon}(\mathbf{A})| \le \delta$. Thus for any $t > 0$

$$\mathbb{P}\left(\left|\tilde{\Phi}_{\lambda,\epsilon}(\mathbf{A}) - c_{\lambda,\epsilon}\right| > t\right) \le \mathbb{P}\left(\left|\tilde{\Phi}_{\lambda,\epsilon}(\mathbf{A}) - \tilde{\Phi}_{\lambda,\epsilon}^m(\mathbf{A})\right| > \frac{t}{2}\right) + \mathbb{P}\left(\left|\tilde{\Phi}_{\lambda,\epsilon}^m(\mathbf{A}) - c_{\lambda,\epsilon}^m\right| > \frac{t}{2}\right) \quad (22)$$

If $\tilde{\Phi}_{\lambda,\epsilon}^m(\mathbf{A}) \xrightarrow{\mathbb{P}} c_{\lambda,\epsilon}^m$, then by (22), we have $\tilde{\Phi}_{\lambda,\epsilon}(\mathbf{A}) \xrightarrow{\mathbb{P}} c_{\lambda,\epsilon}$. A similar argument holds for

$$\tilde{\Phi}_\epsilon^m(\mathbf{A}) := \frac{1}{n}\min_{\mathbf{A}\mathbf{w}=\mathbf{y},\mathbf{w}\in\mathcal{S}_\mathbf{w}} f_m(\mathbf{w}) + \epsilon\psi(\mathbf{w})$$

Furthermore, subsequent results also hold for $\psi(\mathbf{w}_{\Phi_\lambda^m(\mathbf{A})})$ and $\psi(\mathbf{w}_{\Phi_\lambda(\mathbf{A})})$, $\psi(\mathbf{w}_{\Phi^m(\mathbf{A})})$ and $\psi(\mathbf{w}_{\Phi(\mathbf{A})})$

## B.3 Proof of Part 2 of Theorem 1

### B.3.1 Proof of Part 2 of Theorem 1 for a regular test function when $\Psi = \Phi_\lambda$

Now we prove the second part of Theorem 1. Since we assumed $\Phi_{\lambda,0}(\mathbf{A})$ converges to some $c_{\lambda,0}$ in probability, then for a small enough $\epsilon$, we also have $\Phi_{\lambda,\epsilon}(\mathbf{A}) \xrightarrow{\mathbb{P}} c_{\lambda,\epsilon}$. Therefore we may write by an application of triangle inequality and the union bound

$$\mathbb{P}\left(\left|\frac{\Phi_{\lambda,\epsilon}(\mathbf{A}) - \Phi_{\lambda,0}(\mathbf{A})}{\epsilon} - \frac{c_{\lambda,\epsilon} - c_{\lambda,0}}{\epsilon}\right| > t\right) \leq \mathbb{P}\left(\left|\Phi_{\lambda,\epsilon}(\mathbf{A}) - c_{\lambda,\epsilon}\right| > \frac{\epsilon t}{2}\right) + \mathbb{P}\left(\left|\Phi_{\lambda,0}(\mathbf{A}) - c_{\lambda,0}\right| > \frac{\epsilon t}{2}\right)$$

$$(23)$$

For every $\epsilon, t > 0$, the RHS of (23) goes to zero. Similarly we have $\mathbb{P}\left(\left|\frac{\Phi_{\lambda,0}(\mathbf{A}) - \Phi_{\lambda,-\epsilon}(\mathbf{A})}{\epsilon} - \frac{c_{\lambda,0} - c_{\lambda,-\epsilon}}{\epsilon}\right| > t\right) \to 0.$

Now we observe that by triangle inequality

$$\mathbb{P}\left(\left|\psi(\mathbf{w}_{\Phi_\lambda(\mathbf{A})}) - \frac{dc_{\lambda,\epsilon}}{d\epsilon}\Big|_{\epsilon=0}\right| > t\right) \leq \mathbb{P}\left(\psi(\mathbf{w}_{\Phi_\lambda(\mathbf{A})}) > \frac{dc_{\lambda,\epsilon}}{d\epsilon}\Big|_{\epsilon=0} + t\right) + \mathbb{P}\left(\psi(\mathbf{w}_{\Phi_\lambda(\mathbf{A})}) < \frac{dc_{\lambda,\epsilon}}{d\epsilon}\Big|_{\epsilon=0} - t\right)$$

We take $\hat{\epsilon}$ to be small enough such that

$$\left|\frac{dc_{\lambda,\epsilon}}{d\epsilon}\Big|_{\epsilon=0} - \frac{c_{\lambda,\hat{\epsilon}} - c_{\lambda,0}}{\hat{\epsilon}}\right| < \frac{t}{2}, \quad \left|\frac{dc_{\lambda,\epsilon}}{d\epsilon}\Big|_{\epsilon=0} - \frac{c_{\lambda,0} - c_{\lambda,-\hat{\epsilon}}}{\hat{\epsilon}}\right| < \frac{t}{2}$$

This implies

$$\mathbb{P}\left(\left|\psi(\mathbf{w}_{\Phi_\lambda(\mathbf{A})}) - \frac{dc_{\lambda,\epsilon}}{d\epsilon}\Big|_{\epsilon=0}\right| > t\right) \leq \mathbb{P}\left(\psi(\mathbf{w}_{\Phi_\lambda(\mathbf{A})}) > \frac{c_{\lambda,0} - c_{\lambda,-\hat{\epsilon}}}{\hat{\epsilon}} + \frac{t}{2}\right) + \mathbb{P}\left(\psi(\mathbf{w}_{\Phi_\lambda(\mathbf{A})}) < \frac{c_{\lambda,\hat{\epsilon}} - c_{\lambda,0}}{\hat{\epsilon}} - \frac{t}{2}\right)$$

And by Danskin's theorem, $\Phi_{\lambda,\epsilon}(\mathbf{A})$ is minimum of a concave function $\epsilon$, therefore it is also concave and it is differentiable with respect to $\epsilon$, we observe that $\psi(w_{\Phi_\lambda(\mathbf{A})}) = \frac{d\Phi_{\lambda,\epsilon}(\mathbf{A})}{d\epsilon}\Big|_{\epsilon=0}$. Moreover, by concavity over $\hat{\epsilon}$:

$$\frac{\Phi_{\lambda,\hat{\epsilon}}(\mathbf{A}) - \Phi_{\lambda,0}(\mathbf{A})}{\hat{\epsilon}} \leq \psi(w_{\Phi_\lambda(\mathbf{A})}) \leq \frac{\Phi_{\lambda,0}(\mathbf{A}) - \Phi_{\lambda,-\hat{\epsilon}}(\mathbf{A})}{\hat{\epsilon}}$$

Whence

$$\mathbb{P}\left(\left|\psi(\mathbf{w}_{\Phi_\lambda(\mathbf{A})}) - \frac{dc_{\lambda,\epsilon}}{d\epsilon}\Big|_{\epsilon=0}\right| > t\right) \leq \mathbb{P}\left(\frac{\Phi_{\lambda,0}(\mathbf{A}) - \Phi_{\lambda,-\hat{\epsilon}}(\mathbf{A})}{\hat{\epsilon}} > \frac{c_{\lambda,0} - c_{\lambda,-\hat{\epsilon}}}{\hat{\epsilon}} + \frac{t}{2}\right)$$

$$+ \mathbb{P}\left(\frac{\Phi_{\lambda,\hat{\epsilon}}(\mathbf{A}) - \Phi_{\lambda,0}(\mathbf{A})}{\hat{\epsilon}} < \frac{c_{\lambda,\hat{\epsilon}} - c_{\lambda,0}}{\hat{\epsilon}} - \frac{t}{2}\right)$$

$$\leq \mathbb{P}\left(\left|\frac{\Phi_{\lambda,0}(\mathbf{A}) - \Phi_{\lambda,-\hat{\epsilon}}(\mathbf{A})}{\hat{\epsilon}} - \frac{c_{\lambda,0} - c_{\lambda,-\hat{\epsilon}}}{\hat{\epsilon}}\right| > \frac{t}{2}\right)$$

$$+ \mathbb{P}\left(\left|\frac{\Phi_{\lambda,\hat{\epsilon}}(\mathbf{A}) - \Phi_{\lambda,0}(\mathbf{A})}{\hat{\epsilon}} - \frac{c_{\lambda,\hat{\epsilon}} - c_{\lambda,0}}{\hat{\epsilon}}\right| > \frac{t}{2}\right)$$

By (23), the RHS goes to zero, thus $\psi(\mathbf{w}_{\Phi_\lambda(\mathbf{A})}) \xrightarrow{\mathbb{P}} \frac{dc_{\lambda,\epsilon}}{d\epsilon}\Big|_{\epsilon=0}$. Furthermore, we have

$$\mathbb{P}\left(\left|\psi(\mathbf{w}_{\Phi_\lambda(\mathbf{B})}) - \frac{dc_{\lambda,\epsilon}}{d\epsilon}\Big|_{\epsilon=0}\right| > t\right) \leq \mathbb{P}\left(\left|\frac{\Phi_{\lambda,0}(\mathbf{B}) - \Phi_{\lambda,-\hat{\epsilon}}(\mathbf{B})}{\hat{\epsilon}} - \frac{c_{\lambda,0} - c_{\lambda,-\hat{\epsilon}}}{\hat{\epsilon}}\right| > \frac{t}{2}\right)$$

$$+ \mathbb{P}\left(\left|\frac{\Phi_{\lambda,\hat{\epsilon}}(\mathbf{B}) - \Phi_{\lambda,0}(\mathbf{B})}{\hat{\epsilon}} - \frac{c_{\lambda,\hat{\epsilon}} - c_{\lambda,0}}{\hat{\epsilon}}\right| > \frac{t}{2}\right)$$

$$\leq \mathbb{P}\left(\left|\Phi_{\lambda,\epsilon}(\mathbf{B}) - c_{\lambda,\epsilon}\right| > \frac{\epsilon t}{2}\right) + \mathbb{P}\left(\left|\Phi_{\lambda,-\epsilon}(\mathbf{B}) - c_{\lambda,-\epsilon}\right| > \frac{\epsilon t}{2}\right)$$

$$+ 2\mathbb{P}\left(\left|\Phi_{\lambda,0}(\mathbf{B}) - c_{\lambda,0}\right| > \frac{\epsilon t}{2}\right)$$

$$(24)$$

By the result of the first part, we know that the RHS (24) goes to zero as $n \to \infty$. Hence, $\psi(\mathbf{w}_{\Phi_\lambda(\mathbf{B})}) \xrightarrow{\mathbb{P}} \left. \frac{dc_{\lambda,\epsilon}}{d\epsilon} \right|_{\epsilon=0}$.

Moreover, if $\psi$ is bounded, by definition of convergence in distribution, $|\mathbb{E}_\mathbf{A}\psi(\mathbf{w}_{\Phi_\lambda(\mathbf{A})}) - \mathbb{E}_\mathbf{B}\psi(\mathbf{w}_{\Phi_\lambda(\mathbf{B})})| \to 0$.

### B.3.2  Proof of Part 2 of Theorem 1 for a regular test function for $\Psi = \Phi$

Defining

$$\tilde{\Phi}_\epsilon(\mathbf{A}) := \frac{1}{n} \min_{\mathbf{A}\mathbf{w}=\mathbf{y}, \mathbf{w} \in \mathcal{S}_\mathbf{w}} f(\mathbf{w}) + \epsilon\psi(\mathbf{w})$$

Since $sup_{\lambda>0}\tilde{\Phi}_{\lambda,\epsilon}(\mathbf{A}) \xrightarrow{\mathbb{P}} sup_{\lambda>0}c_{\lambda,\epsilon}$, or equivalently $\tilde{\Phi}_\epsilon(\mathbf{A}) \xrightarrow{\mathbb{P}} c_\epsilon$, we can use the same argument from previous section and Danskin's theorem, to observe that $\psi(\mathbf{w}_{\tilde{\Phi}_\lambda(\mathbf{A})}) \xrightarrow{\mathbb{P}} \left. \frac{dc_\epsilon}{d\epsilon} \right|_{\epsilon=0}$ and $\psi(\mathbf{w}_{\tilde{\Phi}_\lambda(\mathbf{B})}) \xrightarrow{\mathbb{P}} \left. \frac{dc_\epsilon}{d\epsilon} \right|_{\epsilon=0}$.

### B.3.3  Sequences of regular test functions

By a similar approach, it follows that for a sequence of regular functions $\psi_m$ converging uniformly to $\psi$, one also has $\psi(\mathbf{w}_{\Phi_\lambda}) \xrightarrow{\mathbb{P}} \left. \frac{dc_{\lambda,\epsilon}}{d\epsilon} \right|_{\epsilon=0} := \lim_{m\to\infty} \left. \frac{dc^m_\epsilon}{d\epsilon} \right|_{\epsilon=0}$. Indeed,

$$\mathbb{P}\left( \left| \psi(\mathbf{w}_{\Phi_\lambda(\mathbf{A})}) - \left.\frac{dc_{\lambda,\epsilon}}{d\epsilon}\right|_{\epsilon=0} \right| > t \right) \le \mathbb{P}\left( \left| \psi(\mathbf{w}_{\Phi_\lambda(\mathbf{A})}) - \psi_m(\mathbf{w}_{\Phi_\lambda(\mathbf{A})}) \right| > \frac{t}{2} \right) + \mathbb{P}\left( \left| \psi_m(\mathbf{w}_{\Phi_\lambda(\mathbf{A})}) - \left.\frac{dc_{\lambda,\epsilon}}{d\epsilon}\right|_{\epsilon=0} \right| > \frac{t}{2} \right)$$

Since we have uniform convergence, for $\frac{t}{2}$, there exists an $m_0$ such that for all $m \ge m_0$ and all $x \in \mathbb{R}^d$, we have $|\psi_m(x) - \psi(x)| < \frac{t}{2}$. Thus for such $m$'s, $\mathbb{P}\left( \left| \psi(\mathbf{w}_{\Phi_\lambda(\mathbf{A})}) - \psi_m(\mathbf{w}_{\Phi_\lambda(\mathbf{A})}) \right| > \frac{t}{2} \right) = 0$. Then by triangle inequality

$$\mathbb{P}\left( \left| \psi(\mathbf{w}_{\Phi_\lambda(\mathbf{A})}) - \left.\frac{dc_{\lambda,\epsilon}}{d\epsilon}\right|_{\epsilon=0} \right| > t \right) \le \mathbb{P}\left( \left| \psi_m(\mathbf{w}_{\Phi_\lambda(\mathbf{A})}) - \left.\frac{dc^m_{\lambda,\epsilon}}{d\epsilon}\right|_{\epsilon=0} \right| > \frac{t}{4} \right) + \mathbb{P}\left( \left| \left.\frac{dc_{\lambda,\epsilon}}{d\epsilon}\right|_{\epsilon=0} - \left.\frac{dc^m_{\lambda,\epsilon}}{d\epsilon}\right|_{\epsilon=0} \right| > \frac{t}{4} \right)$$

By assumption $\mathbb{P}\left( \left| \psi_m(\mathbf{w}_{\Phi_\lambda(\mathbf{A})}) - \left.\frac{dc^m_{\lambda,\epsilon}}{d\epsilon}\right|_{\epsilon=0} \right| > \frac{t}{4} \right) \to 0$. And by the definition of $\left.\frac{dc_{\lambda,\epsilon}}{d\epsilon}\right|_{\epsilon=0}$ the claim follows.

## C  Proof of Corollary 1

We will utilize the result of the following Theorem.

**Theorem 4.** *(Bobkov [2003]) For an isotropic random vector $\mathbf{X} \in \mathbb{R}^d$, and an isotropic gaussian $\mathbf{g} \in \mathbb{R}^d$ if $\mathbb{P}(|\frac{\|\mathbf{X}\|_2}{\sqrt{2}} - 1| \ge \epsilon_d) \le \epsilon_d$, then for all $\delta > 0$*

$$\nu\left( \sup_{t\in\mathbb{R}} \left| \mathbb{P}(\mathbf{X}^T\boldsymbol{\theta} \le t) - \mathbb{P}(\mathbf{g}^T\boldsymbol{\theta} \le t) \right| \ge 4\epsilon_d + \delta \right) \le 4d^{3/8}e^{-cd\delta^4}$$

*Where $\boldsymbol{\theta} \sim \nu$ the uniform measure on the unit sphere $\mathbb{S}^{d-1}$*

Essentially, through CGMT, we observe that when we use a quadratic regularizer or run SGD, the solution, $\mathbf{w}_{\Psi(\mathbf{G})}$ converges weakly in distribution to a Gaussian vector, $\mathbf{g}_{AO}$. Whence we conclude that $\mathbf{w}_{\Psi(\mathbf{G})}$ behaves the same as a generic vector. Therefore, by choosing $\mathbf{X} := \mathbf{a} - \mathbb{E}\mathbf{a}$, a row of our data matrix $\mathbf{A}$, Theorem 4 enables us to prove the universality of the classification error. Note that in general $\mathbb{E}\mathbf{w}_{\Psi(\mathbf{G})} \ne \mathbf{0}$, but by Assumptions 4, the means of the classes are generic which allows us to employ Theorem 4 to conclude the proof.

## D    Lemmata for the proof of Theorem 1

**Proposition 2.** *If for any Lipschitz function $g : \mathbb{R} \to \mathbb{R}$ and $t_1 > 0$, $\mathbb{P}\Big(|g(\Phi_{\lambda,\epsilon}(\mathbf{B})) - c| > t_1\Big) \to 0$ and for every twice differentiable $\tilde{g} : \mathbb{R} \to \mathbb{R}$ with bounded second derivative, we have that $\lim_{n\to\infty}\Big|\mathbb{E}_{\mathbf{A},\mathbf{y}}\tilde{g}(\Phi_{\lambda,\epsilon}(\mathbf{B})) - \mathbb{E}_{\mathbf{B},\mathbf{y}}\tilde{g}(\Phi_{\lambda,\epsilon}(\mathbf{A}))\Big| \to 0$ then for any $t_2 > t_1$.*

$$\lim_{n\to\infty} \mathbb{P}\Big(|g(\Phi_{\lambda,\epsilon}(\mathbf{A})) - c| > t_2\Big) = 0$$

*Proof.* First note that $\mathbb{P}(|\Phi_{\lambda,\epsilon}(\mathbf{B}) - c| > t) \to 0$, implies $\mathbb{P}(|g(\Phi_{\lambda,\epsilon}(\mathbf{B})) - g(c)| > Lt) \to 0$. Thus it suffices to verify $\mathbb{P}(|\Phi_{\lambda,\epsilon}(\mathbf{B}) - c| > t) \to 0$. This follows similar to Panahi and Hassibi [2017]. Let $\xi(x) := 2\mathbb{1}_{(-\infty,-t_2]\cup[t_2,\infty)}(x) + \left(\frac{t_1+t_2}{2} - (|x| - t_2)^2\right)\mathbb{1}_{(-t_2,-\frac{t_1+t_2}{2}]\cup[\frac{t_1+t_2}{2},t_2)}(x) + (|x| - t_1)^2\mathbb{1}_{(-\frac{t_1+t_2}{2},-t_1]\cup[t_1,\frac{t_1+t_2}{2})}(x)$. Then we have

$$\mathbb{P}\Big(|\Phi_{\lambda,\epsilon}(\mathbf{A}) - c| > t_2\Big) = \mathbb{P}\Big(\xi(\Phi_{\lambda,\epsilon}(\mathbf{A}) - c) > 2\Big)$$

Then by Markov's and triangle inequality

$$\mathbb{P}\Big(|\Phi_{\lambda,\epsilon}(\mathbf{A}) - c| > t_2\Big) \le \frac{1}{2}\mathbb{E}_{\mathbf{A},\mathbf{y}}\xi\big(\Phi_{\lambda,\epsilon}(\mathbf{A}) - c\big)$$

$$\le \frac{1}{2}\Big|\mathbb{E}_{\mathbf{A},\mathbf{y}}\xi\big(\Phi_{\lambda,\epsilon}(\mathbf{A}) - c\big) - \mathbb{E}_{\mathbf{B},\mathbf{y}}\xi\big(\Phi_{\lambda,\epsilon}(\mathbf{B}) - c\big)\Big| + \frac{1}{2}\mathbb{E}_{\mathbf{B},\mathbf{y}}\xi\big(\Phi_{\lambda,\epsilon}(\mathbf{B}) - c\big)$$

Note that $\xi(x) \le 2\mathbb{1}_{(-\infty,-t_1]\cup[t_1,\infty)}$. Thus $\frac{1}{2}\mathbb{E}_{\mathbf{B},\mathbf{y}}\xi\big(\Phi_{\lambda,\epsilon}(\mathbf{B}) - c\big) \le \mathbb{P}\Big(|\Phi_{\lambda,\epsilon}(\mathbf{B}) - c| > t_1\Big)$. Moreover, we know that $\xi(.)$ has bounded second derivative derivatives almost everywhere and by assumption on $g'$, we would have

$$\mathbb{P}\Big(|\Phi_{\lambda,\epsilon}(\mathbf{A}) - c| > t_2\Big) \le \frac{1}{2}\Big|\mathbb{E}_{\mathbf{A},\mathbf{y}}\xi\big(\Phi_{\lambda,\epsilon}(\mathbf{A}) - c\big) - \mathbb{E}_{\mathbf{B},\mathbf{y}}\xi\big(\Phi_{\lambda,\epsilon}(\mathbf{B}) - c\big)\Big| + \mathbb{P}\Big(|\Phi_{\lambda,\epsilon}(\mathbf{B}) - c| > t_1\Big) \to 0$$

$\square$

**Lemma 3.** *We have the following:*

1. $\mathcal{L}(\mathbf{a}) - \Theta = \frac{\lambda}{2n}\frac{(\mathbf{a}^T\mathbf{u} - y)^2}{1 + \lambda\mathbf{a}^T\mathbf{\Omega}^{-1}\mathbf{a}}$

2. $\mathbf{w}_{\mathcal{L}} - \mathbf{u} = \lambda\frac{y - \mathbf{a}^T\mathbf{u}}{1 + \lambda\mathbf{a}^T\mathbf{\Omega}^{-1}\mathbf{a}}\mathbf{\Omega}^{-1}\mathbf{a}$

*Proof.* We know for $\Theta$, taking derivative yields

$$\lambda\tilde{\mathbf{A}}^T\tilde{\mathbf{A}}\mathbf{u} - \lambda\tilde{\mathbf{A}}^T\tilde{\mathbf{y}} + \mathbf{g} = 0$$

This implies $-\lambda\tilde{\mathbf{A}}^T\tilde{\mathbf{y}} + \mathbf{g} = -\lambda\tilde{\mathbf{A}}^T\tilde{\mathbf{A}}\mathbf{u}$.

Note that for $\mathcal{L}(\mathbf{a})$, since the objective is quadratic in $\mathbf{w}$, we solve the optimization in closed form, we have

$$\mathcal{L}(\mathbf{a}) = \frac{1}{n}\min_{\mathbf{w}}\begin{pmatrix}\mathbf{w}^T & 1\end{pmatrix}\begin{pmatrix}\frac{1}{2}(\lambda\tilde{\mathbf{A}}^T\tilde{\mathbf{A}} + \lambda\mathbf{a}\mathbf{a}^T + \mathbf{H}) & \frac{1}{2}(-\lambda\tilde{\mathbf{A}}^T\tilde{\mathbf{y}} - \lambda y\mathbf{a} + \mathbf{g} - \mathbf{H}\mathbf{u}) \\ \frac{1}{2}(-\lambda\tilde{\mathbf{A}}^T\tilde{\mathbf{y}} - \lambda y\mathbf{a} + \mathbf{g} - \mathbf{H}\mathbf{u})^T & f(\mathbf{u}) + \epsilon\psi(\mathbf{u}) - \mathbf{g}^T\mathbf{u} + \frac{1}{2}\mathbf{u}^T\mathbf{H}\mathbf{u} + \frac{\lambda}{2}(\|\tilde{\mathbf{y}}\|^2 + y^2)\end{pmatrix}\begin{pmatrix}\mathbf{w} \\ 1\end{pmatrix}$$

Let $\mathbf{\Omega} := \lambda\tilde{\mathbf{A}}^T\tilde{\mathbf{A}} + \mathbf{H}$. hence

$$\mathcal{L}(\mathbf{a}) = \frac{1}{n}\left[f(\mathbf{u}) + \epsilon\psi(\mathbf{u}) - \mathbf{g}^T\mathbf{u} + \frac{1}{2}\mathbf{u}^T\mathbf{H}\mathbf{u} + \frac{\lambda}{2}(\|\tilde{\mathbf{y}}\|^2 + y^2) - \frac{1}{2}(\mathbf{\Omega}\mathbf{u} + \lambda y\mathbf{a})^T(\mathbf{\Omega} + \lambda\mathbf{a}\mathbf{a}^T)^{-1}(\mathbf{\Omega}\mathbf{u} + \lambda y\mathbf{a})\right]$$

Using Sherman-Morrison as $\mathbf{\Omega} > 0$:

$$(\mathbf{\Omega}\mathbf{u} + \lambda y\mathbf{a})^T\Big(\mathbf{\Omega} + \lambda\mathbf{a}\mathbf{a}^T\Big)^{-1}(\mathbf{\Omega}\mathbf{u} + \lambda y\mathbf{a}) = (\mathbf{\Omega}\mathbf{u} + \lambda y\mathbf{a})^T\left(\mathbf{\Omega}^{-1} - \frac{\lambda\mathbf{\Omega}^{-1}\mathbf{a}\mathbf{a}^T\mathbf{\Omega}^{-1}}{1 + \lambda\mathbf{a}^T\mathbf{\Omega}^{-1}\mathbf{a}}\right)(\mathbf{\Omega}\mathbf{u} + \lambda y\mathbf{a}) =$$

$$= (\mathbf{\Omega}\mathbf{u} + \lambda y\mathbf{a})^T\left(\mathbf{u} - \lambda\mathbf{a}^T\mathbf{u}\frac{\mathbf{\Omega}^{-1}\mathbf{a}}{1 + \lambda\mathbf{a}^T\mathbf{\Omega}^{-1}\mathbf{a}} + \lambda y\mathbf{\Omega}^{-1}\mathbf{a} - \lambda^2 y\mathbf{a}^T\mathbf{\Omega}^{-1}\mathbf{a}\frac{\mathbf{\Omega}^{-1}\mathbf{a}}{1 + \lambda\mathbf{a}^T\mathbf{\Omega}^{-1}\mathbf{a}}\right) =$$

$$= (\mathbf{\Omega}\mathbf{u} + \lambda y\mathbf{a})^T\left(\mathbf{u} + \lambda\Big(y - \frac{\lambda y\mathbf{a}^T\mathbf{\Omega}^{-1}\mathbf{a} + \mathbf{a}^T\mathbf{u}}{1 + \lambda\mathbf{a}^T\mathbf{\Omega}^{-1}\mathbf{a}}\Big)\mathbf{\Omega}^{-1}\mathbf{a}\right) = (\mathbf{\Omega}\mathbf{u} + \lambda y\mathbf{a})^T\left(\mathbf{u} + \lambda\frac{y - \mathbf{a}^T\mathbf{u}}{1 + \lambda\mathbf{a}^T\mathbf{\Omega}^{-1}\mathbf{a}}\mathbf{\Omega}^{-1}\mathbf{a}\right)$$

From here we obtain for the solution Thus for the first claim:

$$\mathbf{w}_{\mathcal{L}} = (\mathbf{\Omega} + \lambda \mathbf{a}\mathbf{a}^T)^{-1}(\mathbf{\Omega}\mathbf{u} + \lambda y\mathbf{a}) = \mathbf{u} + \lambda \frac{y - \mathbf{a}^T\mathbf{u}}{1 + \lambda \mathbf{a}^T\mathbf{\Omega}^{-1}\mathbf{a}}\mathbf{\Omega}^{-1}\mathbf{a}$$

Hence we have for the first claim

$$\mathbf{w}_{\mathcal{L}} - \mathbf{u} = \lambda \frac{y - \mathbf{a}^T\mathbf{u}}{1 + \lambda \mathbf{a}^T\mathbf{\Omega}^{-1}\mathbf{a}}\mathbf{\Omega}^{-1}\mathbf{a}$$

Now we continue with the calculation of the objective value

$$\mathbf{u}^T\mathbf{\Omega}\mathbf{u} + \lambda \frac{y - \mathbf{a}^T\mathbf{u}}{1 + \lambda \mathbf{a}^T\mathbf{\Omega}^{-1}\mathbf{a}}\mathbf{a}^T\mathbf{u} + \lambda y\mathbf{a}^T\mathbf{u} + \lambda^2 y \frac{y - \mathbf{a}^T\mathbf{u}}{1 + \lambda \mathbf{a}^T\mathbf{\Omega}^{-1}\mathbf{a}}\mathbf{a}^T\mathbf{\Omega}^{-1}\mathbf{a} =$$

$$= \mathbf{u}^T\mathbf{\Omega}\mathbf{u}$$

$$+ \frac{\lambda}{1 + \lambda \mathbf{a}^T\mathbf{\Omega}^{-1}\mathbf{a}}\left(y\mathbf{a}^T\mathbf{u} - (\mathbf{a}^T\mathbf{u})^2 + y\mathbf{a}^T\mathbf{u} + \lambda y(\mathbf{a}^T\mathbf{u})(\mathbf{a}^T\mathbf{\Omega}^{-1}\mathbf{a}) + \lambda y^2\mathbf{a}^T\mathbf{\Omega}^{-1}\mathbf{a} - \lambda y(\mathbf{a}^T\mathbf{u})(\mathbf{a}^T\mathbf{\Omega}^{-1}\mathbf{a})\right) =$$

$$= \mathbf{u}^T\mathbf{\Omega}\mathbf{u} + \frac{\lambda}{1 + \lambda \mathbf{a}^T\mathbf{\Omega}^{-1}\mathbf{a}}(2y\mathbf{a}^T\mathbf{u} - (\mathbf{a}^T\mathbf{u})^2 + \lambda y^2\mathbf{a}^T\mathbf{\Omega}^{-1}\mathbf{a})$$

Thus

$$\mathcal{L}(\mathbf{a}) = \frac{1}{n}\left[f(\mathbf{u}) + \epsilon\psi(\mathbf{u}) - \mathbf{g}^T\mathbf{u} + \frac{1}{2}\mathbf{u}^T\mathbf{H}\mathbf{u} + \frac{\lambda}{2}(\|\tilde{\mathbf{y}}\|^2 + y^2) - \frac{1}{2}\mathbf{u}^T\mathbf{\Omega}\mathbf{u}\right.$$

$$\left. - \frac{1}{2}\frac{\lambda}{1 + \lambda \mathbf{a}^T\mathbf{\Omega}^{-1}\mathbf{a}}(2y\mathbf{a}^T\mathbf{u} - (\mathbf{a}^T\mathbf{u})^2 + \lambda y^2\mathbf{a}^T\mathbf{\Omega}^{-1}\mathbf{a})\right]$$

$$= \frac{1}{n}\left[f(\mathbf{u}) + \epsilon\psi(\mathbf{u}) - \mathbf{g}^T\mathbf{u} - \frac{\lambda}{2}\mathbf{u}^T\tilde{\mathbf{A}}^T\tilde{\mathbf{A}}\mathbf{u} + \frac{\lambda}{2}\|\tilde{\mathbf{y}}\|^2 + \frac{\lambda}{2}\frac{(\mathbf{a}^T\mathbf{u} - y)^2}{1 + \lambda \mathbf{a}^T\mathbf{\Omega}^{-1}\mathbf{a}}\right]$$

Since $g = \lambda \tilde{\mathbf{A}}^T\tilde{\mathbf{y}} - \lambda \tilde{\mathbf{A}}^T\tilde{\mathbf{A}}u$, then $\mathbf{g}^T\mathbf{u} = \lambda \tilde{\mathbf{y}}^T\tilde{\mathbf{A}}\mathbf{u} - \lambda \mathbf{u}^T\tilde{\mathbf{A}}\tilde{\mathbf{A}}u$, therefore

$$\mathcal{L}(\mathbf{a}) = \frac{1}{n}\left[f(\mathbf{u}) + \epsilon\psi(\mathbf{u}) - \lambda \tilde{\mathbf{y}}^T\tilde{\mathbf{A}}\mathbf{u} + \lambda \mathbf{u}^T\tilde{\mathbf{A}}\tilde{\mathbf{A}}\mathbf{u} - \frac{\lambda}{2}\mathbf{u}^T\tilde{\mathbf{A}}^T\tilde{\mathbf{A}}\mathbf{u} + \frac{\lambda}{2}\|\tilde{\mathbf{y}}\|^2 + \frac{\lambda}{2}\frac{(\mathbf{a}^T\mathbf{u} - y)^2}{1 + \lambda \mathbf{a}^T\mathbf{\Omega}^{-1}\mathbf{a}}\right]$$

$$= \frac{1}{n}\left[f(\mathbf{u}) + \epsilon\psi(\mathbf{u}) + \frac{\lambda}{2}\|\tilde{\mathbf{A}}\mathbf{u} - \tilde{\mathbf{y}}\|^2 + \frac{\lambda}{2}\frac{(\mathbf{a}^T\mathbf{u} - y)^2}{1 + \lambda \mathbf{a}^T\mathbf{\Omega}^{-1}\mathbf{a}}\right] = \Theta + \frac{\lambda}{2n}\frac{(\mathbf{a}^T\mathbf{u} - y)^2}{1 + \lambda \mathbf{a}^T\mathbf{\Omega}^{-1}\mathbf{a}}$$

$\square$

**Lemma 4.** *We have the following:*

1. $\mathcal{M}(\mathbf{a}) - \Theta \leq \frac{\lambda}{2n}(\mathbf{a}^T\mathbf{u} - y)^2$

2. $\ell(\mathbf{a}, \mathbf{w}) - \mathcal{L}(\mathbf{a}) \geq \frac{\rho}{2n}\|\mathbf{w} - w_{\mathcal{L}}\|_2^2$

3. $|m(\mathbf{a}, \mathbf{w}) - \ell(\mathbf{a}, \mathbf{w})| \leq \frac{C_{f+\epsilon\psi}}{n}\|\mathbf{w} - \mathbf{u}\|_3^3$

*Proof.* 1. By definition,

$$\mathcal{M}(\mathbf{a}) \leq m(\mathbf{a}, \mathbf{u}) = \Theta + \frac{\lambda}{2n}(\mathbf{a}^T\mathbf{u} - y)^2$$

Thus $\mathcal{M}(\mathbf{a}) - \Theta \leq \frac{\lambda}{2n}(\mathbf{a}^T\mathbf{u} - y)^2$

2. The idea of the proof is by strong convexity. By assumption, $f(\mathbf{w}) + \epsilon\psi(\mathbf{w})$ for small enough $\epsilon$ is $\rho$-strongly convex. This implies from defintion:

$$\ell(\mathbf{a}, \mathbf{w}) \geq \mathcal{L}(\mathbf{a}) + \frac{\rho}{2n}\|\mathbf{w} - \mathbf{w}_{\mathcal{L}}\|_2^2$$

3. By Taylor remainder theorem for some $0 < t < 1$:

$$m(\mathbf{a}, \mathbf{w}) - \ell(\mathbf{a}, \mathbf{w}) = \frac{1}{n}\left[f(\mathbf{w}) + \epsilon\psi(\mathbf{w}) - f(\mathbf{u}) - \epsilon\psi(\mathbf{u}) - g^T(\mathbf{w} - \mathbf{u}) - \frac{1}{2}(\mathbf{w} - \mathbf{u})^T\mathbf{H}(\mathbf{w} - \mathbf{u})\right]$$

$$= \frac{1}{n}\sum_{|\alpha|=3}\partial^\alpha(f + \epsilon\psi)((1 - t)\mathbf{u} + t\mathbf{w})\frac{(\mathbf{w} - \mathbf{u})^\alpha}{\alpha!}$$

Since $f$ and $\psi$ are separable, $\partial^\alpha(f + \epsilon\psi) = 0$ unless $\alpha = 3e_\alpha$ ($e_\alpha$ is the standard basis vector in $\mathbb{R}^d$). Then by assumption on the third derivative of $f + \epsilon\psi$,

$$|m(\mathbf{a}, \mathbf{w}) - \ell(\mathbf{a}, \mathbf{w})| = \frac{1}{n}\left|\sum_{|\alpha|=3}\partial^\alpha(f + \epsilon\psi)((1-t)\mathbf{u} + t\mathbf{w})\frac{(\mathbf{w} - \mathbf{u})^\alpha}{\alpha!}\right|$$

$$= \frac{1}{n}\left|\sum_{i=1}^{d}\frac{\partial^3(f + \epsilon\psi)((1-t)u_i + tw_i)}{\partial w_i^3}(\mathbf{w}_i - u_i)^3\right| \le \frac{C_{f+\epsilon\psi}}{n}\|\mathbf{w} - \mathbf{u}\|_3^3$$

$\square$

Now we are ready to prove the following key lemma.

**Lemma 5.** *We have* $|\mathcal{M}(\mathbf{a}) - \mathcal{L}(\mathbf{a})| \le \min\{\frac{\lambda}{2n}(\mathbf{a}^T\mathbf{u} - y)^2, \frac{8C_{f+\epsilon\psi}}{n}\|\mathbf{w}_\mathcal{L} - \mathbf{u}\|_3^3\}$

*Proof.* Let $\mathbf{w}_\mathcal{M} = \arg\min m(\mathbf{a}, \mathbf{w})$. Note that since $\Theta \le \mathcal{M}(\mathbf{a})$ and by Lemma 4 we have

$$\mathcal{L}(\mathbf{a}) - \mathcal{M}(\mathbf{a}) = \mathcal{L}(\mathbf{a}) - \Theta + \Theta - \mathcal{M}(\mathbf{a}) \le \frac{\lambda}{2n}\frac{(\mathbf{a}^T\mathbf{u} - y)^2}{1 + \lambda\mathbf{a}^T\Omega^{-1}\mathbf{a}}$$

On the other hand, note that $m(\mathbf{a}, \mathbf{u}) = \Theta + \frac{\lambda}{2n}(\mathbf{a}^T\mathbf{u} - y)^2$ and $\Theta \le \mathcal{L}(\mathbf{a})$

$$\mathcal{M}(\mathbf{a}) - \mathcal{L}(\mathbf{a}) = \mathcal{M}(\mathbf{a}) - m(\mathbf{a}, \mathbf{u}) + m(\mathbf{a}, \mathbf{u}) - \Theta + \Theta - \mathcal{L}(\mathbf{a}) \le \frac{\lambda}{2n}(\mathbf{a}^T\mathbf{u} - y)^2$$

Thus as $\frac{\lambda}{2n}(\mathbf{a}^T\mathbf{u} - y)^2 \ge \frac{\lambda}{2n}\frac{(\mathbf{a}^T\mathbf{u} - y)^2}{1 + \lambda\mathbf{a}^T\Omega^{-1}\mathbf{a}}$

$$|\mathcal{M}(\mathbf{a}) - \mathcal{L}(\mathbf{a})| \le \frac{\lambda}{2n}(\mathbf{a}^T\mathbf{u} - y)^2$$

But as we will see, $(\mathbf{a}^T\mathbf{u} - y)^2$ is of order $O(1)$, which will not suffice. Thus we break down $\mathcal{M}(\mathbf{a}) - \mathcal{L}(\mathbf{a})$ depending on the distance of $u$ and $\mathbf{w}_\mathcal{L}$

$$\mathcal{M}(\mathbf{a}) - \mathcal{L}(\mathbf{a}) = m(\mathbf{a}, \mathbf{w}_\mathcal{M}) - m(\mathbf{a}, \mathbf{w}_\mathcal{L}) + m(\mathbf{a}, \mathbf{w}_\mathcal{L}) - \ell(\mathbf{a}, \mathbf{w}_\mathcal{L})$$

$$\le m(\mathbf{a}, \mathbf{w}_\mathcal{L}) - \ell(\mathbf{a}, \mathbf{w}_\mathcal{L}) \le \frac{C_{f+\epsilon\psi}}{n}\|\mathbf{w}_\mathcal{L} - \mathbf{u}\|_3^3$$

For the other direction

$$\mathcal{L}(\mathbf{a}) - \mathcal{M}(\mathbf{a}) = \ell(\mathbf{a}, \mathbf{w}_\mathcal{L}) - \ell(\mathbf{a}, \mathbf{w}_\mathcal{M}) + \ell(\mathbf{a}, \mathbf{w}_\mathcal{M}) - m(\mathbf{a}, \mathbf{w}_\mathcal{M}) \le \frac{C_{f+\epsilon\psi}}{n}\|\mathbf{w}_\mathcal{M} - \mathbf{u}\|_3^3$$

Let $R := \|\mathbf{w}_\mathcal{L} - \mathbf{u}\|_3$ and consider the $\ell_3$ ball centered at $\mathbf{w}_\mathcal{L}$. Now if $\|\mathbf{w}_\mathcal{M} - \mathbf{w}_\mathcal{L}\|_3 \le R$ then $\|\mathbf{w}_\mathcal{M} - \mathbf{u}\|_3 \le 2R$ by a straightforward application of triangle inequality. Which implies $\mathcal{L}(\mathbf{a}) - \mathcal{M}(\mathbf{a}) \le \frac{8C_{f+\epsilon\psi}}{n}R^3$. Now when does $\|\mathbf{w}_\mathcal{M} - \mathbf{w}_\mathcal{L}\|_3 \le R$ hold? We show that for every point on the boundary, $\mathbf{w}$, $m(\mathbf{a}, \mathbf{w}) \ge m(\mathbf{a}, \mathbf{w}_\mathcal{L})$, this implies that $\mathbf{w}_\mathcal{M}$ is indeed inside the ball. We have

$$m(\mathbf{a}, \mathbf{w}) - m(\mathbf{a}, \mathbf{w}_\mathcal{L}) \ge \ell(\mathbf{a}, \mathbf{w}) - \frac{C_{f+\epsilon\psi}}{n}\|\mathbf{w} - \mathbf{u}\|_3^3 - \ell(\mathbf{a}, \mathbf{w}_\mathcal{L}) - \frac{C_{f+\epsilon\psi}}{n}\|\mathbf{w}_\mathcal{L} - \mathbf{u}\|_3^3$$

$$\ge \frac{\rho}{2n}\|\mathbf{w} - \mathbf{w}_\mathcal{L}\|_2^2 - \frac{C_{f+\epsilon\psi}}{n}(\|\mathbf{w} - \mathbf{u}\|_3^3 + R^3)$$

Then using Lemma 4

$$m(\mathbf{a}, \mathbf{w}) - m(\mathbf{a}, \mathbf{w}_\mathcal{L}) \ge \frac{\rho}{2n}\|\mathbf{w} - \mathbf{w}_\mathcal{L}\|_3^2 - \frac{C_{f+\epsilon\psi}}{n}(\|\mathbf{w} - \mathbf{u}\|_3^3 + R^3)$$

$$\ge \frac{\rho}{2n}R^2 - \frac{C_{f+\epsilon\psi}}{n}(8R^3 + R^3) = R^2(\frac{\rho}{2n} - \frac{9C_{f+\epsilon\psi}}{n}R)$$

Now if $R \le \frac{\rho}{18C_{f+\epsilon\psi}}$, then $m(\mathbf{a}, \mathbf{w}) - m(\mathbf{a}, \mathbf{w}_\mathcal{L}) \ge 0$. Thus all in all

$$|\mathcal{L}(\mathbf{a}) - \mathcal{M}(\mathbf{a})| \le \frac{8C_{f+\epsilon\psi}}{n}\|\mathbf{w}_\mathcal{L} - \mathbf{u}\|_3^3$$

$\square$

**Lemma 6.** *We have the following upper bounds*

1. $\|\mathbf{u}\|_2^2 = O(n)$ *with high probabilty.*

2. $\mathbb{E}_{\tilde{\mathbf{A}},\tilde{\mathbf{y}}} \boldsymbol{\mu}^T \mathbf{u} = O(1)$

*Proof.* 1. Since $f + \epsilon\psi$ is $\rho$-strongly convex, one can decompose $f(\mathbf{w}) + \epsilon\psi(\mathbf{w}) = \frac{\rho}{2}\|\mathbf{w}\|_2^2 + h(\mathbf{w})$ for some convex function $h(\cdot)$. Then

$$\frac{\lambda}{2}\|\tilde{\mathbf{A}}\mathbf{u} - \tilde{\mathbf{y}}\|_2^2 + \frac{\rho}{2}\|\mathbf{u}\|^2 + h(\mathbf{u}) \leq \frac{\lambda}{2}\|\tilde{\mathbf{y}}\|_2^2 + h(\mathbf{0}) = O(n)$$

Thus this implies $\|\mathbf{u}\|_2^2 = O(n)$.

2. In a similar fashion, we have for $\mathbb{E}_{\tilde{\mathbf{A}},\tilde{\mathbf{y}}} \boldsymbol{\mu}^T \mathbf{u}$

$$\Theta = \frac{1}{n}\left[\frac{\lambda}{2}\|\tilde{\mathbf{A}}\mathbf{u} - \tilde{\mathbf{y}}\|_2^2 + f(\mathbf{u}) + \epsilon\psi(\mathbf{u})\right] \leq \frac{1}{n}\left[\frac{\lambda}{2}\|\tilde{\mathbf{y}}\|^2 + f(0) + \epsilon\psi(0)\right]$$

Let $\tilde{\mathbf{A}}^T = \begin{pmatrix} \hat{\tilde{\mathbf{A}}}_1^T & \hat{\tilde{\mathbf{A}}}_2^T & \dots & \hat{\tilde{\mathbf{A}}}_k^T \end{pmatrix}$. Now we use the definition of $\tilde{\mathbf{A}}$ to obtain a lower bound:

$$\frac{\lambda}{2}\|\tilde{\mathbf{A}}\mathbf{u} - \tilde{\mathbf{y}}\|_2^2 + f(\mathbf{u}) + \epsilon\psi(\mathbf{u}) \geq \frac{\lambda}{2}\sum_{i=1}^{k}\|\hat{\tilde{\mathbf{A}}}_i\mathbf{u} + \mathbb{1}\boldsymbol{\mu}_i^T\mathbf{u} - \hat{\tilde{\mathbf{y}}}_i\|_2^2$$

$$\geq \frac{\lambda}{2}\|\hat{\tilde{\mathbf{A}}}_i\mathbf{u} + \mathbb{1}\boldsymbol{\mu}_i^T\mathbf{u} - \hat{\tilde{\mathbf{y}}}_i\|_2^2 \geq \frac{n_i\lambda}{2}(\boldsymbol{\mu}_i^T\mathbf{u})^2 + \lambda(\boldsymbol{\mu}_i^T\mathbf{u})\mathbb{1}^T(\hat{\tilde{\mathbf{A}}}_i\mathbf{u} - \hat{\tilde{\mathbf{y}}}_i)$$

Thus plugging in $\mathbf{w} = \mathbf{0}$ in the optimization yields an upper bound, which is a quadratic inequality in $\boldsymbol{\mu}_i^T\mathbf{u}$:

$$\frac{n_i\lambda}{2}(\boldsymbol{\mu}_i^T\mathbf{u})^2 + \lambda(\boldsymbol{\mu}_i^T\mathbf{u})\mathbb{1}^T(\hat{\tilde{\mathbf{A}}}_i\mathbf{u} - \hat{\tilde{\mathbf{y}}}_i) - \frac{\lambda}{2}\|\tilde{\mathbf{y}}\|^2 - f(0) - \epsilon\psi(0) \leq 0 \tag{25}$$

Now note that for a quadratic optimization with $b \geq 0$

$$x^2 - ax - b \leq 0 \iff \frac{a - \sqrt{a^2 + 4b}}{2} \leq x \leq \frac{a + \sqrt{a^2 + 4b}}{2}$$

Whence

$$|x| \leq \frac{|a| + \sqrt{a^2 + 4b}}{2} \leq \frac{|a| + |a| + 2\sqrt{b}}{2} = |a| + \sqrt{b}$$

We apply this result to 25,

$$|\boldsymbol{\mu}_i^T\mathbf{u}| \leq \frac{2}{n_i}|\mathbb{1}^T(\hat{\tilde{\mathbf{A}}}_i\mathbf{u} - \hat{\tilde{\mathbf{y}}}_i)| + \frac{2}{\lambda n_i}\sqrt{\frac{\lambda}{2}\|\tilde{\mathbf{y}}\|^2 + f(0) + \epsilon\psi(0)}$$

$$\leq \frac{2}{n_i}|\mathbb{1}^T\hat{\tilde{\mathbf{y}}}_i| + \frac{2}{n_i}\|\mathbb{1}^T\hat{\tilde{\mathbf{A}}}_i\|_2\|\mathbf{u}\|_2 + \frac{2}{\lambda n_i}\sqrt{\frac{\lambda}{2}\|\tilde{\mathbf{y}}\|^2 + f(0) + \epsilon\psi(0)}$$

Now taking expectation and using a combination Cauchy-Schwarz and Jensen inequalities yields

$$\mathbb{E}_{\tilde{\mathbf{A}},\tilde{\mathbf{y}}}|\boldsymbol{\mu}_i^T\mathbf{u}| \leq \frac{2}{n_i}\mathbb{E}_{\tilde{\mathbf{A}},\tilde{\mathbf{y}}}|\mathbb{1}^T\hat{\tilde{\mathbf{y}}}_i| + \frac{2}{n_i}\sqrt{\mathbb{E}_{\tilde{\mathbf{A}},\tilde{\mathbf{y}}}\|\mathbb{1}^T\hat{\tilde{\mathbf{A}}}_i\|_2^2\mathbb{E}_{\tilde{\mathbf{A}},\tilde{\mathbf{y}}}\|\mathbf{u}\|_2^2} + \frac{2}{\lambda n_i}\sqrt{\frac{\lambda}{2}\mathbb{E}_{\tilde{\mathbf{A}},\tilde{\mathbf{y}}}\|\tilde{\mathbf{y}}\|_2^2 + f(0) + \epsilon\psi(0)}$$

Now note that $\mathbb{E}\hat{\tilde{\mathbf{A}}}_i = 0$, and by the independence of the rows:

$$\mathbb{E}_{\tilde{\mathbf{A}},\tilde{\mathbf{y}}}\|\mathbb{1}^T\hat{\tilde{\mathbf{A}}}_i\|_2^2 = \sum_{j_1=1}^{d}\left(\sum_{j_2=1}^{n}(\hat{\tilde{\mathbf{A}}}_i)_{j_2 j_1}\right)^2 = \sum_{j_1=1}^{d}\sum_{j_2,j_3=1}^{n}\mathbb{E}_{\tilde{\mathbf{A}}}(\hat{\tilde{\mathbf{A}}}_i)_{j_2 j_1}(\hat{\tilde{\mathbf{A}}}_i)_{j_3 j_1} = \sum_{j_2=1}^{d}\mathbb{E}_{\tilde{\mathbf{A}}}\sum_{j_1=1}^{n}(\hat{\tilde{\mathbf{A}}}_i)_{j_2 j_1}^2$$

By assumption, for each row $\hat{\mathbf{a}}_i$ of $\hat{\tilde{\mathbf{A}}}_i$, we have that $\mathbb{E}_{\hat{\mathbf{a}}_i}\|\hat{\mathbf{a}}_i\|_2^2 = O(1)$, which implies $\mathbb{E}_{\tilde{\mathbf{A}},\tilde{\mathbf{y}}}\|\mathbb{1}^T\hat{\tilde{\mathbf{A}}}_i\|_2^2 = O(d)$. Therefore, by the previous part, since $\|\mathbf{u}\|_2^2 \leq \frac{\lambda}{\rho}\|\tilde{\mathbf{y}}\|_2^2 + \frac{2}{\rho}h(\mathbf{0})$, then $\mathbb{E}_{\tilde{\mathbf{y}}}\|\mathbf{u}\|_2^2 \leq \frac{2}{\rho}h(\mathbf{0}) + \frac{\lambda}{\rho}\mathbb{E}_{\tilde{\mathbf{y}}}\|\tilde{\mathbf{y}}\|_2^2 = O(d)$ by assumption on the $\tilde{\mathbf{y}}$. Hence it can be seen that $\mathbb{E}_{\tilde{\mathbf{A}},\tilde{\mathbf{y}}}\boldsymbol{\mu}^T\mathbf{u} = O(1)$ $\qquad\square$

Now we are ready to conclude the proof with the following lemma.

**Lemma 7.** *We have the following*

1. $\mathbb{E}_{\tilde{\mathbf{A}}, \tilde{\mathbf{y}}, \mathbf{a}, y}(\mathbf{a}^T \mathbf{u} - y)^{2q} = O(1)$ *for* $q \in [3]$

2. $\mathbb{E}_{\tilde{\mathbf{A}}, \tilde{\mathbf{y}}, \mathbf{a}, y} \|\mathbf{u} - \mathbf{w}_{\mathcal{L}}\|_3^3 = O(\frac{1}{d})$

*Proof.* 1. First note that

$$\mathbb{E}_{\tilde{\mathbf{A}}, \tilde{\mathbf{y}}, \mathbf{a}, y}(\mathbf{a}^T \mathbf{u} - y)^{2q} = \mathbb{E}_{\tilde{\mathbf{A}}, \tilde{\mathbf{y}}, \mathbf{a}, y} \sum_{j=1}^{2q} \binom{2q}{j} ((\mathbf{a} - \boldsymbol{\mu} + \boldsymbol{\mu})^T \mathbf{u})^j (-y)^{2q-y}$$

Moreover,

$$\begin{aligned}
\mathbb{E}_{\tilde{\mathbf{A}}, \tilde{\mathbf{y}}, \mathbf{a}, y}(\mathbf{a}^T \mathbf{u} - y)^{2q} &\leq \sum_{j=1}^{2q} \binom{2q}{j} \mathbb{E}_{\tilde{\mathbf{A}}, \tilde{\mathbf{y}}, \mathbf{a}, y} \left[ |(\mathbf{a} - \boldsymbol{\mu} + \boldsymbol{\mu})^T \mathbf{u}|^j |y|^{2q-y} \right] \\
&\leq \sum_{j=1}^{2q} \binom{2q}{j} \mathbb{E}_{\tilde{\mathbf{A}}, \tilde{\mathbf{y}}, \mathbf{a}, y} \left[ |y|^{2q-y} 2^j \max\{|\boldsymbol{\mu}^T \mathbf{u}|, |(\mathbf{a} - \boldsymbol{\mu})^T \mathbf{u}|\}^j \right] \\
&\leq \sum_{j=1}^{2q} 2^j \binom{2q}{j} \mathbb{E}_{\tilde{\mathbf{A}}, \tilde{\mathbf{y}}, \mathbf{a}, y} \left[ |y|^{2q-y} (|\boldsymbol{\mu}^T \mathbf{u}|^j + |(\mathbf{a} - \boldsymbol{\mu})^T \mathbf{u}|^j) \right] \\
&\leq \sum_{j=1}^{2q} 2^j \binom{2q}{j} \sqrt{\mathbb{E}|y|^{4q-2y}} \left( \sqrt{\mathbb{E}|\boldsymbol{\mu}^T \mathbf{u}|^{2j}} + \sqrt{\mathbb{E}|(\mathbf{a} - \boldsymbol{\mu})^T \mathbf{u}|^{2j}} \right)
\end{aligned}$$

By assumption $\mathbb{E}|y|^{2q-y} \leq C$ for some constant $C > 0$ that is dimension-independent. Moreover, by our assumption, $\mathbb{E}|(\mathbf{a} - \boldsymbol{\mu})^T \mathbf{u}|^j \leq \tilde{C} \frac{\|\mathbf{u}\|_2^j}{d^{j/2}}$. Also, from the previous lemma, $\mathbb{E}|\boldsymbol{\mu}^T \mathbf{u}|^{2j} = O(1)$. The claim follows.

2. First note that $\|\mathbf{u} - \mathbf{w}_{\mathcal{L}}\|_3^3 = \lambda^3 \left| \frac{y - \mathbf{a}^T \mathbf{u}}{1 + \lambda \mathbf{a}^T \boldsymbol{\Omega}^{-1} \mathbf{a}} \right|^3 \|\boldsymbol{\Omega}^{-1} \mathbf{a}\|_3^3$ and since $\boldsymbol{\Omega} > 0$, $1 + \lambda \mathbf{a}^T \boldsymbol{\Omega}^{-1} \mathbf{a} > 0$ hence by Minkowski's inequality

$$\begin{aligned}
\mathbb{E}_{\tilde{\mathbf{A}}, \tilde{\mathbf{y}}, \mathbf{a}, y} \|\mathbf{u} - \mathbf{w}_{\mathcal{L}}\|_3^3 &\leq \lambda^3 \mathbb{E}_{\tilde{\mathbf{A}}, \tilde{\mathbf{y}}, \mathbf{a}, y} |\mathbf{a}^T \mathbf{u} - y|^3 \|\boldsymbol{\Omega}^{-1} \mathbf{a}\|_3^3 \\
&\leq \lambda^3 \mathbb{E}_{\tilde{\mathbf{A}}, \tilde{\mathbf{y}}, \mathbf{a}, y} |\mathbf{a}^T \mathbf{u} - y|^3 \|\boldsymbol{\Omega}^{-1}(\mathbf{a} - \boldsymbol{\mu} + \boldsymbol{\mu})\|_3^3 \\
&\leq 8\lambda^3 \mathbb{E}_{\tilde{\mathbf{A}}, \tilde{\mathbf{y}}, \mathbf{a}, y} |\mathbf{a}^T \mathbf{u} - y|^3 \max\{\|\boldsymbol{\Omega}^{-1}(\mathbf{a} - \boldsymbol{\mu})\|_3, \|\boldsymbol{\Omega}^{-1} \boldsymbol{\mu}\|_3\}^3 \\
&\leq 8\lambda^3 \mathbb{E}_{\tilde{\mathbf{A}}, \tilde{\mathbf{y}}, \mathbf{a}, y} |\mathbf{a}^T \mathbf{u} - y|^3 \|\boldsymbol{\Omega}^{-1}(\mathbf{a} - \boldsymbol{\mu})\|_3^3 + 8\lambda^3 \mathbb{E}_{\tilde{\mathbf{A}}, \tilde{\mathbf{y}}, \mathbf{a}, y} |\mathbf{a}^T \mathbf{u} - y|^3 |\boldsymbol{\Omega}^{-1} \boldsymbol{\mu}\|_3^3 \\
&\leq 8\lambda^3 \sqrt{\mathbb{E}_{\tilde{\mathbf{A}}, \tilde{\mathbf{y}}, \mathbf{a}, y}(\mathbf{a}^T \mathbf{u} - y)^6 \mathbb{E}_{\tilde{\mathbf{A}}, \tilde{\mathbf{y}}, \mathbf{a}, y} \|\boldsymbol{\Omega}^{-1}(\mathbf{a} - \boldsymbol{\mu})\|_3^6} \\
&\quad + 8\lambda^3 \sqrt{\mathbb{E}_{\tilde{\mathbf{A}}, \tilde{\mathbf{y}}, \mathbf{a}, y}(\mathbf{a}^T \mathbf{u} - y)^6 \mathbb{E}_{\tilde{\mathbf{A}}, \tilde{\mathbf{y}}, \mathbf{a}, y} \|\boldsymbol{\Omega}^{-1} \boldsymbol{\mu}\|_3^6}
\end{aligned}$$

Now since we assumed that the objective is $\rho$-strongly convex, or there exists $\rho > 0$ that $\|\boldsymbol{\Omega}^{-1}\|_{op} \leq \frac{1}{\rho}$, denoting $\boldsymbol{\omega}_j$ as the $j$th row of $\boldsymbol{\Omega}^{-1}$, we have by the inequality between Frobenius and Operator norm, $\|\boldsymbol{\omega}_k\|_2 \leq \frac{1}{\rho}$. Now by the assumptions we have

$$\mathbb{E}_{\tilde{\mathbf{A}}, \tilde{\mathbf{y}}, \mathbf{a}, y} \|\boldsymbol{\Omega}^{-1}(\mathbf{a} - \boldsymbol{\mu})\|_3^6 = \mathbb{E}_{\tilde{\mathbf{A}}, \tilde{\mathbf{y}}, \mathbf{a}, y} \left( \sum_{j=1}^{d} |(\mathbf{a} - \boldsymbol{\mu})^T \boldsymbol{\omega}_j|^3 \right)^2$$

$$= \mathbb{E}_{\tilde{\mathbf{A}}, \tilde{\mathbf{y}}} \sum_{j=1}^{d} \mathbb{E}_{\mathbf{a}, y} |(\mathbf{a} - \boldsymbol{\mu})^T \boldsymbol{\omega}_j|^6 + \sum_{j, l=1, j \neq l}^{d} \mathbb{E}|(\mathbf{a} - \boldsymbol{\mu})^T \boldsymbol{\omega}_j|^3 |(\mathbf{a} - \boldsymbol{\mu})^T \boldsymbol{\omega}_l|^3$$

$$\leq C \sum_{j=1}^{d} \frac{\|\boldsymbol{\omega}_j\|_2^6}{d^3} + C \sum_{j, l=1, j \neq l}^{d} \sqrt{\mathbb{E}_{\mathbf{a}} |(\mathbf{a} - \boldsymbol{\mu})^T \boldsymbol{\omega}_j|^6 \mathbb{E}_{\mathbf{a}} |(\mathbf{a} - \boldsymbol{\mu})^T \boldsymbol{\omega}_l|^6} \leq Cd \frac{1}{\rho^6 d^3} + C(d^2 - d) \frac{1}{\rho^6 d^3} \leq \frac{C}{\rho^6 d}$$

We also have for the other term:

$$\mathbb{E}_{\tilde{\mathbf{A}},\tilde{\mathbf{y}},\mathbf{a},y}\|\mathbf{\Omega}^{-1}\boldsymbol{\mu}\|_3^6 \leq \mathbb{E}_{\tilde{\mathbf{A}},\tilde{\mathbf{y}},\mathbf{a},y}\|\mathbf{\Omega}^{-1}\boldsymbol{\mu}\|_2^6$$

Note that $\|\mathbf{\Omega}^{-1}\boldsymbol{\mu}\|_2 \leq \|\mathbf{\Omega}^{-1/2}\|_{op}\|\mathbf{\Omega}^{-1/2}\boldsymbol{\mu}\|_2$. Thus it is sufficient to analyze $\boldsymbol{\mu}^T\mathbf{\Omega}^{-1}\boldsymbol{\mu}^T$. Let $t \in \mathbb{R}$ be an upper bound for $\boldsymbol{\mu}^T\mathbf{\Omega}^{-1}\boldsymbol{\mu}^T$, that is $\boldsymbol{\mu}^T\mathbf{\Omega}^{-1}\boldsymbol{\mu}^T \leq t$. Note that by the Schur complement property, we need to find the smallest $t > 0$ such that $\mathbf{\Omega} - \frac{1}{t}\boldsymbol{\mu}\boldsymbol{\mu}^T \succeq \mathbf{0}$ with holds with high probability. Recall that $\mathbf{\Omega} := \lambda\tilde{\mathbf{A}}^T\tilde{\mathbf{A}} + \mathbf{H}$ and by assumption $\mathbf{H} \succeq \rho\mathbf{I}$, thus it suffices to find the largest $t > 0$ that $\lambda\tilde{\mathbf{A}}^T\tilde{\mathbf{A}} + \rho\mathbf{I} - \frac{1}{t}\boldsymbol{\mu}\boldsymbol{\mu}^T \succeq \mathbf{0}$. Let $\tilde{\mathbf{A}}' := \tilde{\mathbf{A}} - \tilde{\mathbf{M}}$ be centered. This problem is equivalent to having $\|(\tilde{\mathbf{A}}' + \tilde{\mathbf{M}})\mathbf{v}\|_2^2 + \rho\|\mathbf{v}\|_2^2 - \frac{1}{t}(\boldsymbol{\mu}^T\mathbf{v})^2 \geq 0$ for every $\mathbf{v} \in \mathbb{R}^d$ .

Thus we should have

$$t \geq \frac{(\boldsymbol{\mu}^T\mathbf{v})^2}{\|(\tilde{\mathbf{A}}' + \tilde{\mathbf{M}})\mathbf{v}\|_2^2 + \rho\|\mathbf{v}\|_2^2}$$

Let $t^* := \sup_{\mathbf{v}} \frac{(\boldsymbol{\mu}^T\mathbf{v})^2}{\|(\tilde{\mathbf{A}}'+\tilde{\mathbf{M}})\mathbf{v}\|_2^2+\rho\|\mathbf{v}\|_2^2}$. We show $t^* = O(n^{-s})$ for some $s > 0$ with high probability. Let $\mathbf{v}^* := \arg\max \frac{(\boldsymbol{\mu}^T\mathbf{v})^2}{\|(\tilde{\mathbf{A}}'+\tilde{\mathbf{M}})\mathbf{v}\|_2^2+\rho\|\mathbf{v}\|_2^2}$. This implies

$$\|(\tilde{\mathbf{A}}' + \tilde{\mathbf{M}})\mathbf{v}^*\|_2^2 \leq \frac{1}{t^*}(\boldsymbol{\mu}^T\mathbf{v}^*)^2 \leq \frac{\|\boldsymbol{\mu}\|_2^2}{t^*}\|\mathbf{v}^*\|_2^2$$

On the other hand, by triangle inequality we have

$$(\boldsymbol{\mu}^T\mathbf{v}^*)^2 \leq \frac{1}{n_\ell}\|\tilde{\mathbf{M}}\mathbf{v}^*\|_2^2 \leq \frac{4}{n_\ell}\left(\|(\tilde{\mathbf{A}}' + \tilde{\mathbf{M}})\mathbf{v}^*\|_2^2 + \|\tilde{\mathbf{A}}'\mathbf{v}^*\|_2^2\right) \leq \frac{4}{n_\ell}\left(\frac{\|\boldsymbol{\mu}\|_2^2}{t^*}\|\mathbf{v}^*\|_2^2 + \|\tilde{\mathbf{A}}'\|_{op}^2\|\mathbf{v}^*\|_2^2\right)$$

. Now we have

$$t^* = \frac{(\boldsymbol{\mu}^T\mathbf{v}^*)^2}{\|(\tilde{\mathbf{A}}' + \tilde{\mathbf{M}})\mathbf{v}^*\|_2^2 + \rho\|\mathbf{v}^*\|_2^2} \leq \frac{\frac{4}{n_\ell}\left(\frac{\|\boldsymbol{\mu}\|_2^2}{t^*}\|\mathbf{v}^*\|_2^2 + \|\tilde{\mathbf{A}}'\|_{op}^2\|\mathbf{v}^*\|_2^2\right)}{\rho\|\mathbf{v}^*\|_2^2} = \frac{\frac{4\|\boldsymbol{\mu}\|_2^2}{t^*} + 4\|\tilde{\mathbf{A}}'\|_{op}^2}{\rho n_\ell}$$

We obtain a quadratic inequality in $t^*$

$$t^{*2} - \frac{4\|\tilde{\mathbf{A}}'\|_{op}^2}{\rho n_\ell}t^* - \frac{4\|\boldsymbol{\mu}\|_2^2}{\rho n_\ell} \leq 0$$

Which implies the following upper bound

$$\boldsymbol{\mu}^T\mathbf{\Omega}^{-1}\boldsymbol{\mu}^T \leq \min\{\frac{4\|\tilde{\mathbf{A}}'\|_{op}^2}{\rho n_\ell} + 4\sqrt{\frac{\|\boldsymbol{\mu}\|_2^2}{\rho n_\ell}}, \frac{1}{\rho}\}$$

According to Lemma (8) $\|\tilde{\mathbf{A}}'\|_{op} \leq n^{\frac{1}{3}+s}$ with probability $1 - C\frac{\log(d+n)}{n^s}$.Thus using the law of total expectation, we also observe that

$$\mathbb{E}_{\tilde{\mathbf{A}},\tilde{\mathbf{y}},\mathbf{a},y}\|\mathbf{\Omega}^{-1}\boldsymbol{\mu}\|_3^6 \leq \mathbb{E}_{\tilde{\mathbf{A}},\tilde{\mathbf{y}},\mathbf{a},y}\|\mathbf{\Omega}^{-1}\boldsymbol{\mu}\|_2^6 \leq \mathbb{E}_{\tilde{\mathbf{A}},\tilde{\mathbf{y}},\mathbf{a},y}(\boldsymbol{\mu}^T\mathbf{\Omega}^{-1}\boldsymbol{\mu})^3 \leq \left(\frac{4n^{\frac{2}{3}+2s}}{\rho n_\ell} + 4\sqrt{\frac{\|\boldsymbol{\mu}\|_2^2}{\rho n_\ell}}\right)^3 + C\frac{\log(d + n)}{\rho n^s}$$

For $0 < s < \frac{1}{6}$, as $\|\boldsymbol{\mu}\|_2^2 = O(1)$ by assumption, the RHS goes to zero. $\square$

**Lemma 8.** *Let $\tilde{\mathbf{A}}'$ be a centered random matrix following the assumptions. We have $\|\tilde{\mathbf{A}}'\|_{op} \leq Cn^{\frac{1}{3}+s}$ with probability at least $1 - \frac{C'\log(d+n)}{n^s}$.*

*Proof.* First, we provide an upper bound for the expectation $\mathbb{E}_{\tilde{\mathbf{A}}'}\|\tilde{\mathbf{A}}'\|_{op}$ using the matrix Bernstein inequality and the symmetrization technique. First we construct a corresponding hermitian matrix:

$$\|\tilde{\mathbf{A}}'\|_{op} = \left\|\begin{pmatrix} \mathbf{0} & \tilde{\mathbf{A}}' \\ \tilde{\mathbf{A}}'^T & \mathbf{0} \end{pmatrix}\right\|_{op} = \left\|\sum_{i=1}^n \mathbf{X}_i\right\|_{op}$$

With $\mathbf{X}_i := \begin{pmatrix} \mathbf{0} & \mathbf{E}_{ii}\tilde{\mathbf{A}}' \\ \tilde{\mathbf{A}}'^T\mathbf{E}_{ii} & \mathbf{0} \end{pmatrix}$ where $\mathbf{E}_{ii}$ is the all-zero matrix except $(i,i)$-entry where it is equal to 1.

Note that $\mathbf{I} = \sum_{i=1}^n \mathbf{E}_{ii}$ and $\mathbf{E}_{ii}\tilde{\mathbf{A}}'$ essentially picks the i'th row of $\tilde{\mathbf{A}}'$, which is $\tilde{\mathbf{a}}_i'^T$. Let $\{\epsilon_i\}_{i=1}^n$ be an iid sequence of symmetric Bernoulli random variables supported on $\{\pm 1\}$, independent of $\mathbf{X}_i$'s. Then by the symmetrization lemma Vershynin [2018], we have

$$\mathbb{E}_{\tilde{\mathbf{A}}'}\|\tilde{\mathbf{A}}'\|_{op} = \mathbb{E}_{\tilde{\mathbf{A}}'}\left\|\sum_{i=1}^n \mathbf{X}_i\right\|_{op} \le \mathbb{E}_{\tilde{\mathbf{A}}',\epsilon}\left\|\sum_{i=1}^n \epsilon_i\mathbf{X}_i\right\|_{op}$$

Now conditioning on $\tilde{\mathbf{A}}'$ or equivalently $\mathbf{X}_i$'s, since $\epsilon\mathbf{X}_i$'s are independent zero-mean symmetric matrices of size $(d+n)\times(d+n)$ with bounded operator norm, we can leverage Matrix Bernstein inequality Vershynin [2018] to obtain ($\lesssim$ means up to some constant)

$$\mathbb{E}_{\tilde{\mathbf{A}}'}\|\tilde{\mathbf{A}}'\|_{op} \lesssim \sqrt{\log(d+n)}\mathbb{E}_{\tilde{\mathbf{A}}'}\left\|\sum_{i=1}^n \mathbb{E}_{\epsilon_i}\left(\epsilon_i\mathbf{X}_i\right)^2\right\|_{op}^{1/2} + \log(d+n)\mathbb{E}_{\tilde{\mathbf{A}}'}\max_{1\le i\le n}\left\|\mathbf{X}_i\right\|_{op}$$

$$= \sqrt{\log(d+n)}\mathbb{E}_{\tilde{\mathbf{A}}'}\left\|\sum_{i=1}^n \mathbf{X}_i^2\right\|_{op}^{1/2} + \log(d+n)\mathbb{E}_{\tilde{\mathbf{A}}'}\max_{1\le i\le n}\left\|\mathbf{X}_i\right\|_{op}$$

$$= \sqrt{\log(d+n)}\mathbb{E}_{\tilde{\mathbf{A}}'}\left\|\begin{pmatrix} \sum_{i=1}^n \mathbf{E}_{ii}\tilde{\mathbf{A}}'\tilde{\mathbf{A}}'^T\mathbf{E}_{ii} & \mathbf{0} \\ \mathbf{0} & \sum_{i=1}^n \tilde{\mathbf{a}}_i'\tilde{\mathbf{a}}_i'^T \end{pmatrix}\right\|_{op}^{1/2} + \log(d+n)\mathbb{E}_{\tilde{\mathbf{A}}'}\max_{1\le i\le n}\left\|\mathbf{X}_i\right\|_{op}$$

Note that for $\left\|\mathbf{X}_i\right\|_{op}$, we have

$$\left\|\mathbf{X}_i\right\|_{op} = \left\|\mathbf{E}_{ii}\tilde{\mathbf{A}}'\right\|_{op} = \max_{\|\mathbf{v}\|_2=1}\left\|\mathbf{E}_{ii}\tilde{\mathbf{A}}'\mathbf{v}\right\|_2^2 = \max_{\|\mathbf{v}\|_2=1}(\tilde{\mathbf{a}}_i'^T\mathbf{v})^2 = \|\tilde{\mathbf{a}}_i'\|_2^2$$

Moreover, using triangle inequality

$$\left\|\begin{pmatrix} \sum_{i=1}^n \mathbf{E}_{ii}\tilde{\mathbf{A}}'\tilde{\mathbf{A}}'^T\mathbf{E}_{ii} & \mathbf{0} \\ \mathbf{0} & \sum_{i=1}^n \tilde{\mathbf{a}}_i'\tilde{\mathbf{a}}_i'^T \end{pmatrix}\right\|_{op} \le \left\|\sum_{i=1}^n \mathbf{E}_{ii}\tilde{\mathbf{A}}'\tilde{\mathbf{A}}'^T\mathbf{E}_{ii}\right\|_{op} + \left\|\sum_{i=1}^n \tilde{\mathbf{a}}_i'\tilde{\mathbf{a}}_i'^T\right\|_{op}$$

$$= \left\|\sum_{i=1}^n \|\tilde{\mathbf{a}}_i'\|_2^2\mathbf{E}_{ii}\right\|_{op} + \left\|\sum_{i=1}^n \tilde{\mathbf{a}}_i'\tilde{\mathbf{a}}_i'^T\right\|_{op}$$

$$= \max_{1\le i\le n}\|\tilde{\mathbf{a}}_i'\|_2^2 + \left\|\sum_{i=1}^n \tilde{\mathbf{a}}_i'\tilde{\mathbf{a}}_i'^T\right\|_{op}$$

Combining these results along with Jensen's inequality yields

$$\mathbb{E}_{\tilde{\mathbf{A}}'}\|\tilde{\mathbf{A}}'\|_{op} \lesssim \sqrt{\log(d+n)}\left(\mathbb{E}_{\tilde{\mathbf{A}}'}\max_{1\le i\le n}\|\tilde{\mathbf{a}}_i'\|_2^2 + \mathbb{E}_{\tilde{\mathbf{A}}'}\left\|\sum_{i=1}^n \tilde{\mathbf{a}}_i'\tilde{\mathbf{a}}_i'^T\right\|_{op}\right)^{1/2} + \log(d+n)\mathbb{E}_{\tilde{\mathbf{A}}'}\max_{1\le i\le n}\|\tilde{\mathbf{a}}_i'\|_2^2$$

Now to analyze $\mathbb{E}_{\tilde{\mathbf{A}}'}\left\|\sum_{i=1}^n \tilde{\mathbf{a}}_i'\tilde{\mathbf{a}}_i'^T\right\|_{op}$, we use the symmetrization trick again with iid Bernoulli $\{\epsilon_i'\}_{i=1}^n$ and Bernstein's similar to Vershynin's

$$\mathbb{E}_{\tilde{\mathbf{A}}'}\left\|\sum_{i=1}^n \tilde{\mathbf{a}}_i'\tilde{\mathbf{a}}_i'^T\right\|_{op} \le \mathbb{E}_{\tilde{\mathbf{A}}'}\left\|\sum_{i=1}^n \tilde{\mathbf{a}}_i'\tilde{\mathbf{a}}_i'^T - \mathbb{E}_{\tilde{\mathbf{a}}_i'}\tilde{\mathbf{a}}_i'\tilde{\mathbf{a}}_i'^T\right\|_{op} + \left\|\sum_{i=1}^n \mathbb{E}_{\tilde{\mathbf{a}}_i'}\tilde{\mathbf{a}}_i'\tilde{\mathbf{a}}_i'^T\right\|_{op}$$

$$\le \mathbb{E}_{\tilde{\mathbf{A}}',\epsilon'}\left\|\sum_{i=1}^n \epsilon_i'\tilde{\mathbf{a}}_i'\tilde{\mathbf{a}}_i'^T\right\|_{op} + \sum_{i=1}^n\left\|\mathbb{E}_{\tilde{\mathbf{a}}_i'}\tilde{\mathbf{a}}_i'\tilde{\mathbf{a}}_i'^T\right\|_{op}$$

$$\lesssim \sqrt{\log n}\mathbb{E}_{\tilde{\mathbf{A}}'}\left\|\sum_{i=1}^n \mathbb{E}_{\epsilon_i'}\left(\epsilon_i'\tilde{\mathbf{a}}_i'\tilde{\mathbf{a}}_i'^T\right)^2\right\|_{op}^{1/2} + \log n\mathbb{E}_{\tilde{\mathbf{A}}'}\max_{1\le i\le n}\left\|\tilde{\mathbf{a}}_i'\tilde{\mathbf{a}}_i'^T\right\|_{op} + \sum_{i=1}^n\left\|\mathbb{E}_{\tilde{\mathbf{a}}_i'}\tilde{\mathbf{a}}_i'\tilde{\mathbf{a}}_i'^T\right\|_{op}$$

$$= \sqrt{\log n}\mathbb{E}_{\tilde{\mathbf{A}}'}\left\|\sum_{i=1}^n \|\tilde{\mathbf{a}}_i'\|_2^2\tilde{\mathbf{a}}_i'\tilde{\mathbf{a}}_i'^T\right\|_{op}^{1/2} + \log n\mathbb{E}_{\tilde{\mathbf{A}}'}\max_{1\le i\le n}\|\tilde{\mathbf{a}}_i'\|_2^2 + \sum_{i=1}^n\left\|\mathbb{E}_{\tilde{\mathbf{a}}_i'}\tilde{\mathbf{a}}_i'\tilde{\mathbf{a}}_i'^T\right\|_{op}$$

$$\lesssim \sqrt{\log n}\mathbb{E}_{\tilde{\mathbf{A}}'}\max_{1\le i\le n}\|\tilde{\mathbf{a}}_i'\|_2\left\|\sum_{i=1}^n \tilde{\mathbf{a}}_i'\tilde{\mathbf{a}}_i'^T\right\|_{op}^{1/2} + \log n\mathbb{E}_{\tilde{\mathbf{A}}'}\max_{1\le i\le n}\|\tilde{\mathbf{a}}_i'\|_2^2 + \sum_{i=1}^n\left\|\mathbb{E}_{\tilde{\mathbf{a}}_i'}\tilde{\mathbf{a}}_i'\tilde{\mathbf{a}}_i'^T\right\|_{op}$$

Now for the first term, applying Cauchy Schwartz yields

$$\mathbb{E}_{\tilde{\mathbf{A}}'} \max_{1 \le i \le n} \|\tilde{\mathbf{a}}_i'\|_2 \left\|\sum_{i=1}^n \tilde{\mathbf{a}}_i' \tilde{\mathbf{a}}_i'^T\right\|_{op}^{1/2} \le \left(\mathbb{E}_{\tilde{\mathbf{A}}'} \max_{1 \le i \le n} \|\tilde{\mathbf{a}}_i'\|_2^2 \mathbb{E}_{\tilde{\mathbf{A}}'} \left\|\sum_{i=1}^n \tilde{\mathbf{a}}_i' \tilde{\mathbf{a}}_i'^T\right\|_{op}\right)^{1/2}$$

Moreover, for the last term we have from our assumptions

$$\left\|\mathbb{E}_{\tilde{\mathbf{a}}_i'} \tilde{\mathbf{a}}_i' \tilde{\mathbf{a}}_i'^T\right\|_{op} = \max_{\|\mathbf{v}\|_2=1} \mathbb{E}_{\tilde{\mathbf{a}}_i'}(\tilde{\mathbf{a}}_i'^T \mathbf{v})^2 \le \max_{\|\mathbf{v}\|_2=1} C\frac{\|\mathbf{v}\|_2^2}{d} = \frac{C}{d}$$

Thus we have

$$\mathbb{E}_{\tilde{\mathbf{A}}'} \left\|\sum_{i=1}^n \tilde{\mathbf{a}}_i' \tilde{\mathbf{a}}_i'^T\right\|_{op} \lesssim \sqrt{\log n}\left(\mathbb{E}_{\tilde{\mathbf{A}}'} \max_{1 \le i \le n} \|\tilde{\mathbf{a}}_i'\|_2^2 \mathbb{E}_{\tilde{\mathbf{A}}'} \left\|\sum_{i=1}^n \tilde{\mathbf{a}}_i' \tilde{\mathbf{a}}_i'^T\right\|_{op}\right)^{1/2} + \log n \mathbb{E}_{\tilde{\mathbf{A}}'} \max_{1 \le i \le n} \|\tilde{\mathbf{a}}_i'\|_2^2 + C$$

Solving for $\left(\mathbb{E}_{\tilde{\mathbf{A}}'} \left\|\sum_{i=1}^n \tilde{\mathbf{a}}_i' \tilde{\mathbf{a}}_i'^T\right\|_{op}\right)^{1/2}$ yields

$$\left(\mathbb{E}_{\tilde{\mathbf{A}}'} \left\|\sum_{i=1}^n \tilde{\mathbf{a}}_i' \tilde{\mathbf{a}}_i'^T\right\|_{op}\right)^{1/2} \lesssim \sqrt{\log n}\left(\mathbb{E}_{\tilde{\mathbf{A}}'} \max_{1 \le i \le n} \|\tilde{\mathbf{a}}_i'\|_2^2\right)^{1/2} + 2\left(\log n \mathbb{E}_{\tilde{\mathbf{A}}'} \max_{1 \le i \le n} \|\tilde{\mathbf{a}}_i'\|_2^2 + C\right)^{1/2}$$

Thus

$$\mathbb{E}_{\tilde{\mathbf{A}}'} \left\|\sum_{i=1}^n \tilde{\mathbf{a}}_i' \tilde{\mathbf{a}}_i'^T\right\|_{op} \lesssim \log n \mathbb{E}_{\tilde{\mathbf{A}}'} \max_{1 \le i \le n} \|\tilde{\mathbf{a}}_i'\|_2^2 + C$$

Summarising, we have

$$\mathbb{E}_{\tilde{\mathbf{A}}'} \|\tilde{\mathbf{A}}'\|_{op} \lesssim \sqrt{\log(d+n)}\left(\mathbb{E}_{\tilde{\mathbf{A}}'} \max_{1 \le i \le n} \|\tilde{\mathbf{a}}_i'\|_2^2 + \log n \mathbb{E}_{\tilde{\mathbf{A}}'} \max_{1 \le i \le n} \|\tilde{\mathbf{a}}_i'\|_2^2 + C\right)^{1/2} + \log(d+n)\mathbb{E}_{\tilde{\mathbf{A}}'} \max_{1 \le i \le n} \|\tilde{\mathbf{a}}_i'\|_2^2$$

Thus it remains to provide an upper bound for $\mathbb{E}_{\tilde{\mathbf{A}}'} \max_{1 \le i \le n} \|\tilde{\mathbf{a}}_i'\|_2^2$, we have by Jensen's

$$\left(\mathbb{E}_{\tilde{\mathbf{A}}'} \max_{1 \le i \le n} \|\tilde{\mathbf{a}}_i'\|_2^2\right)^3 \le \mathbb{E}_{\tilde{\mathbf{A}}'} \max_{1 \le i \le n} \|\tilde{\mathbf{a}}_i'\|_2^6 \le \sum_{i=1}^n \mathbb{E}_{\tilde{\mathbf{a}}_i'} \|\tilde{\mathbf{a}}_i'\|_2^6$$

Now by Definition 2 and Holder's we have

$$\mathbb{E}_{\tilde{\mathbf{a}}_i'} \|\tilde{\mathbf{a}}_i'\|_2^6 = \sum_{\alpha_1,\alpha_2,\alpha_3} \mathbb{E}_{\tilde{\mathbf{a}}_i'} \tilde{\mathbf{a}}_{i\alpha_1}'^2 \tilde{\mathbf{a}}_{i\alpha_2}'^2 \tilde{\mathbf{a}}_{i\alpha_3}'^2 \le \sum_{\alpha_1,\alpha_2,\alpha_3} \left(\mathbb{E}_{\tilde{\mathbf{a}}_i'} \tilde{\mathbf{a}}_{i\alpha_1}'^6\right)^{1/3}\left(\mathbb{E}_{\tilde{\mathbf{a}}_i'} \tilde{\mathbf{a}}_{i\alpha_2}'^6\right)^{1/3}\left(\mathbb{E}_{\tilde{\mathbf{a}}_i'} \tilde{\mathbf{a}}_{i\alpha_3}'^6\right)^{1/3} \le \sum_{\alpha_1,\alpha_2,\alpha_3} \frac{C}{d^3} \le C$$

Thus

$$\mathbb{E}_{\tilde{\mathbf{A}}'} \max_{1 \le i \le n} \|\tilde{\mathbf{a}}_i'\|_2^2 \le Cn^{\frac{1}{3}}$$

All in all

$$\mathbb{E}_{\tilde{\mathbf{A}}'} \|\tilde{\mathbf{A}}'\|_{op} \lesssim \log(d+n)n^{\frac{1}{3}}$$

Thus by Markov's inequality for $s > 0$

$$\mathbb{P}\left(\|\tilde{\mathbf{A}}'\|_{op} \ge Cn^{\frac{1}{3}+s}\right) \le \frac{\mathbb{E}_{\tilde{\mathbf{A}}'} \|\tilde{\mathbf{A}}'\|_{op}}{Cn^{\frac{1}{3}+s}} \le \frac{C' \log(d+n)n^{\frac{1}{3}}}{n^{\frac{1}{3}+s}} = \frac{C' \log(d+n)}{n^s}$$

$\square$

**Lemma 9.** *For any function* $\psi : \mathbb{R} \to \mathbb{R}$ *with bounded third derivative,* $\tilde{\psi}(\mathbf{w}) := \mathbb{E}_{\mathbf{a}}\psi(\mathbf{a}^T\mathbf{w})$ *is regular*

*Proof.* We only need to verify the third order Taylor remainder bound,

$$\sum_{\alpha_1,\alpha_2,\alpha_3} \frac{\partial^3 \tilde{\psi}}{\partial w_{\alpha_1} \partial w_{\alpha_2} \partial w_{\alpha_3}} v_{\alpha_1} v_{\alpha_2} v_{\alpha_3} = \sum_{\alpha_1,\alpha_2,\alpha_3} \mathbb{E}_{\mathbf{a}}\psi'''(\mathbf{a}^T\mathbf{w})a_{\alpha_1}a_{\alpha_2}a_{\alpha_3}v_{\alpha_1}v_{\alpha_2}v_{\alpha_3} = \mathbb{E}_{\mathbf{a}}(\mathbf{a}^T\mathbf{v})^3\psi'''(\mathbf{a}^T\mathbf{w})$$

By Cauchy Schwartz, we have

$$\sum_{\alpha_1,\alpha_2,\alpha_3} \frac{\partial^3 \tilde{\psi}}{\partial w_{\alpha_1} \partial w_{\alpha_2} \partial w_{\alpha_3}} v_{\alpha_1} v_{\alpha_2} v_{\alpha_3} \le \sqrt{\mathbb{E}_{\mathbf{a}}\psi'''(\mathbf{a}^T\mathbf{w})^2}\sqrt{\mathbb{E}_{\mathbf{a}}(\mathbf{a}^T\mathbf{v})^6}$$

By boundedness of the third derivative of $\psi$, and assumptions

$$\sum_{\alpha_1,\alpha_2,\alpha_3} \frac{\partial^3 \tilde{\psi}}{\partial w_{\alpha_1} \partial w_{\alpha_2} \partial w_{\alpha_3}} v_{\alpha_1} v_{\alpha_2} v_{\alpha_3} \le C_\psi \frac{\|\mathbf{v}\|_2^3}{d^{3/2}} \le C_\psi \|\mathbf{v}\|_3^3$$

$\square$

# E   Generalized CGMT

After reducing the analysis to the equivalent gaussian model, one can leverage a new generalization of the CGMT Akhtiamov et al. [2024] to analyze $\Phi(\mathbf{G})$ which is stated as follows and recover the precise asymptotic properties of the solutions.

**Theorem 5** (Generalized CGMT). *Let $\mathcal{S}_w \subset \mathbb{R}^d, \mathcal{S}_{v_1} \subset \mathbb{R}^{n_1}, \ldots, \mathcal{S}_{v_k} \subset \mathbb{R}^{n_k}$ be compact convex sets. Denote $\mathcal{S}_v := \mathcal{S}_{v_1} \times \cdots \times \mathcal{S}_{v_k}$, let $\mathbf{v} \in \mathcal{S}_v$ stand for $(\mathbf{v}_1, \ldots, \mathbf{v}_k) \in \mathcal{S}_{v_1} \times \cdots \times \mathcal{S}_{v_k}$ and $\psi(\mathbf{w}, \mathbf{v}) : \mathcal{S}_w \times \mathcal{S}_v \to \mathbb{R}$ be convex on $\mathcal{S}_w$ and concave on $\mathcal{S}_v$. Also let $\mathbf{\Sigma}_1, \ldots, \mathbf{\Sigma}_k \in \mathbb{R}^{d \times d}$ be arbitrary PSD matrices. Furthermore, let $\mathbf{G}_1 \in \mathbb{R}^{n_1 \times d}, \ldots, \mathbf{G}_k \in \mathbb{R}^{n_k \times d}, \mathbf{g}_1, \ldots, \mathbf{g}_k \in \mathbb{R}^d, \mathbf{h}_1 \in \mathbb{R}^{n_1}, \ldots, \mathbf{h}_k \in \mathbb{R}^{n_k}$ all have i.i.d $\mathcal{N}(0,1)$ components and $\mathbf{G} = (\mathbf{G}_1, \ldots, \mathbf{G}_k)$, $\mathbf{g} = (\mathbf{g}_1, \ldots, \mathbf{g}_k)$, $\mathbf{h} = (\mathbf{h}_1, \ldots, \mathbf{h}_k)$ be the corresponding k-tuples. Define*

$$\Psi(\mathbf{G}) := \min_{\mathbf{w} \in \mathcal{S}_w} \max_{\mathbf{v} \in \mathcal{S}_v} \sum_{\ell=1}^{k} \mathbf{v}_\ell^T \mathbf{G}_\ell \mathbf{\Sigma}_\ell^{\frac{1}{2}} \mathbf{w} + \psi(\mathbf{w}, \mathbf{v}) \tag{26}$$

$$\phi(\mathbf{g}, \mathbf{h}) := \min_{\mathbf{w} \in \mathcal{S}_w} \max_{\mathbf{v} \in \mathcal{S}_v} \sum_{\ell=1}^{k} \left[ \|\mathbf{v}_\ell\|_2 \mathbf{g}_\ell^T \mathbf{\Sigma}_\ell^{\frac{1}{2}} \mathbf{w} + \|\mathbf{\Sigma}_\ell^{\frac{1}{2}} \mathbf{w}\|_2 \mathbf{h}_\ell^T \mathbf{v}_\ell \right] + \psi(\mathbf{w}, \mathbf{v}) \tag{27}$$

*Let $\mathcal{S}$ be an arbitrary open subset of $\mathcal{S}_w$ and $\mathcal{S}^c = \mathcal{S}_w \setminus \mathcal{S}$. Let $\phi_{\mathcal{S}}$ be the optimal value of the optimization 27 when $\mathcal{S}_w$ is restricted to $\mathcal{S}$. Assume also that there exist $\epsilon, \delta > 0$, $\bar{\phi}, \bar{\phi}_{\mathcal{S}^c}$ such that*

- $\bar{\phi}_{\mathcal{S}^c} \geq \bar{\phi} + 3\delta$

- $\phi(g, h) < \bar{\phi} + \delta$ *with probability at least* $1 - \epsilon$

- $\phi_{\mathcal{S}^c} > \bar{\phi}_{\mathcal{S}^c} - \delta$ *with probability at least* $1 - \epsilon$

*Then for the optimal solution $\mathbf{w}$ of the optimization 26, we have $\mathbb{P}(\mathbf{w}_\Psi(\mathbf{G}) \in \mathcal{S}) \geq 1 - 2^{k+1}\epsilon$*

# F   Proof of Theorem 2

By leveraging our universality theorem 1 we are ready to analyze the linear regression problem described in 2 in full details

*Proof.* First note that for the ground truth we have $w_* = argmin\mathbb{E}_P(y - \mathbf{x}^T\mathbf{w})^2 = \mathbf{R}_x^{-1}\mathbf{R}_{xy}$. By a simple transformation, the constraint could also be written as

$$\begin{pmatrix} \mathbf{X} & \mathbf{y} \end{pmatrix} \begin{pmatrix} \mathbf{w} \\ -1 \end{pmatrix} = 0$$

We would like to write for a $\mathbf{G}$ with i.i.d entries:

$$\begin{pmatrix} \mathbf{X} & \mathbf{y} \end{pmatrix} = \mathbf{GS}$$

Indeed

$$\mathbf{S}^T \mathbb{E} \mathbf{G}^T \mathbf{GS} = \mathbf{S}^T \mathbf{S} = \mathbb{E} \begin{pmatrix} \mathbf{X}^T \\ \mathbf{y}^T \end{pmatrix} \begin{pmatrix} \mathbf{X} & \mathbf{y} \end{pmatrix} = \mathbb{E} \begin{pmatrix} \mathbf{R}_x & \mathbf{R}_{xy} \\ \mathbf{R}_{yx} & R_y \end{pmatrix}$$

Now we can denote $\mathbf{S} = \begin{pmatrix} \mathbf{R}_x^{1/2} & \mathbf{R}_x^{-T/2}\mathbf{R}_{xy} \\ 0 & R_\Delta^{1/2} \end{pmatrix}$ with $R_\Delta := R_y - \mathbf{R}_{yx}\mathbf{R}_x^{-1}\mathbf{R}_{xy}$. And $\mathbf{R}_x^{-T/2}\mathbf{R}_{xy} = \mathbf{R}_x^{1/2}\mathbf{w}_*$.

Thus we can write the optimization as

$$\min_{\mathbf{w}} \|\mathbf{w} - \mathbf{w}_0\|^2$$

$$s.t \quad \mathbf{G} \begin{pmatrix} \mathbf{R}_x^{1/2}(\mathbf{w} - \mathbf{w}_*) \\ -R_\Delta^{1/2} \end{pmatrix} = 0$$

we can write the optimization as

$$\min_{\mathbf{w}} \|\mathbf{w} - \mathbf{w}_0\|^2$$

$$s.t \quad \mathbf{G}\begin{pmatrix} \mathbf{R}_x^{1/2}(\mathbf{w} - \mathbf{w}_*) \\ -R_\Delta^{1/2} \end{pmatrix} = 0$$

Let $\mathbf{u} := \mathbf{R}_x^{1/2}(\mathbf{w} - \mathbf{w}_*)$ and $\mathbf{x} = \mathbf{w}_* - \mathbf{w}_0$, then

$$\min_{\mathbf{w}} \|\mathbf{R}_x^{-1/2}\mathbf{u} + \mathbf{x}\|^2$$

$$s.t \quad \mathbf{G}\begin{pmatrix} \mathbf{u} \\ -R_\Delta^{1/2} \end{pmatrix} = 0$$

We Lagrange multiplier to bring in the constraint:

$$\min_{\mathbf{u}} \max_{\mathbf{v}} \|\mathbf{R}_x^{-1/2}\mathbf{u} + \mathbf{x}\|^2 + \mathbf{v}^T \mathbf{G}\begin{pmatrix} \mathbf{u} \\ -R_\Delta^{1/2} \end{pmatrix}$$

Now we can appeal to CGMT to obtain the AO:

$$\min_{\mathbf{u}} \max_{\mathbf{v}} \|\mathbf{R}_x^{-1/2}\mathbf{u} + \mathbf{x}\|^2 + \|\mathbf{v}\|(\mathbf{g}^T\mathbf{u} + g'R_\Delta^{1/2}) + \left\|\begin{pmatrix} \mathbf{u} \\ -R_\Delta^{1/2} \end{pmatrix}\right\| \mathbf{h}^T\mathbf{v}$$

Dropping the $g'R_\Delta^{1/2}$ term and doing the optimization over the direction of $\mathbf{v}$, yields

$$\min_{\mathbf{u}} \max_{\beta \geq 0} \|\mathbf{R}_x^{-1/2}\mathbf{u} + \mathbf{x}\|^2 + \beta\|\mathbf{h}\|\sqrt{\|\mathbf{u}\|^2 + R_\Delta} + \beta\mathbf{g}^T\mathbf{u}$$

Using the identity $\sqrt{x} = \min_{\tau \geq 0} \frac{\tau}{2} + \frac{x^2}{2\tau}$

$$\min_{\tau \geq 0, \mathbf{u}} \max_{\beta \geq 0} \|\mathbf{R}_x^{-1/2}\mathbf{u} + \mathbf{x}\|^2 + \frac{\beta\|\mathbf{h}\|\tau}{2} + \frac{\beta\|\mathbf{h}\|}{2\tau}(\|\mathbf{u}\|^2 + R_\Delta) + \beta\mathbf{g}^T\mathbf{u}$$

Now we obtain a quadratic optimization over $u$ which could be rewritten as

$$\begin{pmatrix} \mathbf{u}^T & 1 \end{pmatrix}\begin{pmatrix} \mathbf{R}_x^{-1} + \frac{\beta\sqrt{n}}{2\tau}\mathbf{I} & \mathbf{R}_x^{-1/2}\mathbf{x} + \frac{\beta}{2}\mathbf{g} \\ \mathbf{x}^T\mathbf{R}_x^{-T/2} + \frac{\beta}{2}\mathbf{g}^T & \|\mathbf{x}\|^2 \end{pmatrix}\begin{pmatrix} \mathbf{u} \\ 1 \end{pmatrix}$$

Doing the optimization over $\mathbf{u}$, we are left with

$$\min_{\tau \geq 0} \max_{\beta \geq 0} \|\mathbf{x}\|^2 + \frac{\beta\tau\sqrt{n}}{2} + \frac{\beta\sqrt{n}}{2\tau}R_\Delta - (\mathbf{R}_x^{-1/2}\mathbf{x} + \frac{\beta}{2}\mathbf{g})^T(\mathbf{R}_x^{-1} + \frac{\beta\sqrt{n}}{2\tau}\mathbf{I})^{-1}(\mathbf{R}_x^{-1/2}\mathbf{x} + \frac{\beta}{2}\mathbf{g})$$

Expanding the third term yields

$$\min_{\tau \geq 0} \max_{\beta \geq 0} \|\mathbf{x}\|^2 + \frac{\beta\tau\sqrt{n}}{2} + \frac{\beta\sqrt{n}}{2\tau}R_\Delta - \mathbf{x}^T\mathbf{R}_x^{-T/2}(\mathbf{R}_x^{-1} + \frac{\beta\sqrt{n}}{2\tau}\mathbf{I})^{-1}\mathbf{R}_x^{-1/2}\mathbf{x} - \frac{\beta^2}{4}\mathbf{g}^T(\mathbf{R}_x^{-1} + \frac{\beta\sqrt{n}}{2\tau}\mathbf{I})^{-1}\mathbf{g}$$

Let $\mathbf{z} := \mathbf{R}_x^{1/2}\mathbf{x}$, we assume $\mathbb{E}\mathbf{z}\mathbf{z}^T = \frac{\|\mathbf{z}\|^2}{d}I$, and $e_a = \mathbf{x}^T\mathbf{R}_x\mathbf{x} = \|\mathbf{z}\|^2$ using concentration:

$$\min_{\tau \geq 0} \max_{\beta \geq 0} \|x\|^2 + \frac{\beta\tau\sqrt{n}}{2} + \frac{\beta\sqrt{n}}{2\tau}R_\Delta - d\frac{1}{d}\text{Tr}\mathbf{R}_x^{-T}(\mathbf{R}_x^{-1} + \frac{\beta\sqrt{n}}{2\tau}\mathbf{I})^{-1}\mathbf{R}_x^{-1} - d\frac{\beta^2}{4}\frac{1}{d}\text{Tr}(\mathbf{R}_x^{-1} + \frac{\beta\sqrt{n}}{2\tau}\mathbf{I})^{-1}$$

Therefore

$$\min_{\tau \geq 0} \max_{\beta \geq 0} \|\mathbf{x}\|^2 + \frac{\beta\tau\sqrt{n}}{2} + \frac{\beta\sqrt{n}}{2\tau}R_\Delta - e_a\int p(r)\frac{1}{r(1 + \frac{\beta\sqrt{n}}{2\tau}r)}dr - d\frac{\beta^2}{4}\int p(r)\frac{r}{1 + \frac{\beta\sqrt{n}}{2\tau}r}dr$$

Dropping the constant term $\|\mathbf{x}\|^2$, we arrive at the following scalarized optimization

$$\min_{\tau \geq 0} \max_{\beta \geq 0} \frac{\beta\tau\sqrt{n}}{2} + \frac{\beta\sqrt{n}}{2\tau}R_\Delta - d\int p(r)\frac{\frac{e_a}{d} + \frac{\beta^2}{4}r^2}{r(1 + \frac{\beta\sqrt{n}}{2\tau}r)}dr$$

Let $e'_a := \frac{e_a}{d}$. To analyze the optimization further, noting that the objective is differentiable w.r.t $\beta, \tau$, by taking derivatives we have at the optimal point:

$$\frac{\tau\sqrt{n}}{2} + \frac{\sqrt{n}R_\Delta}{2\tau} - d\int p(r)\frac{\tau(\sqrt{n}r^2\beta^2 + 4r\tau\beta - 4e'_a\sqrt{n})}{2(2\tau + r\beta\sqrt{n})^2}dr = 0$$

$$\frac{\beta\sqrt{n}}{2} - \frac{\beta R_\Delta\sqrt{n}}{2\tau^2} - d\int p(r)\frac{\beta\sqrt{n}(\beta^2r^2 + 4e'_a)}{2(2\tau + r\beta\sqrt{n})^2}dr = 0$$

Rearranging

$$\tau^2\sqrt{n} + \sqrt{n}R_\Delta = d\int p(r)\frac{\tau^2(\sqrt{n}r^2\beta^2 + 4r\tau\beta - 4e'_a\sqrt{n})}{(2\tau + r\beta\sqrt{n})^2}dr$$

$$\tau^2\sqrt{n} - \sqrt{n}R_\Delta = d\int p(r)\frac{\tau^2(\sqrt{n}\beta^2r^2 + \sqrt{n}4e'_a)}{(2\tau + r\beta\sqrt{n})^2}dr$$

Summing and subtracting:

$$n = d\int p(r)\frac{nr^2\beta^2 + 2\sqrt{n}r\tau\beta}{(2\tau + r\beta\sqrt{n})^2}dr = d\int p(r)(1 - \frac{2\sqrt{n}r\tau\beta + 4\tau^2}{(2\tau + r\beta\sqrt{n})^2})dr$$

$$\sqrt{n}R_\Delta = d\int p(r)\frac{\tau^2(2r\tau\beta - 4e'_a\sqrt{n})}{(2\tau + r\beta\sqrt{n})^2}dr$$

Thus

$$\frac{d-n}{d} = \int p(r)\frac{2\sqrt{n}r\tau\beta + 4\tau^2}{(2\tau + r\beta\sqrt{n})^2}dr = \int p(r)\frac{2\tau}{2\tau + r\beta\sqrt{n}}dr = \frac{2\tau}{\beta\sqrt{n}}S_P(-\frac{2\tau}{\beta\sqrt{n}} + i0)$$

$$\sqrt{n}R_\Delta = d\int p(r)\frac{\tau^2(2r\tau\beta - 4e'_a\sqrt{n})}{(2\tau + r\beta\sqrt{n})^2}dr$$

If one can solve $\frac{d-n}{d} = \frac{2\tau}{\beta\sqrt{n}}S_P(-\frac{2\tau}{\beta\sqrt{n}} + i0)$ to find $\theta := \frac{\beta}{\tau}$, then plugging into the second equation yields

$$\sqrt{n}R_\Delta^2 = d\int p(r)\frac{\tau^2(2r\theta\tau^2 - 4e'_a\sqrt{n})}{(2\tau + r\theta\tau\sqrt{n})^2}dr = d\int p(r)\frac{2r\theta\tau^2 - 4e'_a\sqrt{n}}{(2 + r\theta\sqrt{n})^2}dr$$

Which yields the following equation for $\tau^2$

$$\tau^2 = \frac{\sqrt{n}}{2\theta d\int p(r)\frac{r}{(2+r\theta\sqrt{n})^2}dr}\left[R_\Delta + 4de'_a\int\frac{p(r)}{(2+r\theta\sqrt{n})^2}dr\right]$$

Now note that

$$\frac{d-n}{d} = \int p(r)\frac{2\tau}{2\tau + r\beta\sqrt{n}}dr = \int p(r)\frac{2}{2 + r\theta\sqrt{n}}dr = \int p(r)\frac{2(2 + r\theta\sqrt{n})}{(2 + r\theta\sqrt{n})^2}dr =$$

$$= 2\theta\sqrt{n}\int p(r)\frac{r}{(2 + r\theta\sqrt{n})^2}dr + 4\int\frac{p(r)}{(2 + r\theta\sqrt{n})^2}dr$$

Let $\alpha := 4\int\frac{p(r)}{(2+r\theta\sqrt{n})^2}dr$, $\alpha' := 2\theta\sqrt{n}\int p(r)\frac{r}{(2+r\theta\sqrt{n})^2}dr$ and . Then the expression for $\tau$ could be written as

$$\tau^2 = \frac{n}{d\alpha'}(R_\Delta + Be_a)$$

As $\alpha + \alpha' = \frac{d-n}{d}$, one can write

$$\tau^2 = \frac{n}{d(\frac{d-n}{d} - \alpha)}(R_\Delta + \alpha e_a) = \frac{n}{d-n-\alpha d}(R_\Delta + \alpha e_a) = \frac{n}{d(1-\alpha)-n}R_\Delta + \frac{\alpha n}{d(1-\alpha)-n}e_a$$

Now the generalization error is

$$\tau^2 - R_\Delta = \frac{2n - d(1-\alpha)}{d(1-\alpha)-n}R_\Delta + \frac{nB}{d(1-\alpha)-n}e_a = \frac{(2n-d)R_\Delta + (dR_\Delta + ne_a)\alpha}{d-n-d\alpha}$$

With $\alpha = 4 \int \frac{p(r)}{(2+r\theta\sqrt{n})^2} dr$. Now note that using Cauchy Schwarz applied to $\sqrt{p(r)}$ and $\frac{2\sqrt{p(r)}}{2+r\theta\sqrt{n}}$

$$(\frac{d-n}{d})^2 = (\int p(r) \frac{2}{2+r\theta\sqrt{n}} dr)^2 \leq \int p(r) dr \int \frac{4p(r)}{(2+r\theta\sqrt{n})^2} dr = \alpha$$

Thus $\alpha \geq (\frac{d-n}{d})^2$. Now for the scalar distribution $p(r) = \delta(r - r_0)$, since $2 + r_0\theta\sqrt{n} = 2 + \frac{2n}{d-n} = \frac{2d}{d-n}$, then $\alpha = (\frac{d-n}{d})^2$. Thus $\alpha$ achieves its lower bound (uniquely) for the scalar distribution. Now note that assuming $n, d, e_a, R_\Delta$ are fixed, the generalization error is increasing in $\alpha$, thus the error expression for the generalization error in the scalar distribution case is a lower bound. $\qquad \square$

## G  Useful Results

**Lemma 10.** *Given a weight vector* $\mathbf{w}$*, and assuming the feature vectors are equally likely to be drawn from class 1 or class 2 and satisfy part 2 of Assumptions 4, the corresponding classification error is given by*

$$\frac{1}{2}Q(\frac{\boldsymbol{\mu}_1^T \mathbf{w}}{\sqrt{\mathbf{w}^T \boldsymbol{\Sigma}_1 \mathbf{w}}}) + \frac{1}{2}Q(-\frac{\boldsymbol{\mu}_2^T \mathbf{w}}{\sqrt{\mathbf{w}^T \boldsymbol{\Sigma}_2 \mathbf{w}}}) \tag{28}$$

**Lemma 11.** *Assume that the feature vectors are equally likely to be drawn from class 1 or class 2, Assumptions 4 hold and* $\mathbf{w}_0$ *is defined as in Assumption 2. Then the classification error corresponding to* $\mathbf{w}_0$ *is equal to*

$$\frac{1}{2}Q\left(\frac{\sqrt{1-r}Tr(\boldsymbol{\Sigma}_1 + \boldsymbol{\Sigma}_2)^{-1}}{\sqrt{2Tr(\boldsymbol{\Sigma}_1(\boldsymbol{\Sigma}_1 + \boldsymbol{\Sigma}_2)^{-2}) + \frac{t_\eta^2}{t_*^2}Tr\boldsymbol{\Sigma}_1}}\right) + \frac{1}{2}Q\left(\frac{\sqrt{1-r}Tr(\boldsymbol{\Sigma}_1 + \boldsymbol{\Sigma}_2)^{-1}}{\sqrt{2Tr(\boldsymbol{\Sigma}_2(\boldsymbol{\Sigma}_1 + \boldsymbol{\Sigma}_2)^{-2}) + \frac{t_\eta^2}{t_*^2}Tr\boldsymbol{\Sigma}_2}}\right)$$

## H  Proof of Lemma 1

Let us proceed to prove Lemma 1. We will take the gradient of (28) w.r.t. $w$ and set it to 0:

*Proof.*

$$\nabla_{\mathbf{w}}\left[\frac{1}{2}Q(-\frac{\boldsymbol{\mu}_2^T \mathbf{w}}{\sqrt{\mathbf{w}^T \boldsymbol{\Sigma}_2 \mathbf{w}}})\right] = \frac{1}{4\sqrt{2\pi}}\left[\exp(-\frac{\mathbf{w}^T \boldsymbol{\mu}_2 \boldsymbol{\mu}_2^T \mathbf{w}}{2\mathbf{w}^T \boldsymbol{\Sigma}_2 \mathbf{w}})\frac{\sqrt{\mathbf{w}^T \boldsymbol{\Sigma}_2 \mathbf{w}}}{\boldsymbol{\mu}_2^T \mathbf{w}}\frac{\mathbf{w}^T \boldsymbol{\Sigma}_2 \mathbf{w}\boldsymbol{\mu}_2\boldsymbol{\mu}_2^T \mathbf{w} - \mathbf{w}^T \boldsymbol{\mu}_2\boldsymbol{\mu}_2^T \mathbf{w}\boldsymbol{\Sigma}_2 \mathbf{w}}{(\mathbf{w}^T \boldsymbol{\Sigma}_2 \mathbf{w})^2}\right] =$$

$$\frac{1}{4\sqrt{2\pi}}\left[\frac{1}{s_2}exp(-\frac{s_2^2}{2})(\boldsymbol{\mu}_2 t_2 - s_2^2 \boldsymbol{\Sigma}_2 \mathbf{w})\right] \quad \text{where } s_2 = \frac{\boldsymbol{\mu}_2^T \mathbf{w}}{\sqrt{\mathbf{w}^T \boldsymbol{\Sigma}_2 \mathbf{w}}}, t_2 = \frac{\boldsymbol{\mu}_2^T \mathbf{w}}{\mathbf{w}^T \boldsymbol{\Sigma}_2 \mathbf{w}}$$

$$\nabla_{\mathbf{w}}\left[\frac{1}{2}Q(\frac{\boldsymbol{\mu}_1^T \mathbf{w}}{\sqrt{\mathbf{w}^T \boldsymbol{\Sigma}_1 \mathbf{w}}})\right] = -\frac{1}{4\sqrt{2\pi}}\left[\exp(-\frac{\mathbf{w}^T \boldsymbol{\mu}_1 \boldsymbol{\mu}_1^T \mathbf{w}}{2\mathbf{w}^T \boldsymbol{\Sigma}_1 \mathbf{w}})\frac{\sqrt{\mathbf{w}^T \boldsymbol{\Sigma}_1 \mathbf{w}}}{\boldsymbol{\mu}_1^T \mathbf{w}}\frac{\mathbf{w}^T \boldsymbol{\Sigma}_1 \mathbf{w}\boldsymbol{\mu}_1\boldsymbol{\mu}_1^T \mathbf{w} - \mathbf{w}^T \boldsymbol{\mu}_1\boldsymbol{\mu}_1^T \mathbf{w}\boldsymbol{\Sigma}_1 \mathbf{w}}{(\mathbf{w}^T \boldsymbol{\Sigma}_1 \mathbf{w})^2}\right] =$$

$$-\frac{1}{4\sqrt{2\pi}}\left[\frac{1}{s_1}\exp(-\frac{s_1^2}{2})(\boldsymbol{\mu}_1 t_1 - s_1^2 \boldsymbol{\Sigma}_1 \mathbf{w})\right] \quad \text{where } s_1 = \frac{\boldsymbol{\mu}_1^T \mathbf{w}}{\sqrt{\mathbf{w}^T \boldsymbol{\Sigma}_1 \mathbf{w}}}, t_1 = \frac{\boldsymbol{\mu}_1^T \mathbf{w}}{\mathbf{w}^T \boldsymbol{\Sigma}_1 \mathbf{w}}$$

Hence, the following equality holds at any optimal $\mathbf{w}$:

$$\frac{1}{s_2}\exp(-\frac{s_2^2}{2})(\boldsymbol{\mu}_2 t_2 - s_2^2 \boldsymbol{\Sigma}_2 \mathbf{w}) = \frac{1}{s_1}\exp(-\frac{s_1^2}{2})(\boldsymbol{\mu}_1 t_1 - s_1^2 \boldsymbol{\Sigma}_1 \mathbf{w})$$

Therefore, $\mathbf{w} = (\tilde{s}_1 \boldsymbol{\Sigma}_1 + \tilde{s}_2 \boldsymbol{\Sigma}_2)^{-1}(\tilde{t}_1 \boldsymbol{\mu}_1 - \tilde{t}_2 \boldsymbol{\mu}_2)$, where we have

$$\tilde{s}_1 = s_1 \exp(-\frac{s_1^2}{2}), \quad \tilde{s}_2 = -s_2 \exp(-\frac{s_2^2}{2}),$$

$$\tilde{t}_1 = \frac{t_1}{s_1}\exp(-\frac{s_1^2}{2}), \quad \tilde{t}_2 = \frac{t_2}{s_2}\exp(-\frac{s_2^2}{2})$$

Now we can consider the following optimization over 4 scalar variables:

$$\min_{\tilde{s}_1,\tilde{s}_2,\tilde{t}_1,\tilde{t}_2} \frac{1}{2}Q\left(\frac{\boldsymbol{\mu}_1^T\mathbf{w}}{\sqrt{\mathbf{w}^T\boldsymbol{\Sigma}_1\mathbf{w}}}\right) + \frac{1}{2}Q\left(-\frac{\boldsymbol{\mu}_2^T\mathbf{w}}{\sqrt{\mathbf{w}^T\boldsymbol{\Sigma}_2\mathbf{w}}}\right) \tag{29}$$

Note that since the optimization (29) is low-dimensional, by the Convexity Lemma Liese and Miescke [2008], we can interchange the limit in $n$ and min, implying that we can analyze the concentrated optimization instead. We then arrive at the following fixed point equations using Assumptions 4:

$$\frac{s_1^2}{t_1^2} = \text{Tr}(\boldsymbol{\Sigma}_1(\tilde{s}_1\boldsymbol{\Sigma}_1 + \tilde{s}_2\boldsymbol{\Sigma}_2)^{-2})(\tilde{t}_1^2 + \tilde{t}_2^2 - 2\tilde{t}_1\tilde{t}_2 r)d \tag{30}$$

$$\frac{s_1^2}{t_1} = \text{Tr}(\tilde{s}_1\boldsymbol{\Sigma}_1 + \tilde{s}_2\boldsymbol{\Sigma}_2)^{-1}(\tilde{t}_1 - \tilde{t}_2 r)d \tag{31}$$

$$\frac{s_2^2}{t_2^2} = \text{Tr}(\boldsymbol{\Sigma}_2(\tilde{s}_1\boldsymbol{\Sigma}_1 + \tilde{s}_2\boldsymbol{\Sigma}_2)^{-2})(\tilde{t}_1^2 + \tilde{t}_2^2 - 2\tilde{t}_1\tilde{t}_2 r)d \tag{32}$$

$$\frac{s_2^2}{t_2} = \text{Tr}(\tilde{s}_1\boldsymbol{\Sigma}_1 + \tilde{s}_2\boldsymbol{\Sigma}_2)^{-1}(\tilde{t}_2 - \tilde{t}_1 r)d \tag{33}$$

Note that, since we assume $p$ is exchangeable in its arguments, the system of equations (30) is invariant under the transformation $(s_1, t_1, s_2, t_2) \rightarrow (-s_2, t_2, -s_1, t_1)$. As such, $s_1 = -s_2$ and $t_1 = t_2$ at the optimal point, implying that $\mathbf{w}_* = (\boldsymbol{\Sigma}_1 + \boldsymbol{\Sigma}_2)^{-1}(\boldsymbol{\mu}_1 - \boldsymbol{\mu}_2)$ is the optimal classifier, as for the chosen linear classification procedure the performance in invariant to the rescalings of the classifier vector and therefore we can completely drop $s_1, s_2, t_1, t_2$ from the equations.

$\square$

# I   Proof of Theorem 3

By Gaussian universality proved in Theorem 1, we analyze the binary classification problem as follows

*Proof.* Define

$$\mathbf{R}_0 := \mathbb{E}_{\boldsymbol{\mu}_1,\boldsymbol{\mu}_2}\mathbf{w}_0\mathbf{w}_0^T = \mathbb{E}_{\boldsymbol{\mu}_1,\boldsymbol{\mu}_2}(t_*\mathbf{w}_* + t_\eta\boldsymbol{\eta})(t_*\mathbf{w}_* + t_\eta\boldsymbol{\eta})^T = \frac{t_\eta^2(1-r)}{d}I_d + t_*^2\mathbb{E}_{\boldsymbol{\mu}_1,\boldsymbol{\mu}_2}\mathbf{w}_*\mathbf{w}_*^T$$

$$= \frac{t_\eta^2(1-r)}{d}I_d + 2\frac{t_*^2}{d}(1-r)(\boldsymbol{\Sigma}_1 + \boldsymbol{\Sigma}_2)^{-2}$$

Note that the eigenvalue density of $\mathbf{R}_0$ is captured by $\phi(s_1, s_2) = \frac{t_\eta^2(1-r)}{d} + \frac{2t_*^2(1-r)}{d}(s_1 + s_2)^{-2}$, assuming $(s_1, s_2) \sim p$, where $p$ is the joint EPD function of $\boldsymbol{\Sigma}_1$ and $\boldsymbol{\Sigma}_2$.

We would like to start with finding $\alpha$. Note that the data matrix $\mathbf{X}$ can be decomposed as follows where $\mathbf{G}_1, \mathbf{G}_2$ are i.i.d. standard normal:

$$\mathbb{X} = \begin{pmatrix} \mathbf{G}_1\boldsymbol{\Sigma}_1^{1/2} \\ \mathbf{G}_2\boldsymbol{\Sigma}_2^{1/2} \end{pmatrix} + \begin{pmatrix} \mathbb{1} & 0 \\ 0 & \mathbb{1} \end{pmatrix}\begin{pmatrix} \boldsymbol{\mu}_1^T \\ \boldsymbol{\mu}_2^T \end{pmatrix}$$

We then have:

$$\mathbf{X}\mathbf{w}_0 = \begin{pmatrix} \mathbf{G}_1\boldsymbol{\Sigma}_1^{1/2}\mathbf{w}_0 \\ \mathbf{G}_2\boldsymbol{\Sigma}_2^{1/2}\mathbf{w}_0 \end{pmatrix} + \begin{pmatrix} \mathbb{1} & 0 \\ 0 & \mathbb{1} \end{pmatrix}\begin{pmatrix} \boldsymbol{\mu}_1^T\mathbf{w}_0 \\ \boldsymbol{\mu}_2^T\mathbf{w}_0 \end{pmatrix} \tag{34}$$

It is immediate to see that the first term of (34) is a centered normally distributed random vector with covariance $\text{diag}(\mathbf{w}_0^T \boldsymbol{\Sigma}_1 \mathbf{w}_0, \ldots, \mathbf{w}_0^T \boldsymbol{\Sigma}_1 \mathbf{w}_0, \mathbf{w}_0^T \boldsymbol{\Sigma}_2 \mathbf{w}_0, \ldots, \mathbf{w}_0^T \boldsymbol{\Sigma}_2 \mathbf{w}_0)$, and therefore its norm concentrates to $\sqrt{\frac{n}{2} \mathbf{w}_0^T (\boldsymbol{\Sigma}_1 + \boldsymbol{\Sigma}_2) \mathbf{w}_0} = \Theta(\sqrt{\text{Tr}(\boldsymbol{\Sigma}_1 + \boldsymbol{\Sigma}_2)} \|\mathbf{w}_0\|)$ according to our assumptions. The second term of (34) is a deterministic vector of norm $\Theta(\sqrt{n} \|\mathbf{w}_0\| \sqrt{1-r})$.

Similarly:

$$\mathbf{y}^T \mathbf{X} \mathbf{w}_0 = \begin{pmatrix} \mathbb{1}^T & -\mathbb{1}^T \end{pmatrix} \left( \begin{pmatrix} \mathbf{G}_1 \boldsymbol{\Sigma}_1^{1/2} \mathbf{w}_0 \\ \mathbf{G}_2 \boldsymbol{\Sigma}_2^{1/2} \mathbf{w}_0 \end{pmatrix} + \begin{pmatrix} \mathbb{1} & 0 \\ 0 & \mathbb{1} \end{pmatrix} \begin{pmatrix} \boldsymbol{\mu}_1^T \mathbf{w}_0 \\ \boldsymbol{\mu}_2^T \mathbf{w}_0 \end{pmatrix} \right) = \begin{pmatrix} \mathbb{1}^T & -\mathbb{1}^T \end{pmatrix} \begin{pmatrix} \mathbf{G}_1 \boldsymbol{\Sigma}_1^{1/2} \mathbf{w}_0 \\ \mathbf{G}_2 \boldsymbol{\Sigma}_2^{1/2} \mathbf{w}_0 \end{pmatrix} + \frac{n}{2} (\boldsymbol{\mu}_1 - \boldsymbol{\mu}_2)^T \mathbf{w}_0$$

$$(35)$$

Note that we can drop the first term of (35) compared to the second term. Indeed, the first term is a centered normal random variable with variance $\frac{n}{2} \mathbf{w}_0^T (\boldsymbol{\Sigma}_1 + \boldsymbol{\Sigma}_2) \mathbf{w}_0 = \Theta(\|\mathbf{w}_0\|^2 \text{Tr}(\boldsymbol{\Sigma}_1 + \boldsymbol{\Sigma}_2))$ and the second term is a deterministic quantity of order $\Theta(n \|\mathbf{w}_0\| \sqrt{1-r})$. Thus

$$\alpha = \frac{\mathbf{y}^T \mathbf{X} \mathbf{w}_0}{\|\mathbf{X} \mathbf{w}_0\|^2} = \frac{\frac{n}{2} (\boldsymbol{\mu}_1 - \boldsymbol{\mu}_2)^T \mathbf{w}_0}{\frac{n}{2} \mathbf{w}_0^T (\boldsymbol{\Sigma}_1 + \boldsymbol{\Sigma}_2) \mathbf{w}_0 + \frac{n}{2} (\boldsymbol{\mu}_1^T \mathbf{w}_0)^2 + \frac{n}{2} (\boldsymbol{\mu}_2^T \mathbf{w}_0)^2}$$

Moreover, by independence of $\boldsymbol{\eta}$ from $\boldsymbol{\mu}_i$ we can drop the first term in $(\boldsymbol{\mu}_1 - \boldsymbol{\mu}_2)^T \mathbf{w}_0 = t_\eta (\boldsymbol{\mu}_1 - \boldsymbol{\mu}_2)^T \boldsymbol{\eta} + t_* (\boldsymbol{\mu}_1 - \boldsymbol{\mu}_2)^T \mathbf{w}_*$ and have

$$(\boldsymbol{\mu}_1 - \boldsymbol{\mu}_2)^T \mathbf{w}_* = (\boldsymbol{\mu}_1 - \boldsymbol{\mu}_2)^T (\boldsymbol{\Sigma}_1 + \boldsymbol{\Sigma}_2)^{-1} (\boldsymbol{\mu}_1 - \boldsymbol{\mu}_2) \xrightarrow{\mathbb{P}} 2(1-r) \text{Tr}(\boldsymbol{\Sigma}_1 + \boldsymbol{\Sigma}_2)^{-1}$$

On the other hand:

$$\boldsymbol{\mu}_1^T \mathbf{w}_0 = t_\eta \boldsymbol{\mu}_1^T \boldsymbol{\eta} + t_* \boldsymbol{\mu}_1^T \mathbf{w}_* = t_\eta \boldsymbol{\mu}_1^T \boldsymbol{\eta} + t_* \boldsymbol{\mu}_1^T (\boldsymbol{\Sigma}_1 + \boldsymbol{\Sigma}_2)^{-1} (\boldsymbol{\mu}_1 - \boldsymbol{\mu}_2)$$

Which implies that we can drop the cross term in the expansion of $(\boldsymbol{\mu}_1^T \mathbf{w}_0)^2$:

$$(\boldsymbol{\mu}_1^T \mathbf{w}_0)^2 \approx t_\eta^2 (\boldsymbol{\mu}_1^T \boldsymbol{\eta})^2 + t_*^2 \left( \boldsymbol{\mu}_1^T (\boldsymbol{\Sigma}_1 + \boldsymbol{\Sigma}_2)^{-1} (\boldsymbol{\mu}_1 - \boldsymbol{\mu}_2) \right)^2 \xrightarrow{\mathbb{P}} \frac{1-r}{d} t_\eta^2 + \frac{t_*^2 (1-r)^2}{d^2} \text{Tr}^2 (\boldsymbol{\Sigma}_1 + \boldsymbol{\Sigma}_2)^{-1}$$

We can drop the first without loss of generality as $t_\eta$ usually does not grow with $d$. This implies that we have for $\alpha$

$$\alpha = \frac{\frac{n}{2} \frac{t_*(1-r)}{d} 2 \text{Tr}(\boldsymbol{\Sigma}_1 + \boldsymbol{\Sigma}_2)^{-1}}{\frac{n}{2} \text{Tr}(\mathbf{R}_0(\boldsymbol{\Sigma}_1 + \boldsymbol{\Sigma}_2)) + n \frac{t_*^2(1-r)^2}{d^2} \text{Tr}^2(\boldsymbol{\Sigma}_1 + \boldsymbol{\Sigma}_2)^{-1}}$$

$$= \frac{\frac{1}{2} \frac{t_*(1-r)}{d} 2 \text{Tr}(\boldsymbol{\Sigma}_1 + \boldsymbol{\Sigma}_2)^{-1}}{\frac{1}{2} \text{Tr}(\mathbf{R}_0(\boldsymbol{\Sigma}_1 + \boldsymbol{\Sigma}_2)) + \frac{t_*^2(1-r)^2}{d^2} \text{Tr}^2(\boldsymbol{\Sigma}_1 + \boldsymbol{\Sigma}_2)^{-1}}$$

Then SGD converges to

$$\min_{\mathbf{w}} \|\mathbf{w} - \alpha \mathbf{w}_0\|^2$$

$$s.t \quad \mathbf{y} = \mathbf{X} \mathbf{w}$$

Introducing a Lagrange multiplier $\mathbf{v} \in \mathbb{R}^n$:

$$\min_{\mathbf{w}} \max_{\mathbf{v}} \|\mathbf{w} - \alpha \mathbf{w}_0\|^2 + \mathbf{v}^T \mathbf{G} \begin{pmatrix} \boldsymbol{\Sigma}_1^{1/2} \\ \boldsymbol{\Sigma}_2^{1/2} \end{pmatrix} \mathbf{w} + \mathbf{v}^T \mathbf{M} \mathbf{w} - \mathbf{v}^T \mathbf{y}$$

Using Theorem 4 from Akhtiamov et al. [2024],

$$\min \max \| - \alpha_0 \|^2 + \sum_{i=1}^{2} \|v_i\|_i^T \boldsymbol{\Sigma}_i^{1/2} + \|\boldsymbol{\Sigma}_i^{1/2}\| \mathbb{h}_i^T{}_i + {}_i^T \mathbb{1} \boldsymbol{\mu}_i^T + {}_i^T \mathbb{1} (-1)^i$$

Denoting $\beta_i := \|\mathbf{v}_i\|$ and performing optimization over the direction of $v_i$:

$$\min \max_{\beta_1,\beta_2 \geq 0} \| - \alpha_0\|^2 + \sum_{i=1}^{2} \beta_i({}_i^T \Sigma_i^{1/2} + \|\mathbb{h}_i\|\Sigma_i^{1/2}\| + \mu_i^T \mathbb{1} + (-1)^i \mathbb{1}\|)$$

We have

$$\min_{\mathbf{w}} \max_{\beta_1,\beta_2 \geq 0} \|\mathbf{w} - \alpha\mathbf{w}_0\|^2 + \sum_{i=1}^{2} \beta_i(\mathbf{g}_i^T \Sigma_i^{1/2}\mathbf{w} + \sqrt{\frac{n}{2}}\sqrt{\|\Sigma_i^{1/2}\mathbf{w}\|^2 + (\mu_i^T\mathbf{w} + (-1)^i)^2})$$

Applying the square root trick to the norm inside the objective we arrive at:

$$\min_{\tau_1,\tau_2 \geq 0,\mathbf{w}} \max_{\beta_1,\beta_2 \geq 0} \|\mathbf{w} - \alpha\mathbf{w}_0\|^2 + \sum_{i=1}^{2} \beta_i\mathbf{g}_i^T \Sigma_i^{1/2}\mathbf{w} + \sqrt{\frac{n}{2}}\frac{\beta_i\tau_i}{2} + \frac{\beta_i}{2\tau_i}\sqrt{\frac{n}{2}}(\|\Sigma_i^{1/2}\mathbf{w}\|^2 + (\mu_i^T\mathbf{w} + (-1)^i)^2)$$

Introducing $\gamma_i$'s as the Fenchel duals for $(\mu_i^T\mathbf{w} + (-1)^i)^2$ we have:

$$\min_{\tau_1,\tau_2 \geq 0,\mathbf{w}} \max_{\beta_1,\beta_2 \geq 0,\gamma_1,\gamma_2} \|\mathbf{w} - \alpha\mathbf{w}_0\|^2 + \sum_{i=1}^{2} \beta_i\mathbf{g}_i^T \Sigma_i^{1/2}\mathbf{w} + \sqrt{\frac{n}{2}}\frac{\beta_i\tau_i}{2} + \frac{\beta_i}{2\tau_i}\sqrt{\frac{n}{2}}(\|\Sigma_i^{1/2}\mathbf{w}\|^2 + \mu_i^T\mathbf{w}\gamma_i + (-1)^i\gamma_i - \frac{\gamma_i^2}{4})$$

Performing optimization over $\mathbf{w}$ yields:

$$\min_{\tau_1,\tau_2 \geq 0} \max_{\beta_1,\beta_2 \geq 0,\gamma_1,\gamma_2} \alpha^2\|\mathbf{w}_0\|^2 + \sum_{i=1}^{2} \sqrt{\frac{n}{2}}\frac{\beta_i\tau_i}{2} + \sqrt{\frac{n}{2}}\frac{\beta_i}{2\tau_i}(\gamma_i(-1)^i - \frac{\gamma_i^2}{4})$$

$$- \left(-\alpha\mathbf{w}_0 + \frac{1}{2}(\beta_1\Sigma_1^{1/2}\mathbf{g}_1 + \beta_2\Sigma_2^{1/2}\mathbf{g}_2) + \frac{\beta_1}{4\tau_1}\sqrt{\frac{n}{2}}\gamma_1\mu_1 + \frac{\beta_2}{4\tau_2}\sqrt{\frac{n}{2}}\gamma_2\mu_2\right)^T \left(I + \frac{\beta_1}{2\tau_1}\sqrt{\frac{n}{2}}\Sigma_1 + \frac{\beta_2}{2\tau_2}\sqrt{\frac{n}{2}}\Sigma_2\right)^{-1}$$

$$\cdot \left(-\alpha\mathbf{w}_0 + \frac{1}{2}(\beta_1\Sigma_1^{1/2}\mathbf{g}_1 + \beta_2\Sigma_2^{1/2}\mathbf{g}_2) + \frac{\beta_1}{4\tau_1}\sqrt{\frac{n}{2}}\gamma_1\mu_1 + \frac{\beta_2}{4\tau_2}\sqrt{\frac{n}{2}}\gamma_2\mu_2\right)$$

We drop the $\alpha^2\|\mathbf{w}_0\|^2$ from the objective because it does not contain any of the variables the optimization is performed over and therefore dropping it will not change the optimal values of $\beta_1, \beta_2, \gamma_1, \gamma_2, \tau_1, \tau_2$. We obtain:

$$\min_{\tau_1,\tau_2 \geq 0} \max_{\beta_1,\beta_2 \geq 0,\gamma_1,\gamma_2} \sum_{i=1}^{2} \sqrt{\frac{n}{2}}\frac{\beta_i\tau_i}{2} + \sqrt{\frac{n}{2}}\frac{\beta_i}{2\tau_i}(\gamma_i(-1)^i - \frac{\gamma_i^2}{4}) - \alpha^2\mathbf{w}_0^T(I + \frac{\beta_1}{2\tau_1}\sqrt{\frac{n}{2}}\Sigma_1 + \frac{\beta_2}{2\tau_2}\sqrt{\frac{n}{2}}\Sigma_2)^{-1}\mathbf{w}_0-$$

$$-\frac{\beta_1^2}{4}\mathbf{g}_1^T\Sigma_1^{T/2}(I + \frac{\beta_1}{2\tau_1}\sqrt{\frac{n}{2}}\Sigma_1 + \frac{\beta_2}{2\tau_2}\sqrt{\frac{n}{2}}\Sigma_2)^{-1}\Sigma_1^{1/2}\mathbf{g}_1 - \frac{\beta_2^2}{4}\mathbf{g}_2^T\Sigma_2^{T/2}(I + \frac{\beta_1}{2\tau_1}\sqrt{\frac{n}{2}}\Sigma_1 + \frac{\beta_2}{2\tau_2}\sqrt{\frac{n}{2}}\Sigma_2)^{-1}\Sigma_2^{1/2}\mathbf{g}_2-$$

$$-\frac{n\beta_1^2\gamma_1^2}{32\tau_1^2}\mu_1^T(I + \frac{\beta_1}{2\tau_1}\sqrt{\frac{n}{2}}\Sigma_1 + \frac{\beta_2}{2\tau_2}\sqrt{\frac{n}{2}}\Sigma_2)^{-1}\mu_1 - \frac{n\beta_2^2\gamma_2^2}{32\tau_2^2}\mu_2^T(I + \frac{\beta_1}{2\tau_1}\sqrt{\frac{n}{2}}\Sigma_1 + \frac{\beta_2}{2\tau_2}\sqrt{\frac{n}{2}}\Sigma_2)^{-1}\mu_2+$$

$$+\frac{\beta_1}{2\tau_1}\sqrt{\frac{n}{2}}\gamma_1\alpha\mathbf{w}_0^T(I + \frac{\beta_1}{2\tau_1}\sqrt{\frac{n}{2}}\Sigma_1 + \frac{\beta_2}{2\tau_2}\sqrt{\frac{n}{2}}\Sigma_2)^{-1}\mu_1 + \frac{\beta_2}{2\tau_2}\sqrt{\frac{n}{2}}\gamma_2\alpha\mathbf{w}_0^T(I + \frac{\beta_2}{2\tau_2}\sqrt{\frac{n}{2}}\Sigma_1 + \frac{\beta_2}{2\tau_2}\sqrt{\frac{n}{2}}\Sigma_2)^{-1}\mu_2-$$

$$-\frac{n\beta_1\beta_2\gamma_1\gamma_2}{16\tau_1\tau_2}\mu_1^T(I + \frac{\beta_1}{2\tau_1}\sqrt{\frac{n}{2}}\Sigma_1 + \frac{\beta_2}{2\tau_2}\sqrt{\frac{n}{2}}\Sigma_2)^{-1}\mu_2$$

The objective above can be rewritten in the integral form in the following way:

$$\min_{\tau_1,\tau_2 \geq 0} \max_{\beta_1,\beta_2 \geq 0,\gamma_1,\gamma_2} \sum_{i=1}^{2} \sqrt{\frac{n}{2}}\frac{\beta_i\tau_i}{2} + \sqrt{\frac{n}{2}}\frac{\beta_i}{2\tau_i}(\gamma_i(-1)^i - \frac{\gamma_i^2}{4}) - \alpha^2\mathbf{w}_0^T(I + \frac{\beta_1}{2\tau_1}\sqrt{\frac{n}{2}}\Sigma_1 + \frac{\beta_2}{2\tau_2}\sqrt{\frac{n}{2}}\Sigma_2)^{-1}\mathbf{w}_0-$$

$$-d\int\int p(s_1,s_2)\frac{\frac{s_1\beta_1^2+s_2\beta_2^2}{4}}{1 + \frac{\beta_1}{2\tau_1}\sqrt{\frac{n}{2}}s_1 + \frac{\beta_2}{2\tau_2}\sqrt{\frac{n}{2}}s_2}ds_1ds_2 - \int\int p(s_1,s_2)\frac{\frac{n\beta_1^2\gamma_1^2}{32\tau_1^2} + \frac{n\beta_2^2\gamma_2^2}{32\tau_2^2}}{1 + \frac{\beta_1}{2\tau_1}\sqrt{\frac{n}{2}}s_1 + \frac{\beta_2}{2\tau_2}\sqrt{\frac{n}{2}}s_2}ds_1ds_2+$$

$$+\frac{\beta_1}{2\tau_1}\sqrt{\frac{n}{2}}\gamma_1\alpha\mathbf{w}_0^T(I + \frac{\beta_1}{2\tau_1}\sqrt{\frac{n}{2}}\Sigma_1 + \frac{\beta_2}{2\tau_2}\sqrt{\frac{n}{2}}\Sigma_2)^{-1}\mu_1 + \frac{\beta_2}{2\tau_2}\sqrt{\frac{n}{2}}\gamma_2\alpha\mathbf{w}_0^T(I + \frac{\beta_2}{2\tau_2}\sqrt{\frac{n}{2}}\Sigma_1 + \frac{\beta_2}{2\tau_2}\sqrt{\frac{n}{2}}\Sigma_2)^{-1}\mu_2$$

$$-2r\int\int p(s_1,s_2)\frac{\frac{n\beta_1\beta_2\gamma_1\gamma_2}{32\tau_1\tau_2}}{1 + \frac{\beta_1}{2\tau_1}\sqrt{\frac{n}{2}}s_1 + \frac{\beta_2}{2\tau_2}\sqrt{\frac{n}{2}}s_2}ds_1ds_2$$

Note that

$$\alpha^2 \mathbf{w}_0^T (I + \frac{\beta_1}{2\tau_1}\sqrt{\frac{n}{2}}\Sigma_1 + \frac{\beta_2}{2\tau_2}\sqrt{\frac{n}{2}}\Sigma_2)^{-1}\mathbf{w}_0 \to \alpha^2 \text{Tr}(\mathbf{R}_0(I + \frac{\beta_1}{2\tau_1}\sqrt{\frac{n}{2}}\Sigma_1 + \frac{\beta_2}{2\tau_2}\sqrt{\frac{n}{2}}\Sigma_2)^{-1})$$

Moreover

$$\text{Tr}(\mathbf{R}_0(I + \frac{\beta_1}{2\tau_1}\sqrt{\frac{n}{2}}\Sigma_1 + \frac{\beta_2}{2\tau_2}\sqrt{\frac{n}{2}}\Sigma_2)^{-1}) \to d \int \int p(s_1, s_2) \frac{\phi(s_1, s_2)}{1 + \frac{\beta_1}{2\tau_1}\sqrt{\frac{n}{2}}s_1 + \frac{\beta_2}{2\tau_2}\sqrt{\frac{n}{2}}s_2} ds_1 ds_2$$

And

$$\mathbf{w}_0^T (I + \frac{\beta_1}{2\tau_1}\sqrt{\frac{n}{2}}\Sigma_1 + \frac{\beta_2}{2\tau_2}\sqrt{\frac{n}{2}}\Sigma_2)^{-1}\mu_1 \to \frac{t_*(1-r)}{d}\text{Tr}((I + \frac{\beta_1}{2\tau_1}\sqrt{\frac{n}{2}}\Sigma_1 + \frac{\beta_2}{2\tau_2}\sqrt{\frac{n}{2}}\Sigma_2)^{-1}(\Sigma_1 + \Sigma_2)^{-1})$$

The latter term can be derived in the integral form as:

$$\text{Tr}\left((I + \frac{\beta_1}{2\tau_1}\sqrt{\frac{n}{2}}\Sigma_1 + \frac{\beta_2}{2\tau_2}\sqrt{\frac{n}{2}}\Sigma_2)^{-1}(\Sigma_1 + \Sigma_2)^{-1}\right) \to d \int \int p(s_1, s_2) \frac{(s_1 + s_2)^{-1}}{1 + \frac{\beta_1}{2\tau_1}\sqrt{\frac{n}{2}}s_1 + \frac{\beta_2}{2\tau_2}\sqrt{\frac{n}{2}}s_2} ds_1 ds_2$$

Since $p(s_1, s_2)$ is exchangeable in its arguments, we see that the latter objective remains invariant under the transformation that swaps $(\beta_1, \tau_1, \gamma_1)$ with $(\beta_2, \tau_2, -\gamma_2)$. Therefore, $\beta := \beta_1 = \beta_2$, $\tau := \tau_1 = \tau_2$ and $\gamma := \gamma_2 = -\gamma_1$ holds at the optimal point. The objective then reduces to the following:

$$\min_{\tau \geq 0} \max_{\beta \geq 0, \gamma} \sqrt{\frac{n}{2}}\beta\tau + \sqrt{\frac{n}{2}}\frac{\beta}{\tau}(\gamma - \frac{\gamma^2}{4}) - \int \int p(s_1, s_2) \frac{d\alpha^2\phi(s_1, s_2) + d\frac{(s_1+s_2)\beta^2}{4} + \frac{n\beta^2\gamma^2}{16\tau^2}(1-r)}{1 + \frac{\beta}{2\tau}(s_1 + s_2)\sqrt{\frac{n}{2}}} ds_1 ds_2 \tag{36}$$

$$- 2t_*(1-r)\alpha\sqrt{\frac{n}{2}} \int \int p(s_1, s_2) \frac{\frac{\beta\gamma}{2\tau}(s_1 + s_2)^{-1}}{1 + \frac{\beta}{2\tau}\sqrt{\frac{n}{2}}(s_1 + s_2)} \tag{37}$$

Note that $\gamma$ can be found explicitly in terms of $\beta$ and $\tau$ as the optimization over $\gamma$ is quadratic. Denote $e_0 = \frac{\|w_0\|^2}{d}$. To facilitate derivations, let us introduce a few notations. Let

$$F_1(\frac{\beta}{\tau}, \gamma) := \int \int p(s_1, s_2) \frac{d\alpha^2\phi(s_1, s_2) + \frac{n\beta^2\gamma^2}{16\tau^2}(1-r) + 2t_*(1-r)\alpha\sqrt{\frac{n}{2}}\frac{\beta\gamma}{2\tau}(s_1 + s_2)^{-1}}{1 + \frac{\beta}{2\tau}(s_1 + s_2)\sqrt{\frac{n}{2}}} ds_1 ds_2$$

$$F_2(\frac{\beta}{\tau}, \beta) := d \int \int p(s_1, s_2) \frac{\frac{(s_1+s_2)\beta^2}{4}}{1 + \frac{\beta}{2\tau}\sqrt{\frac{n}{2}}(s_1 + s_2)} ds_1 ds_2$$

Thus the objective is

$$\min_{\tau \geq 0} \max_{\beta \geq 0, \gamma} -F_1(\frac{\beta}{\tau}, \gamma) - F_2(\frac{\beta}{\tau}, \beta) + \sqrt{\frac{n}{2}}\beta\tau + \sqrt{\frac{n}{2}}\frac{\beta}{\tau}(\gamma - \frac{\gamma^2}{4})$$

Now taking derivatives w.r.t $\tau$ yields

$$\partial_\tau F_1(\frac{\beta}{\tau}, \gamma) = -\frac{\beta}{\tau^2}\partial_x F_1(x, \gamma)$$

$$\partial_\tau F_2(\frac{\beta}{\tau}, \beta) = -\frac{\beta}{\tau^2}\partial_x F_2(x, \beta)$$

Now for $\beta$:

$$\partial_\beta F_1(\frac{\beta}{\tau}, \gamma) = \frac{1}{\tau}\partial_x F_1(x, \gamma)$$

$$\partial_\beta F_2(\frac{\beta}{\tau}, \beta) = \frac{2}{\beta}F_2(\frac{\beta}{\tau}, \beta) + \frac{1}{\tau}\partial_x F_2(x, \beta)$$

Setting the derivative of the objective w.r.t $\tau$ to 0:

$$\frac{\beta}{\tau^2}\partial_x F_1(x, \gamma) + \frac{\beta}{\tau^2}\partial_x F_2(x, \beta) + \sqrt{\frac{n}{2}}\beta - \sqrt{\frac{n}{2}}\frac{\beta}{\tau^2}(\gamma - \frac{\gamma^2}{4}) = 0$$

Multiplying by $\tau^2$ and dropping $\beta$ yields:

$$\partial_x F_1(x, \gamma) + \partial_x F_2(x, \beta) + \sqrt{\frac{n}{2}}\tau^2 - \sqrt{\frac{n}{2}}(\gamma - \frac{\gamma^2}{4}) = 0 \tag{38}$$

Derivative w.r.t. $\beta$

$$-\frac{1}{\tau}\partial_x F_1(x, \gamma) - \frac{2}{\beta}F_2(\frac{\beta}{\tau}, \beta) - \frac{1}{\tau}\partial_x F_2(x, \beta) + \sqrt{\frac{n}{2}}\tau + \sqrt{\frac{n}{2}}\frac{1}{\tau}(\gamma - \frac{\gamma^2}{4}) = 0$$

Multiplying by $\tau$,

$$-\partial_x F_1(x, \gamma) - \frac{2\tau}{\beta}F_2(\frac{\beta}{\tau}, \beta) - \partial_x F_2(x, \beta) + \sqrt{\frac{n}{2}}\tau^2 + \sqrt{\frac{n}{2}}(\gamma - \frac{\gamma^2}{4}) = 0$$

Therefore the set of equations is:

$$\begin{cases} \partial_x F_1(x, \gamma) + \partial_x F_2(x, \beta) + \sqrt{\frac{n}{2}}\tau^2 - \sqrt{\frac{n}{2}}(\gamma - \frac{\gamma^2}{4}) = 0 \\ -\partial_x F_1(x, \gamma) - \frac{2\tau}{\beta}F_2(\frac{\beta}{\tau}, \beta) - \partial_x F_2(x, \beta) + \sqrt{\frac{n}{2}}\tau^2 + \sqrt{\frac{n}{2}}(\gamma - \frac{\gamma^2}{4}) = 0 \end{cases}$$

Summing the equations yields

$$-\frac{2\tau}{\beta}F_2(\frac{\beta}{\tau}, \beta) + 2\sqrt{\frac{n}{2}}\tau^2 = 0 \rightarrow \frac{2}{\beta}F_2(\frac{\beta}{\tau}, \beta) = 2\sqrt{\frac{n}{2}}\tau$$

Which implies

$$d \int \int p(s_1, s_2) \frac{\frac{(s_1+s_2)\beta}{2}}{1 + \frac{\beta}{2\tau}\sqrt{\frac{n}{2}}(s_1 + s_2)} ds_1 ds_2 = 2\sqrt{\frac{n}{2}}\tau$$

Multiplying both sides by $\frac{\sqrt{\frac{n}{2}}}{d\tau}$ and introducing $\theta = \frac{\beta}{\tau}$ we obtain:

$$\int \int p(s_1, s_2) \frac{\frac{\theta}{2}\sqrt{\frac{n}{2}}(s_1 + s_2)}{1 + \frac{\theta}{2}\sqrt{\frac{n}{2}}(s_1 + s_2)}) ds_1 ds_2 = \frac{n}{d}$$

This can be rewritten as:

$$\int \int p(s_1, s_2)(1 - \frac{2\sqrt{2}}{2\sqrt{2} + \theta\sqrt{n}(s_1 + s_2)}) ds_1 ds_2 = \frac{n}{d}$$

We arrive at the following equation for $\theta$:

$$\frac{d-n}{d} = \int \int p(s_1, s_2) \frac{2\sqrt{2}}{2\sqrt{2} + \theta\sqrt{n}(s_1 + s_2)} ds_1 ds_2 \tag{39}$$

The last equation above can be reformulated in terms of the Stiltjes transform as follows:

$$\frac{2\sqrt{2}}{\theta\sqrt{n}}S_{\Sigma_1+\Sigma_2}(-\frac{2\sqrt{2}}{\theta\sqrt{n}}) = \frac{d-n}{d}$$

After finding $\theta$ from the equation above, we proceed to identify the optimal value of $\gamma$. We will do so by taking the derivative of (36) by $\gamma$ and setting it to 0:

$$\sqrt{\frac{n}{2}}\theta(1-\frac{\gamma}{2}) - \int\int p(s_1,s_2)\frac{\frac{n\theta^2\gamma}{8}(1-r)}{1+\frac{\theta}{2}(s_1+s_2)\sqrt{\frac{n}{2}}}ds_1ds_2 - \alpha\sqrt{\frac{n}{2}}\int\int p(s_1,s_2)\frac{\theta t_*(1-r)(s_1+s_2)^{-1}}{1+\frac{\theta}{2}\sqrt{\frac{n}{2}}(s_1+s_2)} = 0$$

Dividing both sides by $\theta$, factoring $\sqrt{n}$ out and using (39) we obtain:

$$\gamma\sqrt{n}\left(\frac{1}{2\sqrt{2}} + \frac{\sqrt{n}\theta(1-r)}{8}\frac{d-n}{d}\right) = \sqrt{\frac{n}{2}} - \alpha\sqrt{\frac{n}{2}}\int\int p(s_1,s_2)\frac{t_*(1-r)(s_1+s_2)^{-1}}{1+\frac{\theta}{2}\sqrt{\frac{n}{2}}(s_1+s_2)}$$

Hence, we can find $\gamma$ via

$$\gamma = \left(\frac{1}{2} + \frac{\sqrt{2n}\theta(1-r)(d-n)}{8d}\right)^{-1}\left(1 - \alpha\int\int p(s_1,s_2)\frac{t_*(1-r)(s_1+s_2)^{-1}}{1+\frac{\theta}{2}\sqrt{\frac{n}{2}}(s_1+s_2)}\right)$$

Finally, to find $\tau$, recall (40)

$$\partial_x F_1(x,\gamma) + \partial_x F_2(x,\beta) + \sqrt{\frac{n}{2}}\tau^2 - \sqrt{\frac{n}{2}}(\gamma - \frac{\gamma^2}{4}) = 0$$

Using $F_2$ is quadratic in its second argument we have:

$$\partial_x F_1(x,\gamma) + \tau^2(\partial_x F_2(x,\theta) + \sqrt{\frac{n}{2}}) = \sqrt{\frac{n}{2}}(\gamma - \frac{\gamma^2}{4})$$

$$\tau^2(\partial_x F_2(x,\theta) + \sqrt{\frac{n}{2}}) = \sqrt{\frac{n}{2}}(\gamma - \frac{\gamma^2}{4}) - \partial_x F_1(x,\gamma)$$

$$\tau^2 = (\partial_x F_2(x,\theta) + \sqrt{\frac{n}{2}})^{-1}(\sqrt{\frac{n}{2}}(\gamma - \frac{\gamma^2}{4}) - \partial_x F_1(x,\gamma))$$

We have

$$F_1(x,\gamma) := \int\int p(s_1,s_2)\frac{d\alpha^2\phi(s_1,s_2) + \frac{nx^2\gamma^2}{16}(1-r) + 2t_*(1-r)\alpha\sqrt{\frac{n}{2}}\frac{x\gamma}{2}(s_1+s_2)^{-1}}{1+\frac{x}{2}(s_1+s_2)\sqrt{\frac{n}{2}}}ds_1ds_2$$

Then

$$\partial_x F_1(x,\gamma) = -\frac{d\alpha^2}{2}\sqrt{\frac{n}{2}}\int\int p(s_1,s_2)\frac{\phi(s_1,s_2)(s_1+s_2)}{(1+\frac{x}{2}(s_1+s_2)\sqrt{\frac{n}{2}})^2}ds_1ds_2$$

$$+\frac{nx\gamma^2(1-r)}{8}\int\int p(s_1,s_2)\frac{1}{1+\frac{x}{2}(s_1+s_2)\sqrt{\frac{n}{2}}}ds_1ds_2$$

$$-\frac{nx^2\gamma^2(1-r)}{32}\sqrt{\frac{n}{2}}\int\int p(s_1,s_2)\frac{(s_1+s_2)}{(1+\frac{x}{2}(s_1+s_2)\sqrt{\frac{n}{2}})^2}ds_1ds_2$$

$$-\alpha t_*(1-r)\gamma\sqrt{\frac{n}{2}}\int\int p(s_1,s_2)\frac{(s_1+s_2)^{-1}}{1+\frac{x}{2}(s_1+s_2)\sqrt{\frac{n}{2}}}ds_1ds_2$$

$$+\frac{\alpha t_*(1-r)x\gamma}{2}\frac{n}{2}\int\int p(s_1,s_2)\frac{1}{(1+\frac{x}{2}(s_1+s_2)\sqrt{\frac{n}{2}})^2}ds_1ds_2$$

Moreover

$$F_2(x,\theta) = d \int \int p(s_1,s_2) \frac{\frac{(s_1+s_2)\theta^2}{4}}{1 + \frac{x}{2}\sqrt{\frac{n}{2}}(s_1+s_2)} ds_1 ds_2$$

Then

$$\partial_x F_2(x,\theta) = -\frac{d\theta^2}{8}\sqrt{\frac{n}{2}} \int \int p(s_1,s_2) \frac{(s_1+s_2)^2}{(1 + \frac{x}{2}\sqrt{\frac{n}{2}}(s_1+s_2))^2} ds_1 ds_2$$

Thus, all in all:

$$\tau^2 = \left( \sqrt{\frac{n}{2}} - \frac{d\theta^2}{8}\sqrt{\frac{n}{2}} \int \int p(s_1,s_2) \frac{(s_1+s_2)^2}{(1 + \frac{\theta}{2}\sqrt{\frac{n}{2}}(s_1+s_2))^2} ds_1 ds_2 \right)^{-1}$$

$$\left[ \sqrt{\frac{n}{2}}(\gamma - \frac{\gamma^2}{4}) + \frac{d\alpha^2}{2}\sqrt{\frac{n}{2}} \int \int p(s_1,s_2) \frac{\phi(s_1,s_2)(s_1+s_2)}{(1 + \frac{\theta}{2}(s_1+s_2)\sqrt{\frac{n}{2}})^2} ds_1 ds_2 \right.$$

$$- \frac{n\theta\gamma^2(1-r)}{8} \int \int p(s_1,s_2) \frac{1}{1 + \frac{\theta}{2}(s_1+s_2)\sqrt{\frac{n}{2}}} ds_1 ds_2$$

$$+ \frac{n\theta^2\gamma^2(1-r)}{32}\sqrt{\frac{n}{2}} \int \int p(s_1,s_2) \frac{(s_1+s_2)}{(1 + \frac{\theta}{2}(s_1+s_2)\sqrt{\frac{n}{2}})^2} ds_1 ds_2$$

$$- \alpha t_*(1-r)\gamma\sqrt{\frac{n}{2}} \int \int p(s_1,s_2) \frac{(s_1+s_2)^{-1}}{1 + \frac{\theta}{2}(s_1+s_2)\sqrt{\frac{n}{2}}} ds_1 ds_2$$

$$+ \left. \frac{\alpha t_*(1-r)\theta\gamma n}{4} \int \int p(s_1,s_2) \frac{1}{(1 + \frac{\theta}{2}(s_1+s_2)\sqrt{\frac{n}{2}})^2} ds_1 ds_2 \right]$$

$$\gamma = \left( \frac{1}{2} + \frac{\sqrt{2n}\theta(1-r)(d-n)}{8d} \right)^{-1} \left( 1 - \alpha \int \int p(s_1,s_2) \frac{t_*(1-r)(s_1+s_2)^{-1}}{1 + \frac{\theta}{2}\sqrt{\frac{n}{2}}(s_1+s_2)} \right) \quad (40)$$

The generalization error can be found through $Q\left( \frac{1 - \frac{\gamma}{2}}{\sqrt{\tau^2 - (\frac{\gamma}{2})^2}} \right)$.

## I.1   Single direction case.

In the special case when $\Sigma_1 = \Sigma_2 = \sigma^2 \frac{\mathbf{I}_d}{d}$ we obtain as $\mathbf{R}_0 = \frac{t_\eta^2(1-r)}{d}I_d + 2\frac{t_*^2}{d}(1-r)(\Sigma_1 + \Sigma_2)^{-2}$:

$$\alpha = \frac{\mathbf{y}^T\mathbf{X}\mathbf{w}_0}{\|\mathbf{X}\mathbf{w}_0\|^2} = \frac{\frac{t_*(1-r)}{d}\mathrm{Tr}(\Sigma_1+\Sigma_2)^{-1}}{\frac{1}{2}\mathrm{Tr}(\mathbf{R}_0(\Sigma_1+\Sigma_2)) + \frac{t_*^2(1-r)^2}{d^2}\mathrm{Tr}^2(\Sigma_1+\Sigma_2)^{-1}}$$

$$= \frac{t_*\frac{d}{2\sigma^2}}{\frac{1}{d}\sigma^2(t_\eta^2 + t_*^2\frac{d^2}{2\sigma^4}) + t_*^2(1-r)\frac{d^2}{4\sigma^4}}$$

We also have

$$\alpha = \frac{t_*(1-r)\frac{d}{2\sigma^2}}{\frac{t_\eta^2(1-r)}{2d}\mathrm{Tr}\frac{2\sigma^2}{d}\mathbf{I} + \frac{t_*^2}{d}(1-r)\mathrm{Tr}(\frac{2\sigma^2}{d}\mathbf{I})^{-1} + \frac{t_*^2(1-r)^2}{d^2}\mathrm{Tr}^2(\frac{2\sigma^2}{d}\mathbf{I})^{-1}}$$

$$= \frac{t_*(1-r)\frac{d}{2\sigma^2}}{\frac{t_\eta^2(1-r)\sigma^2}{d} + \frac{dt_*^2(1-r)}{2\sigma^2} + \frac{t_*^2(1-r)^2 d^2}{4\sigma^4}} = \frac{t_*\frac{d}{2\sigma^2}}{\frac{t_\eta^2\sigma^2}{d} + \frac{dt_*^2}{2\sigma^2} + \frac{t_*^2(1-r)d^2}{4\sigma^4}} = \frac{t_*\frac{d}{2\sigma^2}}{\frac{\sigma^2}{d}(t_\eta^2 + t_*^2\frac{d^2}{2\sigma^4}) + t_*^2(1-r)\frac{d^2}{4\sigma^4}}$$

Multiplying both sides by $t^*(1-r)\frac{d}{2\sigma^2}$ we arrive at:

$$\Gamma := \alpha t^*(1-r)\frac{d}{2\sigma^2} = \frac{1}{\frac{\sigma^2}{d(1-r)}\left(\frac{t_\eta^2}{t_*^2}\frac{4\sigma^4}{d^2}+2\right)+1} \tag{41}$$

Note that plugging in $\mathbf{w}_0 = t_*(\mathbf{\Sigma}_1+\mathbf{\Sigma}_2)^{-1}(\boldsymbol{\mu}_1-\boldsymbol{\mu}_2)+t_\eta\boldsymbol{\eta}$ for $\mathbf{\Sigma}_1=\mathbf{\Sigma}_2=\frac{\sigma^2}{d}\mathbf{I}_d$ into the expression for the generalization error yields

$$e_a = Q\left(\frac{\boldsymbol{\mu}_1^T\mathbf{w}_0}{\sqrt{\mathbf{w}_0^T\mathbf{\Sigma}_1\mathbf{w}_0}}\right) = Q\left(\frac{t_*(1-r)\frac{d}{2\sigma^2}}{\sqrt{t_*^2(1-r)\frac{d}{2\sigma^2}+\frac{\sigma^2}{d}t_\eta^2(1-r)}}\right)$$

Taking $Q^{-1}$ of both sides and then squaring them we arrive at:

$$Q^{-1}(e_a)^2 = \frac{t_*^2(1-r)^2\frac{d^2}{4\sigma^4}}{t_*^2(1-r)\frac{d}{2\sigma^2}+\frac{\sigma^2}{d}t_\eta^2(1-r)} = \frac{t_*^2(1-r)^2\frac{d^2}{4\sigma^4}}{t_*^2(1-r)\frac{d}{2\sigma^2}+\frac{\sigma^2}{d}t_\eta^2(1-r)} = \frac{1}{\frac{\sigma^2}{(1-r)d}\left(2+4\frac{\sigma^4}{d^2}\frac{t_\eta^2}{t_*^2}\right)}$$

This yields that by taking $\rho := \frac{d(1-r)}{\sigma^2}$

$$\frac{t_\eta}{t_*} = \frac{d}{2\sigma^2}\sqrt{\frac{\rho}{Q^{-1}(e_a)^2}-2} = \frac{\rho}{2(1-r)}\sqrt{\frac{\rho}{Q^{-1}(e_a)^2}-2}$$

We thus obtain from (41):

$$\Gamma = \frac{1}{Q^{-1}(e_a)^{-2}+1} = \frac{1}{Q^{-1}(e_a)^{-2}+1} = \frac{Q^{-1}(e_a)^2}{Q^{-1}(e_a)^2+1} = 1-\frac{1}{Q^{-1}(e_a)^2+1} \tag{42}$$

We proceed to find $\theta$ next. From the general case we had

$$\frac{2\sqrt{2}}{\theta\sqrt{n}}S_{\mathbf{\Sigma}_1+\mathbf{\Sigma}_2}\left(-\frac{2\sqrt{2}}{\theta\sqrt{n}}\right) = \frac{d-n}{d}$$

Now plugging for the distribution of $\mathbf{\Sigma}_1,\mathbf{\Sigma}_2$ yields

$$1+\theta\sqrt{\frac{n}{2}}\frac{\sigma^2}{d} = \frac{d}{d-n}$$

$$\theta\sqrt{\frac{n}{2}}\frac{\sigma^2}{d} = \frac{n}{d-n}$$

$$\theta = \frac{d\sqrt{2n}}{(d-n)\sigma^2}$$

Using (41) we can simplify the last expression above:

$$\gamma = \left(\frac{1}{2}+\frac{\sqrt{2n}\theta(1-r)(d-n)}{8d}\right)^{-1}\left(1-\alpha\int\int p(s_1,s_2)\frac{t_*(1-r)(s_1+s_2)^{-1}}{1+\frac{\theta}{2}\sqrt{\frac{n}{2}}(s_1+s_2)}\right) =$$

$$= \left(\frac{1}{2}+\frac{\sqrt{2n}\theta(1-r)(d-n)}{8d}\right)^{-1}\left(1-\frac{\Gamma}{1+\frac{\theta}{2}\sqrt{\frac{n}{2}}(s_1+s_2)}\right) =$$

$$\left(\frac{1}{2}+\frac{\sqrt{2n}\frac{d\sqrt{2n}}{(d-n)\sigma^2}(1-r)(d-n)}{8d}\right)^{-1}\left(1-\frac{\Gamma}{1+\frac{d\sqrt{2n}}{2(d-n)\sigma^2}\sqrt{\frac{n}{2}}\frac{2\sigma^2}{d}}\right) =$$

Thus we can write for $\gamma$

$$\gamma = \left(\frac{1}{2}+\frac{n\frac{2}{\sigma^2}(1-r)}{8}\right)^{-1}\left(1-\frac{\Gamma}{1+\frac{n}{d-n}}\right)$$

$$= \left(\frac{1}{2}+\frac{n(1-r)}{4\sigma^2}\right)^{-1}\left(1-\frac{d-n}{d}\left(1-\frac{1}{Q^{-1}(e_a)^2+1}\right)\right) = \frac{\frac{n}{d}+\frac{d-n}{d}\frac{1}{Q^{-1}(e_a)^2+1}}{\frac{1}{2}+\frac{\rho}{4\kappa}} \quad \text{where } \rho := \frac{d(1-r)}{\sigma^2}$$

Furthermore, we have for $\tau^2$

$$\tau^2 = \left(\sqrt{\frac{n}{2}} - \frac{d\theta^2}{8}\sqrt{\frac{n}{2}}\int\int p(s_1,s_2)\frac{(s_1+s_2)^2}{(1+\frac{\theta}{2}\sqrt{\frac{n}{2}}(s_1+s_2))^2}ds_1ds_2\right)^{-1}$$

$$\left[\sqrt{\frac{n}{2}}(\gamma-\frac{\gamma^2}{4})+\frac{d\alpha^2}{2}\sqrt{\frac{n}{2}}\int\int p(s_1,s_2)\frac{\phi(s_1,s_2)(s_1+s_2)}{(1+\frac{\theta}{2}(s_1+s_2)\sqrt{\frac{n}{2}})^2}ds_1ds_2\right.$$

$$-\frac{n\theta\gamma^2(1-r)}{8}\int\int p(s_1,s_2)\frac{1}{1+\frac{\theta}{2}(s_1+s_2)\sqrt{\frac{n}{2}}}ds_1ds_2$$

$$+\frac{n\theta^2\gamma^2(1-r)}{32}\sqrt{\frac{n}{2}}\int\int p(s_1,s_2)\frac{(s_1+s_2)}{(1+\frac{\theta}{2}(s_1+s_2)\sqrt{\frac{n}{2}})^2}ds_1ds_2$$

$$-\alpha t_*(1-r)\gamma\sqrt{\frac{n}{2}}\int\int p(s_1,s_2)\frac{(s_1+s_2)^{-1}}{1+\frac{\theta}{2}(s_1+s_2)\sqrt{\frac{n}{2}}}ds_1ds_2$$

$$\left.+\frac{\alpha t_*(1-r)\theta\gamma n}{4}\int\int p(s_1,s_2)\frac{1}{(1+\frac{\theta}{2}(s_1+s_2)\sqrt{\frac{n}{2}})^2}ds_1ds_2\right]=$$

Simplifying furthermore yields

$$\tau^2 = \left(1 - \frac{d}{n}\int\int p(s_1,s_2)\frac{\frac{n^2}{(d-n)^2}}{\frac{d^2}{(d-n)^2}}ds_1ds_2\right)^{-1}$$

$$\left[(\gamma-\frac{\gamma^2}{4})+\int\int\frac{\alpha^2 p(s_1,s_2)(1-r)^2(t_\eta^2+t_*^2\frac{d^2}{2\sigma^4})\sigma^2}{d(1-r)(\frac{d}{d-n})^2}ds_1ds_2\right.$$

$$-\frac{n\frac{2d}{(d-n)\sigma^2}\gamma^2(1-r)}{8}\int\int p(s_1,s_2)\frac{1}{\frac{d}{d-n}}ds_1ds_2$$

$$+\frac{n\frac{2nd^2}{(d-n)^2\sigma^4}\gamma^2(1-r)}{32}\int\int p(s_1,s_2)\frac{\frac{2\sigma^2}{d}}{(\frac{d}{d-n})^2}ds_1ds_2$$

$$-\alpha t_*(1-r)\gamma\int\int p(s_1,s_2)\frac{\frac{d}{2\sigma^2}}{\frac{d}{d-n}}ds_1ds_2$$

$$\left.+\frac{\alpha t_*(1-r)\frac{2d}{(d-n)\sigma^2}\gamma n}{4}\int\int p(s_1,s_2)\frac{1}{(\frac{d}{d-n})^2}ds_1ds_2\right]=$$

Further manipulation leads to

$$\tau^2 = \frac{\kappa}{\kappa-1}\left[\gamma-\frac{\gamma^2}{4}+\frac{\Gamma^2}{Q^{-1}(e_a)^2}(1-\frac{1}{\kappa})^2-\frac{\rho\gamma^2}{4\kappa}+\frac{\rho\gamma^2}{8\kappa^2}-\Gamma\gamma(1-\frac{1}{\kappa})+(1-\frac{1}{\kappa})\frac{1}{\kappa}\Gamma\gamma\right]$$

$$=\frac{\kappa}{\kappa-1}\left[\gamma-\frac{\gamma^2}{4}+\frac{\Gamma^2}{Q^{-1}(e_a)^2}(1-\frac{1}{\kappa})^2-\frac{\rho\gamma^2}{4\kappa}+\frac{\rho\gamma^2}{8\kappa^2}-(1-\frac{1}{\kappa})^2\Gamma\gamma\right]$$

$$=\frac{\kappa}{\kappa-1}\left[\left(\frac{\rho}{8\kappa^2}-\frac{\rho}{4\kappa}-\frac{1}{4}\right)\gamma^2+\left(1-(1-\frac{1}{\kappa})^2\Gamma\right)\gamma+\frac{\Gamma^2}{Q^{-1}(e_a)^2}(1-\frac{1}{\kappa})^2\right]$$

Recall that $\Gamma = 1 - \frac{1}{Q^{-1}(e_a)^2+1}$ and $\frac{\Gamma}{Q^{-1}(e_a)^2} = 1-\Gamma$. Hence, we obtain:

$$\tau^2 = \frac{\kappa}{\kappa-1}\left[\left(\frac{\rho}{8\kappa^2}-\frac{\rho}{4\kappa}-\frac{1}{4}\right)\gamma^2+\left(1-(1-\frac{1}{\kappa})^2\Gamma\right)\gamma+\Gamma(1-\Gamma)(1-\frac{1}{\kappa})^2\right] \qquad (43)$$

On the other hand, $\gamma = \frac{1-(1-\frac{1}{\kappa})\Gamma}{\frac{1}{2}+\frac{\rho}{4\kappa}}$ and $\Gamma = 1 - \frac{1}{Q^{-1}(e_a)^2+1}$ combining these fact with 43 yields

$$e_p = Q\left(\frac{\left(Q^{-1}(e_a)^2(2\kappa+\rho-2)+\rho\right)\sqrt{\kappa(\kappa-1)}}{\sqrt{(2\kappa+\rho)((4\kappa-2)Q^{-1}(e_a)^4+Q^{-1}(e_a)^2\left(2\kappa^3+\kappa^2\rho-2\kappa(\rho-1)+\rho\right)+2\kappa^2)}}\right)$$

□

