# OpenReview forum: "Universality in Transfer Learning for Linear Models"
_NeurIPS.cc/2024/Conference — NeurIPS 2024 poster_

### Official Review · Reviewer_hU8n · 2024-06-28

**Soundness:** 3
**Presentation:** 3
**Contribution:** 2
**Rating:** 6
**Confidence:** 3

**Summary:**

The paper derives a new universality result, which can be leveraged to solve a mirror descent optimization problem for a large family of data distributions by relating the solution to a Gaussian distribution with matching first and second-order statistics. This result is then applied to analyze transfer learning for linear models in the setting of regression and binary classification.

**Strengths:**

The paper derives a new universality result by adapting an existing method. The universality is used to extend results about transfer learning in linear regression models to a larger class of data distributions, and further results are developed in the setting of binary classification. The paper is overall well-written and easy to follow.

**Weaknesses:**

Parts of Assumptions 2 and 3 seem unintuitive to me, see the questions below.

**Questions:**

In assumption 2.2, why should the covariance depend on the mean $\mu$?

In assumption 2.3 and 3, how can the condition hold for any matrix of bounded operator norm? It seems possible to construct a matrix for which the condition does not hold.

**Limitations:**

Limitations are adequately addressed.

---

> ### Author Rebuttal · Authors · 2024-08-07
>
> We would like to thank the reviewer for their comments and suggestions to help us improve the quality of our work. Below we address the questions and other points raised in the review one by one.
>
> **In assumption 2.2, why should the covariance depend on the mean $\mu$**
>
> We apologize for the confusion. Assumption 2.2 contains a typo that we will make sure to fix in the final version. It should indeed be $\mathcal{N}(a^T\mu, a^T\Sigma a)$. We would also like to remark that this typo was contained only in assumption 2.2 and did not affect our proofs and calculations.
>
> **In assumption 2.3 and 3, how can the condition hold for any matrix of bounded operator norm? It seems possible to construct a matrix for which the condition does not hold.**
>
> We will illustrate this property in the case when $\eta \sim \mathcal{N}(0, \frac{I_d}{d})$ so that asymptotically $\lVert \eta \rVert \rightarrow 1$. One can then first replace $C$ with $\frac{C + C^T}{2}$ making it symmetric and then diagonalize the latter, meaning we can replace $C$ by $diag(\lambda_1, \dots, \lambda_d)$ where all $|\lambda_i|$ are bounded by a constant $K>0$. Then $\eta^TC\eta$ has mean $\frac{\lambda_1 + \dots + \lambda_d}{d} = \frac{Tr(C)}{d}$ and variance $\frac{2(\lambda_1^2 + \dots + \lambda_d^2)}{d^2} \le \frac{2K^2}{d}$. The latter implies that the variance goes to zero,  and therefore the difference $\eta^TC\eta - \frac{Tr(C)}{d}$ is close to zero with high probability. As it might be a bit counter intuitive, let us consider the case of low rank $C$ in more detail , for example $C= e_1e_1^T$, where $e_1$ is the first basis vector. Then $\eta^TC\eta$ is a $\chi^2$ random variable but it still converges to $0$ because of its normalization. Note that $\frac{Tr(C)}{d} = \frac{1}{d}$ also goes to $0$ and thus both quantities $\eta^TC\eta$ and $\frac{Tr(C)}{d}$ go to zero as $d \to \infty$.

---

> > ### Comment · Reviewer_hU8n · 2024-08-07
> >
> > We thank the authors for their response.
> > - Thanks for the clarification on assumption 2.2.
> > - I am still unsure about a few points about assumption 2.3 and 3, perhaps it is just a misunderstanding on notation/vocabulary. Based on the reviewer's response, it seems as if we need to choose a specific sequence of matrices $C_1, C_2, \dots$ for each dimension $d$ as we take $d \to \infty$. Could the reviewer kindly provide formal mathematical statement corresponding to these assumptions? If this can be clarified, I would be happy to raise my score.

---

> > > ### Author Response · Authors · 2024-08-08
> > >
> > > We thank the reviewer for their very constructive comment.
> > >
> > > We are sorry for formulating these assumptions somewhat  loosely and would like to thank the reviewer for bringing it to our attention. We will make sure to resolve this in the final version. First, our universality result is asymptotic and holds when $d \to \infty$ (albeit in simulations we already see a very close match between the generalization errors for the non-Gaussian data and the matching Gaussian data in the sense of Definition 3 when $d$ is on the order of hundreds). That is, formally we assume that there is a growing sequence of dimensions $d_i \to \infty$ along with sequences of initialization points $w_0^{(i)} \in \mathbb{R}^{d_i}$ and random vectors $a^{(i)} \in \mathbb{R}^{d_i}$ satisfying certain technical properties.
> > > Then for a certain class of optimization objectives in each of these dimensions we evaluate their value on the data matrix $A^{(i)}$ sampled from $a^{(i)}$ and on its matching Gaussian matrix $G^{(i)}$ and show that the difference between the values goes to zero in expectation as $i \to \infty$. As Assumptions 2.3 and 3 are very similar, we will proceed to explain formally what we mean in Assumption 3. As explained earlier, we have a sequence of $w_0^{(i)} \in \mathbb{R}^{d_i}$ and thus a sequence of $\eta^{(i)} \in \mathbb{R}^{d_i}$ normalized according to $\mathbb{E}_{\eta^{(i)}} || \eta^{(i)} || ^ 2 = 1 - r$. Now Assumption 3 says:
> > >
> > > *Given any sequence of matrices $C_i \in \mathbb{R}^{d_i \times d_i}$ satisfying $||C_i||_{op} \le K$ for some constant $K>0$ and all $i>0$, it holds that ${\eta^{(i)}}^TC_i\eta^{(i)} - \frac{Tr(C_i)(1-r)}{d_i} \to 0$ in probability as $i \to \infty$.*
> > >
> > > As an example, note that Assumption $3$ holds for the sequence of Gaussian vectors $\eta^{(i)} \sim \mathcal{N}(0, \frac{(1-r)}{d_i}I_{d_i})$ for any sequence $d_i \to \infty$.

---

> > > > ### Comment · Reviewer_hU8n · 2024-08-08
> > > >
> > > > Thank you for the clarification. The formal specification of Assumptions 2.3 and 3 make sense to me and seem reasonable. However, I am not sure if it is strong enough for the proofs. In particular, the current statement requires the sequence of matrices $C_i$ to be established independently of $\mu$ or $\eta$, but it seems in the proofs that the matrices that are used for $C_i$ depend on $\mu$ (e.g. line 616). I did not go over every step of the proof in detail, feel free to correct me if I have missed anything.

---

> > > > > ### Author Response · Authors · 2024-08-09
> > > > >
> > > > > It is correct that $\tilde{s}_1\Sigma_1 + \tilde{s}_2\Sigma_2$ depends on $\mu_1$ and $\mu_2$ via $\tilde{s}_1$ and $\tilde{s}_2$. But what we used implicitly in the line 616 is that $\tilde{s}_1$ and $\tilde{s}_2$ converge in probability to some constants $c_1$ and $c_2$ and thus can be treated as constants for the purposes of the analysis. Informally, this is true because $\mu_1$ and $\mu_2$ are "generic" vectors and therefore, speaking asymptotically, the scalar quantities like $\mu_1^Tw$ and $w^T\Sigma_1w$ evaluated at the optimal $w$ do not depend on the realizations of $\mu_1$ and $\mu_2$.
> > > > >  To explain formally why $\tilde{s}_1$ and $\tilde{s}_2$ concentrate to constants, note that we showed that the optimal $w$ has to be of the form
> > > > > $$w(\tilde{s}_1, \tilde{s}_2, \tilde{t}_1, \tilde{t}_2) = (\tilde{s}_1\Sigma_1 + \tilde{s}_2\Sigma_2)^{-1}(\tilde{t}_1\mu_1 - \tilde{t}_2\mu_2)\tag{1}$$
> > > > > Moreover, any $w$ of this form is in the feasible set, since we do not have any constraints on $w$. Thus, the optimization problem of interest, namely
> > > > >  $$min_w \frac{1}{2}Q(\frac{\mu_1^Tw}{\sqrt{w^T \Sigma_1 w}}) + \frac{1}{2}Q(-\frac{\mu_2^Tw}{ \sqrt{w^T \Sigma_2 w}})$$
> > > > >  reduces to the following four-dimensional optimization problem, where
> > > > >
> > > > > $x = (\tilde{s}_1, \tilde{s}_2, \tilde{t}_1, \tilde{t}_2) \in \mathbb{R}^4$ and $w(x)$ is defined via equation *(1)*:
> > > > >
> > > > >    $$min_{x} \frac{1}{2}Q(\frac{\mu_1^Tw(x)}{\sqrt{w(x)^T \Sigma_1 w(x)}}) + \frac{1}{2}Q(-\frac{\mu_2^Tw(x)}{ \sqrt{w(x)^T \Sigma_2 w(x)}})\tag{2}$$
> > > > >   Now, we can apply Assumption 2.3 to every fixed four-dimensional vector $x$ inside the objective *(2)*, since $\Sigma_1, \Sigma_2$ are independent from $\mu_1, \mu_2$, and arrive at the following:
> > > > >    $$ \mu_1^Tw(x) = \frac{1}{d}Tr\left((x_1\Sigma_1 +x_2\Sigma_2)^{-1}(x_3 - x_4r)\right) \tag{3}$$
> > > > > $$\mu_2^Tw(x) = \frac{1}{d}Tr\left((x_1\Sigma_1 +x_2\Sigma_2)^{-1}(x_3r - x_4)\right)\tag{4}$$
> > > > > Plugging equations *(3)* and *(4)* into the objective *(2)* allows  us to rewrite the latter in terms of $x, \Sigma_1, \Sigma_2$ and  $r$ solely. Hence, $x$ converges to a deterministic four-dimensional vector in probability and therefore $\tilde{s}_1$ and $\tilde{s}_2$ converge to deterministic quantities as well.
> > > > >
> > > > >  We would like to thank the reviewer for letting us know that we omitted some details important for understanding the proof of Lemma 1. We will expand the current proof by adding the necessary clarifications and derivations in the final version.

---

> ### Comment · Reviewer_hU8n · 2024-08-09
>
> I am still not fully convinced of the current explanation. I understand (3) and (4) in the sense that for any fixed $x$, the LHS converges in probability to the RHS. However, equation (2) is a separate minimization problem for every $\mu$, and there is no guarantee that for a given $\mu$, equations (3) and (4) hold for all relevant values of $x$. Please let me know if there are any details I missed.

---

> > ### Author Response · Authors · 2024-08-10
> >
> > We would like to thank the reviewer for their thorough attention to details. This and the previous questions are instrumental for improving the readability of our  proofs.
> >
> > The key consideration in this argument is that $x$ has a fixed dimension.  Passing to limits in probability inside low-dimensional optimization problems is a common technical step in many papers applying Gaussian Comparison Inequalities to analyze problems in machine learning. There are two known ways to justify such steps formally: either by referring to different forms of the "convexity lemma" , such as Theorem II.1 from "Cox's Regression Model for Counting Processes: A Large Sample Study" by
> > Andersen and Gill or by constructing other covering arguments tailored to a specific problem of interest. The former normally suffices to justify exchanging  finite-dimensional optimizations and limits but cannot be applied directly in the proof of Lemma 1 because the $Q$-function is neither convex nor concave (albeit $Q(t)$ is convex when $t>0$ and concave otherwise, so maybe there is a smart way to use it here). It is also worth mentioning that the proof of Lemma 1 is the only proof in the paper where the convexity lemma does not suffice to ensure concentration of the encountered finite-dimensional objectives of interest. But another standard covering argument can be used in this case to swap the optimization of a fixed dimension with the limit in probability. Indeed, one can restrict $x$ to the unit sphere, since $w$ is defined uniquely only up to the direction and therefore so is $x$ due to the uniqueness of the correspondence between $x$ and $w(x)$. One would then proceed to take an $\epsilon$-net on the sphere, reduce the desired concentration bound for the objective to the union bound over the centers of the net due to the Lipschitzness of the objective assuming $\epsilon$ is taken to be small enough and deduce that this union bound goes to zero because it has a fixed number of terms each of which goes to zero.

---

> > > ### Comment · Reviewer_hU8n · 2024-08-10
> > >
> > > I would like to thank the author for taking the time to go over the details of the proof. I'm satisfied with the current explanation and have no further questions.

---

### Official Review · Reviewer_KpbC · 2024-07-03

**Soundness:** 2
**Presentation:** 3
**Contribution:** 3
**Rating:** 5
**Confidence:** 4

**Summary:**

The paper investigates the application of transfer learning within the framework of linear models. The authors focus on a model-based approach, where a model pre-trained on a source distribution is fine-tuned on a few samples from a target distribution. Authors extend the concept of universality, traditionally used in random matrix theory and high-dimensional statistics, to the context of transfer learning. They demonstrate that certain properties of linear models trained on large datasets can be transferred to new tasks with minimal data. They also provide a thorough theoretical analysis, establishing conditions under which the transfer learning approach achieves performance close to that of a model trained directly on the target task with abundant data.

**Strengths:**

Universality has traditionally been applied in random matrix theory and high-dimensional statistics, but not in the context of transfer learning. The authors show properties of linear models trained on large datasets can effectively transfer to new tasks with limited data. The authors explore the conditions under which their transfer learning approach is effective. Combination of theoretical analysis and empirical validation provides a comprehensive exploration of this concept.

**Weaknesses:**

1. The empirical validation, while convincing, is somewhat limited in scope. The experiments are conducted on a few specific tasks and datasets, which may not fully capture the diversity of real-world applications. Datasets with different characteristics, such as varying levels of noise, feature distributions, and dimensionalities, could provide a more comprehensive assessment of the proposed methods.
2. The comparison with existing transfer learning methods is limited. The paper could benefit from a more comprehensive comparison to highlight the advantages and limitations of the proposed approach relative to current state-of-the-art methods.
3. The paper provides certain properties of linear models trained on large datasets can be transferred to new tasks with minimal data, but in real applications, is there any indication, when transfer methods are useful?

**Questions:**

1. Can you provide more details on the boundary conditions under which the universality principle might fail? Are there specific cases or distributions where this principle does not hold?
2. Can you elaborate on the choice of datasets and tasks used for empirical validation? Are there plans to test the proposed method on more diverse and complex datasets?

**Limitations:**

While the authors provide thorough theoretical proofs and some experimental validation, the paper lacks comparative experiments with existing methods. This makes it difficult to fully evaluate the practical advantages and limitations of the proposed approach.

---

> ### Author Rebuttal · Authors · 2024-08-07
>
> We would like to thank the reviewer for their comments and suggestions to help us improve the quality of our work. Below we address the questions and other points raised in the review one by one.
>
> **1. The empirical validation, while convincing, is somewhat limited in scope. The experiments are conducted on a few specific tasks and datasets, which may not fully capture the diversity of real-world applications. Datasets with different characteristics, such as varying levels of noise, feature distributions, and dimensionalities, could provide a more comprehensive assessment of the proposed methods.**
>
> We varied levels of noise, feature distributions and dimensionalities in the synthetic examples presented in the paper. This being said, in the final manuscript, we will extend our experiments to some real-world datasets, showcasing that universality holds for them, too, and would like to thank the reviewer for making this point.
>
> **2. The comparison with existing transfer learning methods is limited. The paper could benefit from a more comprehensive comparison to highlight the advantages and limitations of the proposed approach relative to current state-of-the-art methods.**
>
> We do not claim to propose a novel transfer learning method, but, rather, to analyze theoretically a popular approach to transfer learning. We apologize for any confusion and will clarify this in the final version of the manuscript.
>
> **3. The paper provides certain properties of linear models trained on large datasets can be transferred to new tasks with minimal data, but in real applications, is there any indication, when transfer methods are useful?**
>
> To make our analyses applicable to the more real life scenarios, we would need to extend them to the deep models. We believe it should be possible to do by linearizing the networks via the Neural tangent Kernels approach but it requires verifying certain technical properties and is left as a subject for future work.
>
> **1. Can you provide more details on the boundary conditions under which the universality principle might fail? Are there specific cases or distributions where this principle does not hold?**
>
> We provide an example from Han, Q. and Shen, Y. (2023). ``Universality of regularized regression estimators in high dimensions", *The Annals of Statistics*. Take $x$ to be of the form $qg$, where $q$ is a Bernoulli random variable (1 with probability $p$ and 0 with probability $1-p$) and $g \sim \mathcal{N}(0, \frac{I_d}{d})$, them Han and Shen show that Gaussian universality does not hold for the ridge regression objective. In terms of our assumptions, this distribution violates the variance property from Definition 2.5.
>
> **2. Can you elaborate on the choice of datasets and tasks used for empirical validation? Are there plans to test the proposed method on more diverse and complex datasets?**
>
> We would be happy to extend our experiments to the real-world datasets, comparing the test errors for models learned from these datasets and from the Gaussian mixtures with matching means and covariances. We will include the results in the final version of the paper and we appreciate the reviewer's suggestion.

---

### Official Review · Reviewer_LHAw · 2024-07-08

**Soundness:** 3
**Presentation:** 2
**Contribution:** 2
**Rating:** 5
**Confidence:** 3

**Summary:**

The paper studies model-based transfer learning. In this setting, a model is pre-trained on the source data, and the learner aims to fine-tune it on the target data by running SGD initialized at the pre-trained model. The paper focuses on linear regression and classification, aiming to generalize the assumption of Gaussianity to general distributions and investigate the scenarios where fine-tuning on the target data can or cannot be beneficial.

**Strengths:**

The paper derives a universality result that allows replacing Gaussian data with more general distributions. It then analyzes the generalization error in linear regression and classification for SGD initialized at a model pre-trained on the source data. The study shows that when the noise in the data is high, fine-tuning can be harmful, and just using the model pre-trained on the source data results in a lower error. Conversely, when the noise is less than the error of the model pre-trained on the source data, transfer learning improves test performance.

**Weaknesses:**

The paper attempts to generalize the assumption of Gaussianity but is limited to the linear model, which is inherently restrictive. Furthermore, it focuses exclusively on a specific algorithm—SGD initialized at the model pre-trained on the source data—without demonstrating the optimality of these results or determining whether better results are achievable with alternative algorithms.

**Questions:**

It appears that there are some parameters in the paper that have not been defined.

In line 194, "p" is used without definition.

In equation (7), \kappa is referenced without definition.

In line 206, it's unclear what "p(r)" represents.

**Limitations:**

The paper does not have any potential negative societal impact.

---

> ### Author Rebuttal · Authors · 2024-08-07
>
> We would like to thank the reviewer for their comments and suggestions to help us improve the quality of our work. Below we address the questions and other points raised in the review one by one.
>
> **The paper attempts to generalize the assumption of Gaussianity but is limited to the linear model, which is inherently restrictive. Furthermore, it focuses exclusively on a specific algorithm—SGD initialized at the model pre-trained on the source data—without demonstrating the optimality of these results or determining whether better results are achievable with alternative algorithms.**
>
> For extending the current approach to deep architectures, one could linearize the network using the Neural Tangent Kernel approach and carefully verify all necessary technical details. We leave this as a direction for future work. We do not prove universality only for the SGD but rather for a certain class of  *objectives* (described in the statement of Theorem 1) that could potentially be optimized using other optimization methods and would still exhibit universality. Our results apply to the objective coming from the implicit regularization property of the SGD but they extend far beyond that, and also even this objective could potentially be solved by other methods. Finally, studying optimality of SGD for transfer learning goes beyond the scope of this work. We only claim to analyze rigorously a popular approach to transfer learning.
>
> **In line 194, "p" is used without definition.**
>
> Line $194$ was meant to say that the empirical spectral densities $\hat{p}_N$ converge to some limit, that we will denote by $p$. That is, $p$ is the limiting spectral density.  We are sorry for the confusion it arose and will make it more clear in the final version.
>
> **In equation (7), $\kappa$ is referenced without definition.**
>
> $\kappa$ is defined in the beginning of  line $104$ as the ratio of the number of parameters and the number of data points in the target distribution. This being said, we understand that it's a good practice to remind the reader of the notation within the theorem itself and will do so in the final version of the paper.  We will clarify this in the final manuscript.
>
> **In line 206, it's unclear what "p(r)" represents.**
>
> This is the same limiting spectral density $p$ as in line $194$ written as a function of its argument $r$.

---

### Official Review · Reviewer_vrcM · 2024-07-17

**Soundness:** 4
**Presentation:** 3
**Contribution:** 2
**Rating:** 7
**Confidence:** 3

**Summary:**

This work considers transfer learning (fine-tuning) for linear models in the over-parameterized regime in the proportional regime $k / n \rightarrow \infty$.  In this setting, gradient descent converges to the solution a convex optimization problem with linear constraints. This work builds on this result to show that for a distribution $P$, the test error converges to the test error of a Gaussian distribution with mean and covariance equal to that of $P$---in other words, universality holds.

**Strengths:**

This work brings a fairly atypical and and relatively unexplored perspective to transfer learning. The results hold for a large family of distributions (theorem 1), although limited to linear models in the infinite width regime. Finally, the results indicate that Transfer learning depends on the noise levels of the problem, i.e., one should choose to fine-tune the model if the noise level is very high (remark 1). I unfortunately did not check all the proofs carefully but appreciate the authors attempt at distilling the results into the main paper despite the depth of results.

**Weaknesses:**

My main concern is that the setting is not conducive to studying transfer learning. Transfer learning is a fundamentally small-data phenemenon, i.e., the target dataset usually contains a limited number of samples. However, the results (in the proportional regime) do not include: (1) The number of samples in each task or (2) the distance between the two tasks. The only quantity that features in the equations is the noise level.

I also believe the presentation could be better. For example, some text explaining theorem 1 (in english) or explaining what $\theta$ and and $t$ represent in theorem 2 could be helpful to a reader. I also wonder why the authors chose to call in model-based transfer. I have not heard of this specific term before and it would help if the authors cited previous references to model-based transfer learning. Isn't fine-tuning a more appropriate term.

Finally, the looming question is the connection to deep networks. While perhaps out of the scope of this paper, there are questions about how this affects or relates to the practice of deep learning. Are linear models in the proportional regime informative for how we should do Transfer learning in practice?

**Questions:**

1. Why is renormalization needed in in the classification problem setting?
2. In definition 2, the authors claim that the assumptions are satisfied in practice. Could the authors expand on line 156-160 and explain the results of Seddik et al. ?
3. How do I interpret quantities like $t$ and $\theta$ in theorem 2?

**Limitations:**

Limitations are addressed but the authors could comment on the assumptions and their limitations.

---

> ### Author Rebuttal · Authors · 2024-08-07
>
> We would like to thank the reviewer for their comments and suggestions to help us improve the quality of our work. Below we address the questions and other points raised in the review one by one.
>
> **My main concern is that the setting is not conducive to studying transfer learning. Transfer learning is a fundamentally small-data phenemenon, i.e., the target dataset usually contains a limited number of samples. However, the results (in the proportional regime) do not include: (1) The number of samples in each task or (2) the distance between the two tasks. The only quantity that features in the equations is the noise level.**
>
> We apologize for any possible lack of clarity in the exposition. We look at the problem in the following way: instead of explicitly defining the source distribution, and the number of samples from it, we capture the effect of training on the source distribution through the pre-trained solution $w_0$. We use $w_0$ as the initialization point when fine tuning on the target distribution. Then, for example, in the context of regression [Theorem 2, eq.(7)], the distance between the tasks (i.e., the source and target distributions) is essentially reflected by $e_a$, which is defined as the generalization error of the source-trained model, $w_0$, on the target distribution. The number of samples from target distribution is included in our analysis and is reflected by the quantity $\kappa = \frac{d}{n}$, where $d$ is the number of parameters and $n$ is the number of samples. We will do our best to introduce these clarifications in the final version of the paper.
>
> **I also believe the presentation could be better. For example, some text explaining theorem 1 (in english) or explaining what
>  $\theta$ and $t$ and represent in theorem 2 could be helpful to a reader. I also wonder why the authors chose to call in model-based transfer. I have not heard of this specific term before and it would help if the authors cited previous references to model-based transfer learning. Isn't fine-tuning a more appropriate term.**
>
> Theorem 1 says that in a certain technical sense we can examine different properties of the weights trained on data matrix $A$ satisfying Definition 2 by relating them to the properties of the weights trained on its matching Gaussian matrix $G$ (see Definition 3). We will add an expanded version of this explanation into the final version and would like to thank the reviewer for making this point.  Please refer to the answer to the question 3 below for a discussion about $\theta$ and $t$ from Theorem 2.
> We learned of the terms "instance-based transfer learning" and "network-based transfer learning" from the review paper by C. Tan et al. "A survey on deep transfer learning". We then preferred to use "model-based" instead of "network-based" as we introduced these terms only to explain that there are two main approach to transfer learning, one of which (instance-based) mixes the target and the source distributions together and the other keeps the distributions untouched and transfers the *model* from the source to the target. This being said, we agree that calling it fine-tuning could have been a better choice and we will take it into account for the final version.
>
> **Finally, the looming question is the connection to deep networks. While perhaps out of the scope of this paper, there are questions about how this affects or relates to the practice of deep learning. Are linear models in the proportional regime informative for how we should do Transfer learning in practice?**
>
> As far as we understand, sometimes in practice one pre-trains a deep model and then fine-tunes only the last layer on the target distribution (see, e.g., "Last-Layer Fairness Fine-tuning is Simple and Effective for Neural Networks" by
> Mao, Yuzhen et al.), essentially making the model linear for our purposes. This is the most straightforward potential application of our results to the real world models. Another way to extend our results to deep models could be linearizing them using the Neural Tangent Kernel approach. We leave both of these directions for future work as pursuing them requires careful verification of several technical steps.
>
> **1. Why is renormalization needed in in the classification problem setting?**
>
> We find such a renormalization meaningful because, in the classification setting, the outputs of the model depend only on the direction of the vector of weights but not on its magnitude. More explicitly, $w_0$ and $cw_0$ for $c \ne 0$ have the same performance making them equally good solutions. However, initializing at $w_0$ and $cw_0$ yields different solutions after training on source distributions. Putting all these remarks together, we would like to ensure that the output of the fine-tuning step depends only on the directions of $w_0$, which is exactly what the renormalization step does.
>
> **2. In definition 2, the authors claim that the assumptions are satisfied in practice. Could the authors expand on line 156-160 and explain the results of Seddik et al.?**
>
> The Lipschitz Concentration Property (LCP) defined on line 156 says that for any Lipschitz $\phi: \mathbb{R}^d \to \mathbb{R}$ the tails of $\phi(x)$ decay very fast, where $x \in \mathbb{R}^d$ comes from the data distribution. Seddik et al. show that any distribution generated from a GAN with weight matrices of bounded norm satisfies LCP. This follows from the well-known facts that Gaussian random vectors satisfy LCP and applying a Lipschitz function to a distribution satisfying LCP yields a distribution that also satisfies LCP.
>
> **3. How do I interpret quantities like $t$ and $\theta$ in theorem 2?**
>
>  It does not seem to be possible to explain what $\theta$ and $t$ are in plain English, but these are certain scalars determined solely by the covariance matrix $R_x$ of the data. For example, if  $R_x = r_0I_d$,  then the expressions simplify to $t = ({\frac{d-n}{d}})^2$ and $\theta = \frac{2\sqrt{n}}{r_0 (d-n)}$.

---

> > ### Comment · Reviewer_vrcM · 2024-08-11
> >
> > Thank you for detailed responses to all the questions, in particular, the clarifications for definition 2 and theorem 1 were helpful. Regarding my other concerns, I understand that the target samples are included in the term $\kappa$. However, my broader question is whether interesting results can come out of explicitly modeling the source data as opposed to starting from a pre-trained solution $w_0$. For example, if you look at previous work on transfer learning, for example Ben-David et al. [1], the generalization bounds explicitly model the source distributions which leads to defining a distance between tasks (the $H \Delta H$ divergence). Modeling the source data, can you tell *how* to arrive at a good initialization $w_0$. The existing setup is certainly interesting but relates directly to model initialization than to a typical theoretical setup for transfer learning.
> >
> > I will stay at my current score since I do not see immediate connections to explain or answer questions for deep networks (network don't always operate in the linear regime with NTKs). However, I find the lens of universality to be different from prior theoretical work on transfer learning and support the acceptance of this work.
> >
> > [1]  Ben-David, Shai, et al. "A theory of learning from different domains." Machine learning 79 (2010): 151-175.

---

### Decision · Program_Chairs · 2024-09-25

**Decision:**

Accept (poster)

**Comment:**

This paper studies transfer learning in the sense of finetuning an overparametrized linear model with initial weight w_0 on a target distribution P.

Authors state a universality result as the dimensionality goes to infinity for functions of the optimal weight under assumptions on the randomness of the design matrix of input data matrix. This result uses the Lindeberg approach to show convergence of the optimal weight of a matching Gaussian to the design matrix.

Authors uses this result to quantify the generalization error in regression and classification.

In the regression case, they give a lower bound on the generalization error and show that depending on the noise level it may be harmful to finetune the linear model. Similar analysis is derived for classification, with similar conclusions.

Overall this is a good paper nevertheless , it is not very pedagogic in explaining their assumption nor in explaining Theorem 1 after stating it, or stating clearly in theorem the limit is taken on the dimension. Please incorporate all the feedback from reviewers to clarify assumptions, and to back passing to limits with rigorous arguments.